# SUMOylation stabilizes sister kinetochore biorientation to allow timely anaphase

Xue Bessie Su[1], Menglu Wang[1], Claudia Schaffner[1], Olga O. Nerusheva[1], Dean Clift[1,2], Christos Spanos[1], David A. Kelly[1], Michael Tatham[3], Andreas Wallek[4], Yehui Wu[4], Juri Rappsilber[1,5], A. Arockia Jeyaprakash[1], Zuzana Storchova[4,6], Ronald T. Hay[3], and Adèle L. Marston[1]

During mitosis, sister chromatids attach to microtubules from opposite poles, called biorientation. Sister chromatid cohesion resists microtubule forces, generating tension, which provides the signal that biorientation has occurred. How tension silences the surveillance pathways that prevent cell cycle progression and correct erroneous kinetochore–microtubule attachments remains unclear. Here we show that SUMOylation dampens error correction to allow stable sister kinetochore biorientation and timely anaphase onset. The Siz1/Siz2 SUMO ligases modify the pericentromere-localized shugoshin (Sgo1) protein before its tension-dependent release from chromatin. Sgo1 SUMOylation reduces its binding to protein phosphatase 2A (PP2A), and weakening of this interaction is important for stable biorientation. Unstable biorientation in SUMO-deficient cells is associated with persistence of the chromosome passenger complex (CPC) at centromeres, and SUMOylation of CPC subunit Bir1 also contributes to timely anaphase onset. We propose that SUMOylation acts in a combinatorial manner to facilitate dismantling of the error correction machinery within pericentromeres and thereby sharpen the metaphase–anaphase transition.

## Introduction

Mitosis divides the nucleus to produce two genetically identical daughter cells. Prior to mitosis, DNA replication produces sister chromatids, linked together by the cohesin complex. Sister chromatids are aligned at metaphase, thus allowing spindle microtubules to be captured by kinetochores assembled on centromeres. The correct form of attachment is termed "biorientation," meaning that the kinetochores on the two sister chromatids are attached to microtubules emanating from opposite spindle poles (Tanaka, 2010). Biorientation creates tension, because cohesin holding sister chromatids together resists the pulling force of microtubules. The fulfilment of biorientation allows securin degradation and, consequently, activation of the protease separase, which cleaves cohesin, triggering sister chromatid separation (reviewed in Marston [2014]).

The conserved shugoshin protein plays key roles in promoting biorientation in mitosis and preventing cell cycle progression where biorientation fails (Indjeian et al., 2005). Budding yeast possesses a single shugoshin gene, *SGO1*. Sgo1 localizes to both the core ~125-bp centromere, where the kinetochore resides, and the surrounding ~20-kb cohesin-rich chromosomal region called the pericentromere (Kiburz et al., 2005). The kinetochore-localized Bub1 kinase promotes

Sgo1 enrichment at the pericentromere through phosphorylation of S121 on histone H2A (Fernius and Hardwick, 2007; Yamagishi et al., 2010; Nerusheva et al., 2014). Sgo1, in turn, recruits condensin and protein phosphatase 2A, PP2A-Rts1, to the pericentromere and maintains the chromosome passenger complex (CPC) containing Aurora B kinase at centromeres during mitosis (Verzijlbergen et al., 2014; Peplowska et al., 2014). Condensin at pericentromeres is thought to bias the conformation of the sister chromatids to favor biorientation. The CPC recognizes erroneous microtubule–kinetochore attachments and destabilizes them, thereby maintaining the activity of the spindle assembly checkpoint (SAC) to prevent anaphase entry (reviewed in Foley and Kapoor [2013]). In vertebrate cells, PP2A-B56 protects cohesin in pericentromeres from removal via a nonproteolytic mechanism that is independent of separase, called the prophase pathway (McGuinness et al., 2005; Liu et al., 2013). In budding yeast, PP2A-Rts1 is recruited by shugoshin despite the absence of the prophase pathway (Verzijlbergen et al., 2014; Peplowska et al., 2014; Eshleman and Morgan, 2014). Instead, PP2A-Rts1 has been implicated in ensuring the equal segregation of sister chromatids during mitosis, since mutants failing to recruit PP2A-Rts1 to the centromere

---

[1]Wellcome Centre for Cell Biology, Institute of Cell Biology, University of Edinburgh, Edinburgh, UK;   [2]Laboratory of Molecular Biology, Medical Research Council, Cambridge, UK;   [3]Centre for Gene Regulation and Expression, University of Dundee, Dundee, UK;   [4]Max Planck Institute of Biochemistry, Martinsried, Germany;   [5]Institute of Biotechnology, Technische Universität Berlin, Berlin, Germany;   [6]Technische Universität Kaiserslautern, Kaiserslautern, Germany.

Correspondence to Adèle L. Marston: adele.marston@ed.ac.uk



are unable to respond to a lack of intersister kinetochore tension and missegregate chromosomes upon recovery (Peplowska et al., 2014; Eshleman and Morgan, 2014).

Sgo1 both directs and responds to cell cycle cues as chromosomes establish and achieve biorientation, upon which anaphase entry is triggered. A key signal that biorientation has occurred is the tension-dependent removal of Sgo1 and its associated proteins from the pericentromere during metaphase (Nerusheva et al., 2014; Eshleman and Morgan, 2014). Upon anaphase I onset, Sgo1 is ubiquitinated and degraded by anaphase promoting complex/cyclosome (APC/C)-Cdc20 (Eshleman and Morgan, 2014; Marston et al., 2004). Similarly, human shugoshin is degraded in anaphase as a result of APC/C-Cdc20 activity (Salic et al., 2004). Shugoshin can be stabilized by mutation of its APC-Cdc20–dependent destruction sequence in both yeast and human cells; however, this does not impair the metaphase–anaphase transition (Liu et al., 2013; Eshleman and Morgan, 2014; Karamysheva et al., 2009). Nevertheless, *SGO1* overexpression results in a pronounced metaphase delay and a block to cohesin cleavage (Clift et al., 2009), suggesting the existence of a degradation-independent mechanism of Sgo1 inactivation. The *SGO1* overexpression-induced metaphase delay is abrogated by deletion of *BUB1*, indicating that the delay requires pericentromere-localized Sgo1 (Clift et al., 2009).

Here, we identify SUMOylation as a mechanism that inactivates the pericentromeric signaling hub and thereby ensures timely anaphase onset. Small ubiquitin-like modifier (SUMO) is a 12-kD protein that is covalently added to lysine residues of SUMO substrates. SUMOylation is performed by the sequential activities of E1 activation enzyme, E2 conjugation enzyme, and E3 ligase (Hay, 2005). We isolated the E3 SUMO ligase, Siz2, as a negative regulator of Sgo1 through a genetic screen. In the absence of Siz1/Siz2, Sgo1 is stabilized, CPC removal is insufficient, and biorientation is unstable. Furthermore, we found that SUMOylation of a CPC component, Bir1, is also required for timely anaphase entry. We propose that Siz1/Siz2 act to disrupt protein interactions, thereby moderating the pericentromeric signaling pathway to promote the metaphase–anaphase transition.

## Results

### SUMO ligases reverse the effects of SGO1 overexpression

To identify negative regulators of Sgo1, we screened for high copy suppressors of the poor growth caused by *SGO1* overexpression (Clift et al., 2009). We recovered a number of plasmids that improved the growth of cells carrying multiple copies of *SGO1* under galactose-inducible control (*pGAL-SGO1*; Fig. S1 A; Table S1), including one carrying a ~5-kb fragment containing truncated *SLP1* and *PUP1* together with full-length *ISN1* and *SIZ2* (Fig. 1 A). *SIZ2*, encoding one of three budding yeast SUMO E3 ligases and sharing functional redundancy with its paralog, Siz1, is an attractive candidate for an Sgo1 antagonist, since Siz1/Siz2 were previously implicated in chromosome segregation and cell division (Johnson and Gupta, 2001; Makhnevych et al., 2009; Montpetit et al., 2006; Takahashi et al., 2006). We used live-cell imaging to validate and quantify the rescue of *SGO1* overexpression by *SIZ2* (Fig. 1 B). Time elapsed between mitotic entry

(emergence of two Spc42-tdTOMATO foci) and anaphase onset (dispersal of Cdc14-GFP from the nucleolus) was significantly reduced in a *pGAL1-SGO1* strain that carried additional copies of *SIZ2* under copper-inducible control (*pCUP1-SIZ2*) at an ectopic site (Fig. 1, B and C). Hence, Siz2 promotes anaphase onset by antagonizing Sgo1.

Siz1/Siz2 were previously implicated in the metaphase–anaphase transition, as *siz1Δ siz2Δ* cultures accumulate large-budded cells (Johnson and Gupta, 2001). We examined and quantified the metaphase delay by two independent methods. In a mitotic time course, after synchronous release from G1, metaphase spindles accumulated and securin (Pds1) persisted in *siz1Δ siz2Δ* cells (Fig. 1 D). Similarly, single-cell analysis by live-cell imaging showed that *siz1Δ siz2Δ* cells progressed through metaphase slowly, which was measured as the time between formation of a short metaphase spindle (Tub1-YFP) and Cdc14-GFP dispersal from the nucleolus (Fig. 1, E and F). The single *siz1Δ* and *siz2Δ* mutants showed a relatively mild delay compared with the double mutant (Fig. 1 F), suggesting that they act redundantly in metaphase progression. The delay is dependent on Sgo1, as *siz1Δ siz2Δ sgo1Δ* showed a less severe delay in metaphase, similar to that of *sgo1Δ* (Fig. 1 D). To avoid potential cumulative effects of *sgo1Δ*, we also depleted Sgo1 during a single cell cycle using the auxin-induced degron in cells carrying Tub1-YFP and Cdc14-GFP. Addition of auxin significantly reduced the metaphase delay in a time course analysis of *siz1Δ siz2Δ*, confirming the idea that the Sgo1-associated signaling pathway mediates the effect of Siz1/Siz2 on the metaphase–anaphase transition (Fig. 1 G). Deletion of *CDC55*, a PP2A-regulatory subunit that was previously shown to rescue the metaphase delay of *pGAL-SGO1* (Clift et al., 2009), also rescued the metaphase delay in *siz1Δ siz2Δ* (Fig. S1 B), further indicating that ectopic Sgo1 activity is at least partially responsible for the metaphase delay of cells lacking *SIZ1* and *SIZ2*. Overall, we conclude that Siz1/Siz2 promote anaphase onset by antagonizing Sgo1.

### A chromatin-associated pool of Sgo1 is SUMOylated by Siz1 and/or Siz2

To determine whether Siz1/Siz2 might counteract Sgo1 by direct SUMOylation, we assayed Sgo1 SUMO conjugates in vivo. Cells carrying *SGO1-6HA* and a plasmid producing His-tagged yeast SUMO (7His-Smt3) were lysed under denaturing conditions, and SUMOylated proteins were isolated using nickel affinity chromatography. Anti-HA Western blotting identified two slow-migrating bands corresponding to SUMOylated Sgo1 (Fig. 2 A). Sgo1 SUMOylation was reduced in *siz1Δ* and *siz2Δ* single mutants and was eliminated in the double mutant (Fig. 2 A; unmodified Sgo1 also bound nonspecifically to the resin in each condition). Direct SUMOylation of purified Sgo1 (Fig. S2 A) by Siz1 or Siz2 was further confirmed by in vitro SUMOylation assays (Fig. 2 B). Interestingly, *Schizosaccharomyces pombe* shugoshin, Sgo2, is also SUMOylated (Fig. 2 C), raising the possibility that it might be regulated in a similar way in that organism.

We next analyzed the timing and location of Sgo1 SUMOylation. Sgo1 is recruited to the pericentromeric chromatin during S phase, released into the nucleoplasm upon sister kinetochore biorientation at metaphase, and degraded in anaphase

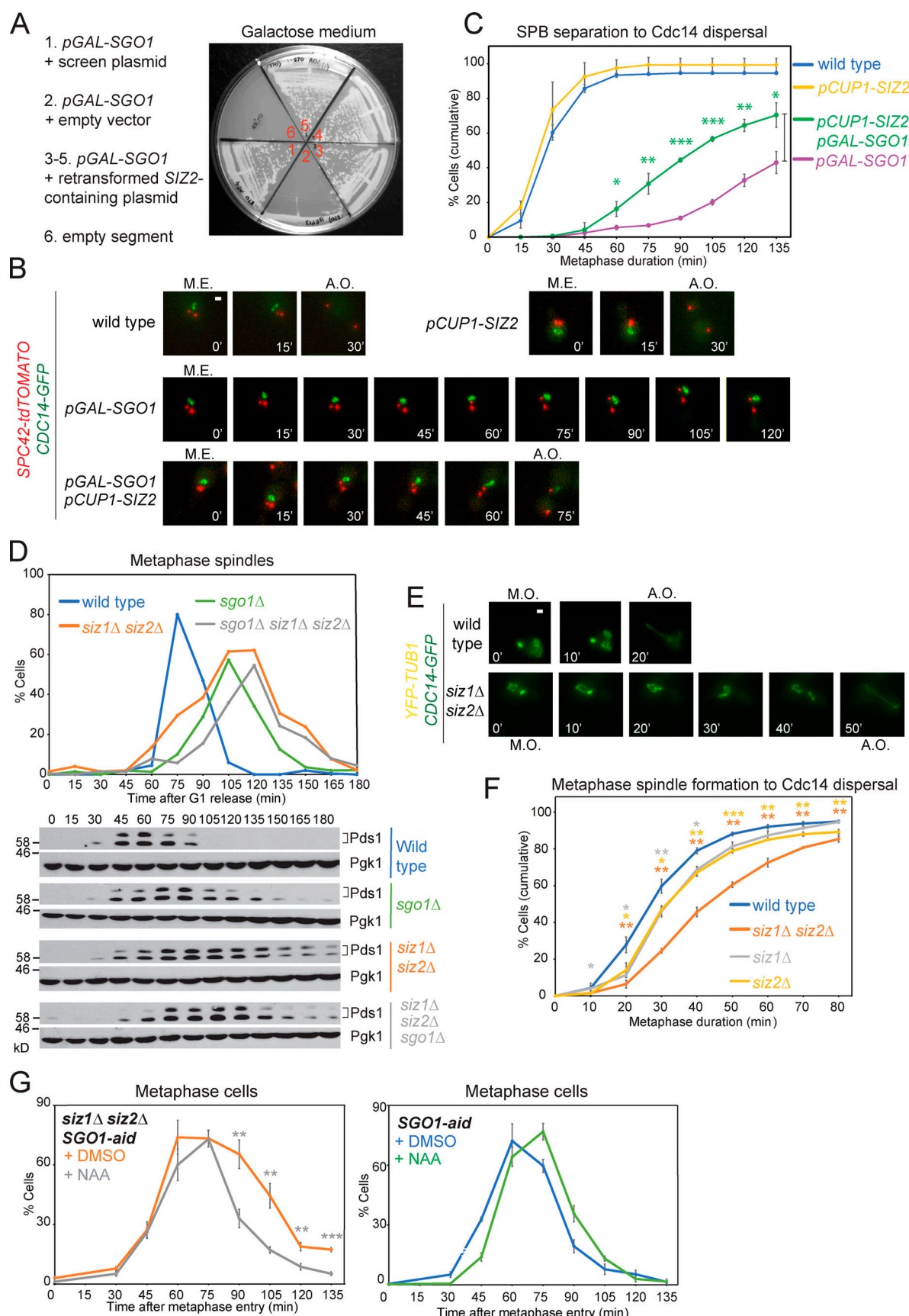

Figure 1. **SUMO ligases promote Sgo1 inactivation to allow timely anaphase progression. (A)** Overexpression of *SIZ2* rescues the slow-growth phenotype of *SGO1* overexpression. A *pGAL-SGO1* (AMy870) strain carrying empty vector (AMp67) or with yEP13-(*SLP1*) *ISN1 SIZ2* (*PUP1*) (AMp1435) was streaked

onto medium containing galactose. **(B and C)** *SIZ2* overexpression partially rescues the metaphase delay of *SGO1*-overexpressing cells. Wild-type (AMy24115), *pGAL-SGO1* (AMy27596), *pGAL-SGO1 pCUP1-SIZ2* (AMy27738), and *pCUP1-SIZ2* (AMy27952) strains carried Spc42-tdTOMATO and Cdc14-GFP. Cells were initially grown in 2% raffinose + 25 μM copper sulfate and synchronized in G1. Following release from G1, 0.2% galactose was added to induce *pGAL-SGO1* expression. Mitotic entry (M.E.) was determined when the spindle pole bodies separated, and anaphase onset (A.O.) was determined when Cdc14-GFP was released from the nucleolus. **(B)** Representative images. Images were acquired as described in Materials and methods and were maximum-intensity projected for Cdc14-GFP and Spc42-tdTOMATO. Scale bar = 1 μm. **(C)** Metaphase progression in different strains. Typically, 50–100 mitotic cells were analyzed for each strain in each experiment. The average percentages of cells that completed metaphase within the indicated duration were calculated from three independent experiments, and the error bars represent standard error. Statistics: one-tailed Student's *t* test (*, P < 0.05; **, P < 0.01; ***, P < 0.001). **(D)** Deletion of *SGO1* reduces the metaphase delay of *siz1Δ siz2Δ* cells. Wild-type (AMy1290), *sgo1Δ* (AMy8466), *siz1Δ siz2Δ* (AMy8465), and *siz1Δ siz2Δ sgo1Δ* (AMy12110) strains carrying *PDS1-6HA* were released from G1 arrest. Spindle morphology was scored after anti-tubulin immunofluorescence, and the percentages of metaphase spindles (<2 μm) are shown (top). Pds1 levels were analyzed by anti-HA Western blot (bottom). Pgk1 is shown as a loading control. **(E and F)** *siz1Δ siz2Δ* cells are delayed in metaphase. Wild-type (AMy24174), *siz1Δ siz2Δ* (AMy24313), *siz1Δ* (AMy29680), and *siz2Δ* (AMy29681) strains carried *YFP-TUB1* and *CDC14-GFP*. The formation of a metaphase spindle marked metaphase onset (M.O.), and the dispersal of Cdc14-GFP from the nucleolus marked anaphase onset (A.O.). **(E)** Representative images showing the metaphase delay in *siz1Δ siz2Δ* cells. Images were acquired as described in Materials and methods and were maximum-intensity projected for Cdc14-GFP and YFP-Tub1. Scale bar = 1 μm. **(F)** *siz1Δ siz2Δ* causes a pronounced delay in metaphase. The single mutants showed a mild but significant delay in metaphase. Shown are the average values of three to four independent experiments, and error bars represent standard error. Statistics: one-tailed Student's *t* test (*, P < 0.05; **, P < 0.01; ***, P < 0.001). **(G)** Conditional degradation of Sgo1 lessened the metaphase delay of *siz1Δ siz2*. *SGO1-aid* (AMy29273) and *SGO1-aid siz1Δ siz2Δ* (AMy29272) strains carried *YFP-TUB1* and *CDC14-GFP*. 0.3 mM auxin (naphthalene-1-acetic acid [NAA]) was added 15 min after releasing from G1 and was re-added every hour. Cells were fixed in 3.7% formaldehyde and washed as described in Materials and methods. Metaphase cells were identified based on spindle morphology and sequestered Cdc14. 100–200 cells were analyzed for each time point, and shown are the average results of three independent experiments, with the error bars representing standard errors. Statistics: one-tailed Student's *t* test (**, P < 0.01; ***, P < 0.001).

(Nerusheva et al., 2014). In a mitotic time course, Sgo1 is maximally SUMOylated in metaphase, coinciding with Sgo1 abundance (Fig. 2 D). We further evaluated Sgo1 SUMOylation in a strain in which expression of the APC/C activator Cdc20 was repressed by addition of methionine (*pMET-CDC20*), leading to efficient arrest in metaphase and Sgo1 stabilization (Fig. 2 E). Sgo1 SUMOylation appeared as cells entered metaphase and subsequently diminished, despite Sgo1 stabilization (Fig. 2 E). To test whether diminishing SUMOylation coincides with Sgo1 removal from pericentromeres in response to tension, we arrested cells in metaphase in either the presence or absence of microtubule-depolymerizing drugs (−/+ tension, respectively). While Sgo1 SUMOylation was observed in the absence of tension, it was greatly reduced in the presence of tension (Fig. 2 F). Loss of Bub1 or inactivation of its kinase activity, which is required for phosphorylation of histone H2A-S121 and Sgo1 recruitment to pericentromeres (Fernius and Hardwick, 2007; Kawashima et al., 2010), diminished Sgo1 SUMOylation (Fig. 2 G). Similarly, the *sgo1-100* and *sgo1-700* alleles, which carry point mutations that delocalize Sgo1 from pericentromeres (Verzijlbergen et al., 2014), also markedly reduced Sgo1 SUMOylation (Fig. 2 G). Hence, maximum Sgo1 SUMOylation requires its chromatin association.

## Sgo1 SUMOylation requires its coiled coil

To attempt to identify the sites on Sgo1 that are SUMOylated, we analyzed purified SUMOylated Sgo1 from an overexpression strain (*pGAL-SGO1-biotin acceptor* transformed with *His-SMT3*) by mass spectrometry (MS). Despite extensive efforts, we were unable to confidently identify any SUMOylation sites, for reasons that are unclear. As an alternative approach, we generated a series of N-terminal Sgo1 truncations (Fig. S2 B), keeping the C-terminal chromatin-binding motif intact to avoid loss of SUMOylation due to mislocalization. SUMOylation was abolished in Sgo1-Δ2–208 and was minimal in Sgo1-Δ2–108 (Fig. S2 C). Interestingly, Sgo1-Δ2–40, which retains the conserved coiled-coil domain, was SUMOylated efficiently (Fig. S2 D). Hence, the

coiled-coil domain of Sgo1, which was previously identified as a direct Rts1 binding site (Xu et al., 2009; Verzijlbergen et al., 2014), is required for its maximum SUMOylation.

Consequently, we mutated the Lys residues in the coiled coil (Fig. 3 A) and found that Sgo1-K56R K64R K70R K85R (Sgo1-4R) showed greatly reduced SUMOylation while maintaining similar levels of expression as wild-type Sgo1 (Fig. 3 B). An in vitro SUMOylation assay using purified Sgo1-4R protein (Fig. S2 A) confirmed the importance of the coiled-coil domain of Sgo1 for maximum SUMOylation (Fig. 3 C). While these findings are consistent with the possibility that Sgo1 lysines 56, 64, 70, and 85 are direct conjugation sites for SUMO, we cannot currently rule out an indirect role of these residues in promoting Sgo1 SUMOylation.

We analyzed the physiological consequences of reduced Sgo1 SUMOylation by synchronous release of wild-type and *sgo1-4R* cells from G1 arrest. Compared with wild type, metaphase spindles accumulated and Pds1 was stabilized in *sgo1-4R* cells (∼30 min; Fig. 3 D). The metaphase delay of *sgo1-4R* cells was also confirmed by live-cell imaging (Fig. 3 E). Although *sgo1-4R* caused a more modest delay than *siz1Δ siz2Δ*, it had no additive effect on *siz1Δ siz2Δ* metaphase duration (Fig. 3 E), indicating that the two mutants impact the same pathway. Furthermore, overexpression of *SIZ2* did not rescue the metaphase delay caused by overexpressed *pGAL-sgo1-4R* (Fig. 3 F), in contrast to its efficient rescue of *pGAL-SGO1*, suggesting that Sgo1 SUMOylation at least partially mediates the effect of Siz1/Siz2. Because both SUMOylation and timely anaphase entry were only partially perturbed in *sgo1-4R* (Fig. 3, B–E), henceforth we evaluated the effects of both *siz1Δ siz2Δ* and *sgo1-4R* on the metaphase–anaphase transition.

## SUMO ligases and SUMO-targeted ubiquitin ligases (STUbLs) do not play a major role in Sgo1 stability

How might Siz1/Siz2 promote metaphase progression? One possibility is that they directly target Sgo1 for ubiquitination and proteosomal degradation. Consistently, Sgo1 was stabilized in

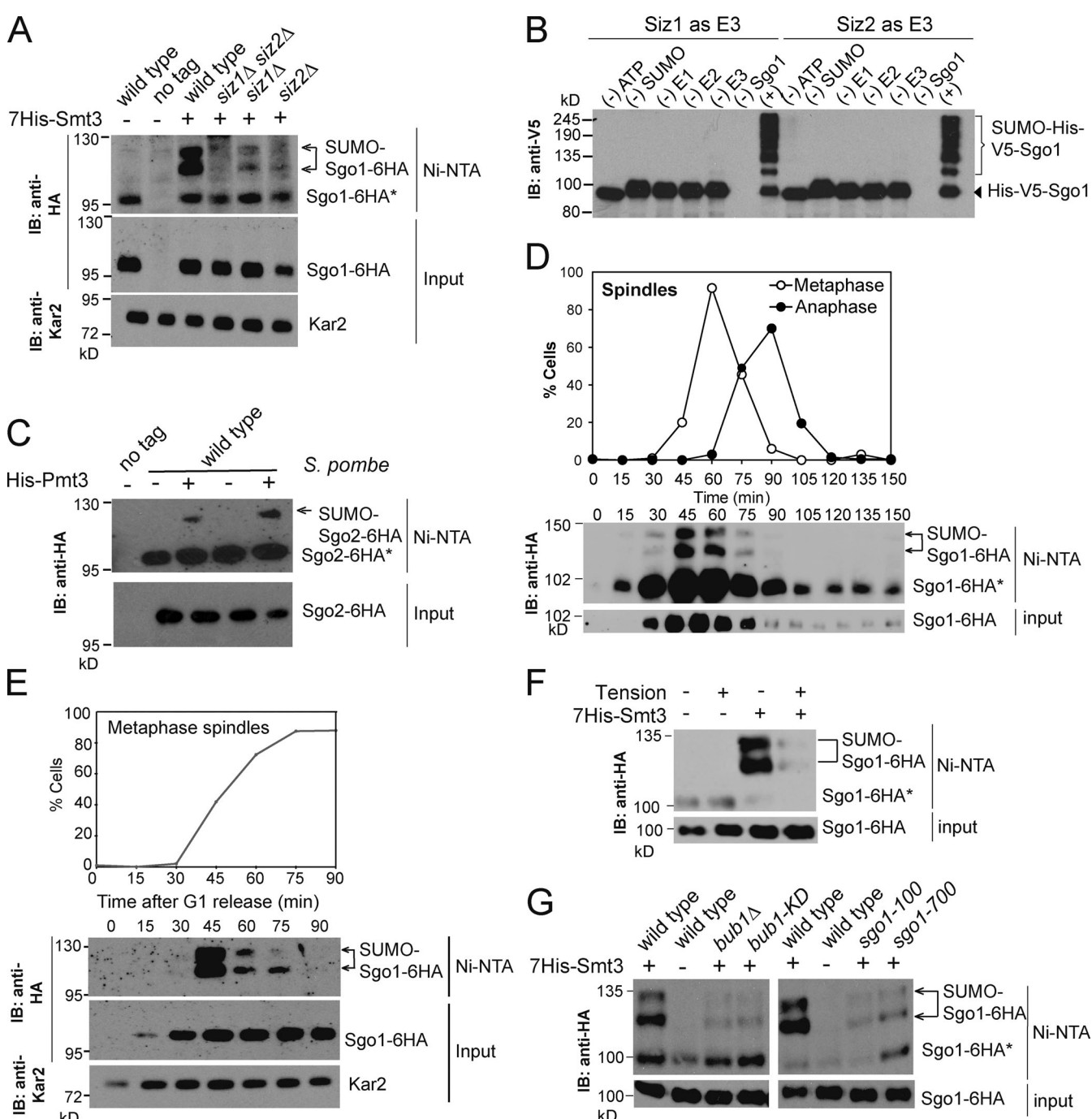

Figure 2. **Sgo1 is SUMOylated, depending on its association with pericentromeres. (A)** Sgo1 is SUMOylated in a Siz1/Siz2-dependent manner. Extracts from untagged (AMy7651), *SGO1-6HA* (AMy7655 and AMy18470), *siz1Δ siz2Δ SGO1-6HA* (AMy8121), *siz1Δ SGO1-6HA* (AMy29477), and *siz2Δ SGO1-6HA* (AMy29478) strains carrying empty vector (pRS426) or with *7×HIS-SMT3* (AMp773) were purified over Ni-NTA resin, and anti-HA immunoblot (IB) was performed on both input and eluate. Arrows and asterisks indicate SUMO-Sgo1-6HA and unmodified Sgo1-6HA, respectively. Note that unmodified Sgo1-6HA occasionally copurified because it was present in insoluble materials and bound nonspecifically to the resin. Kar2 was used as a loading control. **(B)** Sgo1 is SUMOylated by Siz1 and Siz2 in vitro. Purified Sgo1 was incubated with 1 µM of each SUMOylation component (E1, E2, E3, and SUMO) and ATP or missing one component as indicated. The reaction was incubated at 30°C for 3 h. **(C)** Sgo2 in *S. pombe* is SUMOylated. In vivo SUMOylation was assessed for no tag (*S. pombe* AM29) and two independent transformants of Sgo2-6HA (*S. pombe* AM1861) + empty vector (AMp1960) or + *His-Myc-PMT3* (AMp1961). HA-tagged Sgo2 was probed by anti-HA Western blotting. **(D)** Sgo1 SUMOylation occurs in metaphase. Cells carrying *SGO1-6HA* and *7xHIS-SMT3* (AMy7655) were released from G1 and harvested at the indicated intervals, and SUMOylation was analyzed as described in A. Cell cycle stage was monitored by scoring spindle morphology after anti-tubulin immunofluorescence. **(E)** Sgo1 SUMOylation peaks in prometaphase. A *pMET-CDC20 SGO1-6HA* strain transformed with *7HIS-SMT3* (AMy9641) was released from G1 arrest into methionine-containing medium to repress *CDC20* and arrest cells in metaphase. Cells were harvested every 15 min, and SUMO pull-down was performed as described in A. Metaphase cells were identified by scoring spindle morphology after anti-tubulin immunofluorescence. **(F)** Sgo1 SUMOylation is reduced upon the establishment of tension between sister kinetochores. Cells carrying *pMET-CDC20* and either *7xHIS-SMT3* (AM9641) or empty vector (AMy26342) were arrested in metaphase by depletion of Cdc20 in the presence of either benomyl and nocodazole (no tension) or DMSO (tension). **(G)** Chromatin association promotes Sgo1 SUMOylation. Sgo1 SUMOylation was determined as described in A in wild-type (AMy7654 or 7655), *bub1Δ* (AMy10098), *bub1-KD* (catalytically inactive Bub1 kinase, AMy10102), *sgo1-100* (AMy26334), and *sgo1-700* (AMy26336) strains.

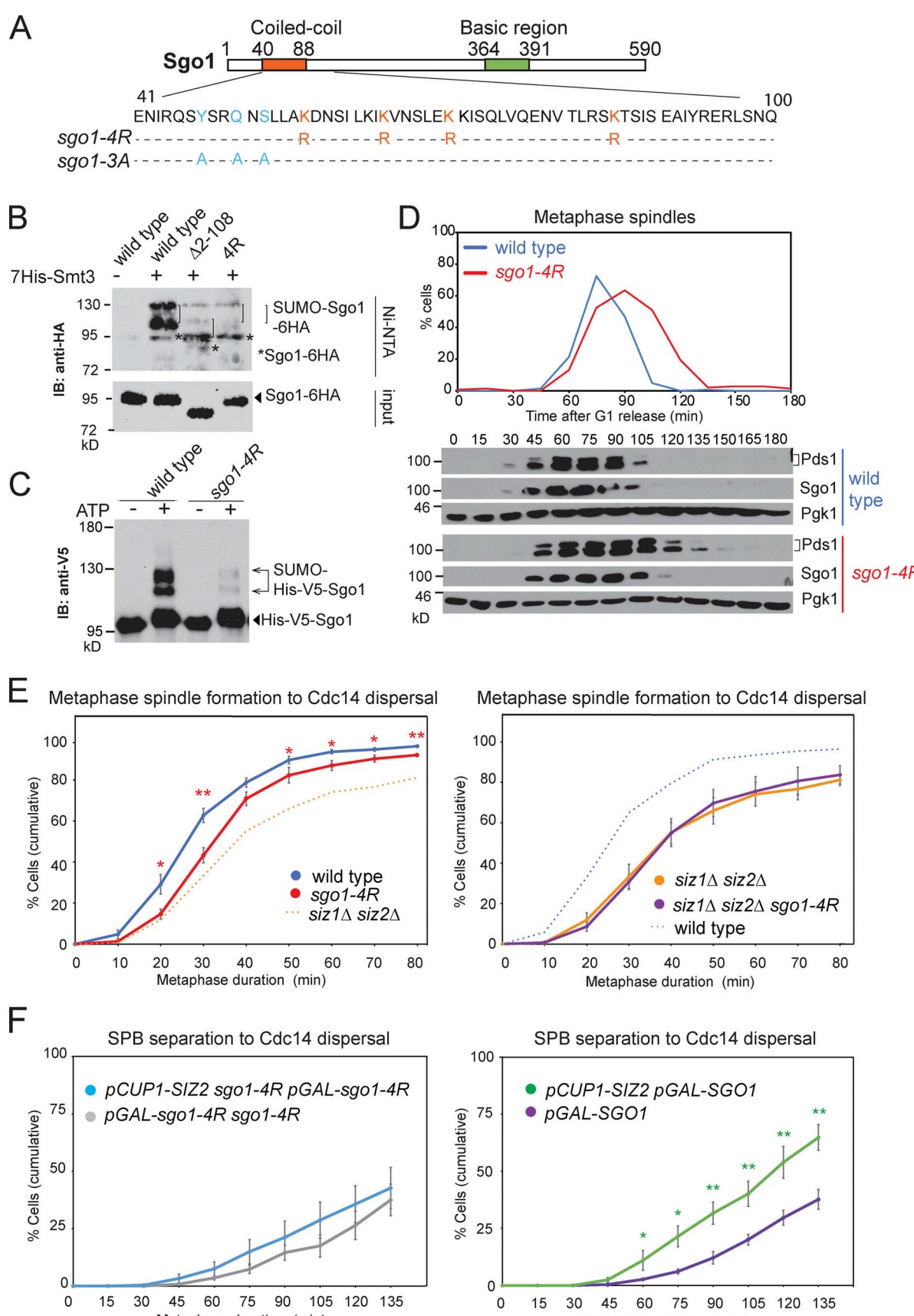

Figure 3. **Sgo1 SUMOylation requires residues within its coiled coil and is important for timely anaphase onset. (A)** Schematic of Sgo1 showing the sequence of the coiled-coil domain (bottom) and residues mutated in the indicated mutants. **(B)** Sgo1 SUMOylation requires residues in the coiled-coil domain.

Strains for in vivo SUMOylation analysis carried Sgo1-6HA and were wild-type (AMy7655), *sgo1-Δ2-108* (AMy14764), and *sgo1-K56R K64R K70R K85R* ('4R', AMy21898). IB, immunoblot. **(C)** Sgo1-4R shows reduced SUMOylation in vitro. Purified Sgo1 and Sgo1-4R proteins were SUMOylated in vitro using 0.1 µM E1–E3, in the presence or absence of ATP. **(D)** The *sgo1-4R* mutant is delayed in metaphase. Cell cycle analysis of wild-type (AMy8467) and *sgo1-K56R K64R K70R K85R*-9Myc (AMy23934) strains carrying *SGO1-9MYC* and *PDS1-3HA* was performed as described in Fig. 1 D. **(E)** Metaphase duration was measured after live-cell imaging of wild-type (AMy24174), *siz1Δ siz2Δ* (AMy24313), *sgo1-4R* (AMy29305), and *sgo1-4R siz1Δ siz2Δ* (AMy29297) strains carrying *YFP-TUB1* and *CDC14-GFP* as described in Fig. 1, E and F. Shown are the average values of five independent experiments. Error bars represent standard errors. Statistics: one-tailed Student's *t* test (*, P < 0.05; **, P < 0.01). **(F)** Overexpression of *SIZ2* does not effectively rescue *sgo1-4R* overexpression. *sgo1-4R pGAL-sgo1-4R* (AMy29525), *sgo1-4R pGAL-sgo1-4R pCUP1-SIZ2* (AMy29524), and *pGAL-SGO1* (AMy27596) and *pGAL-SGO1 pCUP1-SIZ2* (AMy27738) strains carried Spc42-tdTOMATO and Cdc14-GFP. The experiment was performed as described in Fig. 1, B and C. Shown are the average values of five independent experiments. Error bars represent standard errors. Statistics: one-tailed Student's *t* test (*, P < 0.05; **, P < 0.01).

cells lacking E3 ligases, Siz1 and Siz2 (Fig. S1 B), or where E2 Ubc9 function was impaired (Fig. 4 A). We further examined the idea using a cycloheximide pulse-chase experiment, in which ectopic *pGAL-SGO1* expression was induced in G1 phase (Fig. 4 B). Sgo1 was rapidly degraded within 15 min in wild-type cells, while the rate of degradation was moderately slowed in *siz1Δ siz2Δ* (Fig. 4 B).

Slx5 and Slx8 form a heterodimeric STUbL complex that regulates centromeric PP2A-Rts1 levels (van de Pasch et al., 2013; Schweiggert et al., 2016). *slx5Δ* cells were delayed in metaphase and stabilized Sgo1 (Fig. 4 C), like *siz1Δ siz2Δ*. However, His-Ubi pull-down revealed that Sgo1 was ubiquitinated to levels similar to that of the wild type in each of *siz1Δ siz2Δ*, *slx5Δ*, and *sgo1-4R* mutants, whether Sgo1 was overexpressed or endogenously expressed (Fig. 4 D). In contrast, *sgo1-Δdb*, which lacks the APC/C-binding motif (amino acids 494–498), abolishes Sgo1 ubiquitination (Fig. 4 D; Eshleman and Morgan, 2014). Hence, although Siz1/Siz2 has a modest effect on Sgo1 stability, it does not degrade Sgo1 via the canonical STUbL pathway. Instead, the effect may be indirect, for example by modifying components of APC/C (Eifler et al., 2018; Lee et al., 2018).

Importantly, despite its stabilization in anaphase, Sgo1-Δdb did not cause a metaphase delay or exacerbate the delay of *siz1Δ siz2Δ* cells (Eshleman and Morgan, 2014; Fig. 4 E). Hence, Siz1/Siz2-dependent SUMOylation must act at least partially independently of facilitating Sgo1 degradation to promote timely anaphase. Consistently, Sgo1-Δdb undergoes SUMOylation in a manner similar to wild-type Sgo1 (Fig. 4 F).

## SUMOylation is not required for Sgo1 removal from chromatin under tension but prevents its reassociation

Sgo1 is released from pericentromeres under tension (Nerusheva et al., 2014), but whether this is critical for the metaphase–anaphase transition remained unclear. To address this, we analyzed the effect of artificially tethering Sgo1 to the kinetochore. We used heterozygous a/a diploid strains, in which one endogenous copy of *SGO1* was replaced with *pGAL-SGO1-GBP* and one endogenous copy of the kinetochore protein Mtw1 was tagged with GFP (Mtw1-GFP; Fig. 5 A). A low concentration of galactose (0.1%) was added to minimize Sgo1 overexpression. *pGAL-SGO1-GBP MTW1-GFP* cells exhibited a significant metaphase delay compared with *pGAL-SGO1-GBP* alone (Fig. 5 B). In contrast, kinetochore tethering of the Sgo1-3A mutant protein, which was reported to reduce CPC and PP2A-Rts1 binding (Verzijlbergen et al., 2014; Xu et al., 2009), reduced the severity of the metaphase delay (Fig. 5 B). Therefore, tension-dependent

removal of Sgo1, and associated CPC/PP2A-Rts1, is critical for anaphase entry.

We considered the hypothesis that Siz1/Siz2 promote anaphase entry by triggering the spindle tension-dependent release of Sgo1 from the pericentromere. However, chromatin immunoprecipitation (ChIP) followed by quantitative PCR (qPCR) showed that Sgo1 associates with a centromeric site in the absence, but not presence, of spindle tension in metaphase-arrested *siz1Δ siz2Δ* or *sgo1-4R* cells, similar to wild-type cells (Fig. 5, C and D). We further imaged Sgo1-GFP as cells progressed from G1 to anaphase. In wild-type cells, Sgo1-GFP first appeared as a bright focus and dissociated upon splitting of Mtw1-tdTOMATO foci, and these events occurred with similar timing in the SUMO mutants (Fig. 5 E). Intriguingly, however, anaphase onset was delayed after the initial Sgo1 bulk release in the SUMO mutants (Fig. 5 E). Furthermore, we detected reappearance of Sgo1 foci in a small fraction of wild-type cells, which increased in *siz1Δ siz2Δ* and *sgo1-4R* cells, though this was only statistically significant in the case of *siz1Δ siz2Δ* (Fig. 5, F and G). This suggests that Siz1/Siz2 may prevent released Sgo1 from reassociating with chromatin and aberrantly reactivating PP2A/CPC signaling pathways. However, we note that, although coiled-coil–dependent SUMOylation of Sgo1 may contribute, other Siz1/Siz2 targets/residues must also be important in preventing Sgo1 reassociation (Fig. 5 G).

## Siz1/Siz2 and Sgo1 SUMOylation stabilize biorientation

To better understand the nature of the metaphase delay in mutants defective in Sgo1 SUMOylation, we assessed their impact on microtubule–kinetochore attachment, sister kinetochore biorientation, and chromosome segregation. We visualized microtubule (YFP-Tub1)–kinetochore (Mtw1-tdTOMATO) attachment as spindles repolymerized after nocodazole washout. The appearance of bilobed kinetochore foci on the spindle axis occurred with similar timings in *siz1Δ siz2Δ*, *sgo1-4R*, and wild-type cells, suggesting that attachment is not grossly affected in the SUMO mutants (Fig. 6 A).

We subsequently assessed the establishment of sister kinetochore biorientation by analyzing the separation of sister *CEN4-GFP* foci as spindles reformed after nocodazole washout, while maintaining a metaphase arrest (Fig. 6 B). In contrast to *sgo1-3A* cells, which exhibit impaired biorientation (Verzijlbergen et al., 2014), both *siz1Δ siz2Δ* and *sgo1-4R* were proficient in sister kinetochore biorientation (Fig. 6 C). Therefore, establishment of sister kinetochore biorientation occurs independently of Siz1/Siz2-mediated SUMOylation.

Figure 4. **Analysis of factors promoting Sgo1 degradation. (A)** The degradation of Sgo1 depends on SUMO-conjugating protein Ubc9. Wild-type and *ubc9-1* cells were synchronized in nocodazole and released into medium with α-factor to ensure arrest in G1. **(B)** Siz1/Siz2 promote Sgo1 degradation. Cycloheximide pulse chase experiment was performed for *pGAL-SGO1-9MYC* (AMy1392) and *siz1Δ siz2Δ pGAL-SGO1-9MYC* (AMy22584). As described in the scheme, cells were arrested in G1 throughout the experiment, and *pGAL1-SGO1-9MYC* expression was initially prevented by growth of cells in raffinose. Subsequently, a pulse of Sgo1 was provided by the addition of galactose, after which de novo Sgo1-9Myc synthesis was quenched by the addition of glucose (to block *pGAL* expression) and cycloheximide (to block protein synthesis). Representative anti-Myc and anti-Pgk1 (loading control) immunoblots are shown. Signal intensity was quantified using ImageJ. Relative Sgo1 levels were calculated as the ratio of Myc signal to Pgk1 signal, and the ratio was set to 1 for time 0. Shown is the average from three independent experiments, and error bars represent standard error. Statistics: one-tailed Student's *t* test (*, P < 0.05). **(C)** *slx5Δ* cells exhibit a metaphase delay and stabilize Sgo1. Wild-type (AMy8467) and *slx5Δ* (AMy10981) strains carrying *PDS1-3HA* and *SGO1-9MYC* were analyzed as described in Fig. 1 D. **(D)** Sgo1 ubiquitination is dependent on its destruction box and independent of Sgo1 SUMOylation or Slx5/Slx8. Top: Strains with *pGAL-SGO1-9MYC* (AMy27029), *pGAL-sgo1-Δdb-9MYC* (AMy27030), *slx5Δ pGAL-sgo1-Δdb-9MYC* (AMy27031), *siz1Δ siz2Δ pGAL-sgo1-Δdb-9MYC* (AMy27032), *pGAL-sgo1-4R-9MYC* (AMy27033), *pGAL-sgo1-4R-Δdb-9MYC* (AMy27034), *siz1Δ siz2Δ pGAL-SGO1-9MYC* (AMy27035), and *slx5Δ pGAL-SGO1-9MYC* (AMy27036) and carrying His-UBI

(AMp1673) were arrested in G1 in 2% raffinose and Sgo1 overexpression was induced by the addition of 2% galactose. Bottom: Cycling cells of the following strains: *SGO1-9MYC* (AMy29604), *siz1Δ siz2Δ SGO1-9MYC* (AMy29521), *slx5Δ SGO1-9MYC* (AMy29519), *sgo1-4R-9MYC* (AMy29632), and *sgo1-Δdb-9MYC* (AMy29662) carrying His-Ub (AMp1568) or empty vector were analyzed. Ubiquitinated proteins were purified on Ni-NTA resin and Sgo1-9Myc was detected in inputs and elutes by anti-Myc immunoblot (IB). **(E)** Preventing Sgo1 degradation is not sufficient to delay timely anaphase entry. Wild-type (AMy24174), *siz1Δ siz2Δ* (AMy24313), *sgo1-Δdb* (AMy29483), and *siz1Δ siz2Δ sgo1-Δdb* (AMy29484) carried *CDC14-GFP* and *YFP-TUB1*. Metaphase duration was determined as described in Fig. 1, E and F. Shown are the average values of three independent experiments. Error bars represent standard errors. **(F)** Like wild-type Sgo1, Sgo1-Δdb is SUMOylated maximally in prometaphase. Sgo1-Δdb-6HA (AMy18191) carried *His-SMT3* and was released from G1 arrest. Samples were harvested at the indicated time intervals for anti-tubulin immunofluorescence (top) and SUMO pull-down.

We next monitored segregation of a single chromosome as cells progressed into anaphase by live-cell imaging. In this experiment, microtubule attachment was initially depleted by nocodazole, resulting in a single *CEN4-GFP* focus. After nocodazole washout, kinetochores reattach to microtubules and chromosomes biorient, resulting in two *CEN4-GFP* foci. Stable attachment led to further separation of the two *CEN4-GFP* foci and eventual segregation to opposite poles in anaphase (Fig. 6 D). If biorientation is unstable, the two *CEN4-GFP* foci reassociate before segregating to opposite poles, which we refer to as "switching" (Fig. 6, D and E). In *sgo1Δ* and *sgo1-3A* cells, compared with wild type, the emergence of two *CEN4-GFP* foci was delayed and missegregation events were increased, but increased switching was not observed (Fig. 6, F–I). In contrast, *siz1Δ siz2Δ* and *sgo1-4R* mutants were proficient in the initial establishment of biorientation and accurate segregation of *CEN4*, but the bioriented sister chromatids showed an elevated frequency of switching (Fig. 6, F–I). This suggests that the metaphase delay in cells lacking Siz1/Siz2, or with reduced Sgo1 SUMO, is likely due to an inability to maintain the bioriented state, leading to a futile cycle of detachment and reattachment. To avoid the confounding effect of extended metaphase in the SUMO-deficient strains, we also assessed dot switching in cells arrested in metaphase (by depletion of Cdc20) and observed significant increases in dot-switching frequency in the SUMO mutants (Fig. 6 J). Notably, switching rates were similar in *siz1Δ siz2Δ* and *sgo1-4R* cells (Fig. 6, F and J), suggesting that Siz1/Siz2 prevents futile attachment/detachment cycles to allow timely anaphase onset through Sgo1 SUMOylation.

Despite their unstable biorientation phenotype, the *siz1Δ siz2Δ* and *sgo1-4R* cells ultimately segregate *CEN4-GFP* accurately (Fig. 6 I). Consistently, both *siz1Δ siz2Δ* and *sgo1-4R* retained a *CEN*-containing plasmid at the same rate as the wild-type cells (Fig. 6 K) and grew similarly as wild type (*siz1Δ siz2Δ*) or mildly improved growth (*sgo1-4R*) on medium containing microtubule poison benomyl (Fig. 6 L), unlike *sgo1Δ* cells. Therefore, despite the increase in detachment events in the SUMO mutants, the error-correction pathway remains functional in these mutants to ensure accurate chromosome segregation.

### The metaphase delay in *siz1Δ siz2Δ* depends on CPC/SAC

Unstable biorientation generates unattached kinetochores, leading to engagement of the SAC, which halts the metaphase–anaphase transition. Consistently, deletion of the SAC component, *MAD2*, rescued the metaphase delay of *siz1Δ siz2Δ* and *sgo1-4R* cells (Fig. 7, A–C), suggesting that unstable attachments in these SUMO mutant cells cause the metaphase delay by activation of the SAC.

The Aurora B kinase component of the CPC phosphorylates components of the outer kinetochore to generate unattached kinetochores. Sgo1 maintains CPC around the centromeres, allowing tension sensing and error correction. Conditional degradation of CPC component Bir1 using an auxin-induced degron in a strain carrying Cdc14-GFP Tub1-YFP partially but significantly rescued the metaphase delay in *siz1Δ siz2Δ* (Fig. 7 D), suggesting that the delay in *siz1Δ siz2Δ* is at least in part imposed by CPC-dependent error correction. Furthermore, the *sgo1-3A* mutant, which fails to maintain CPC at the centromeres (Verzijlbergen et al., 2014), rescued the metaphase delay in *siz1Δ siz2Δ* (Fig. 7 E). This supports the idea that Sgo1-dependent retention of CPC at centromeres impedes anaphase onset in the absence of Siz1/Siz2-mediated SUMOylation.

CPC relocalizes from centromeres to the spindle midzone upon the establishment of biorientation (Buvelot et al., 2003; Pereira and Schiebel, 2003). Increased levels of CPC could activate the SAC even when tension-generating biorientation has been established. We imaged Ipl1-GFP and its colocalization with Mtw1-tdTOMATO as cells progressed from G1 to anaphase. In wild-type cells, Ipl1-GFP colocalized with Mtw1-tdTOMATO when the two kinetochores were close together, but as the interkinetochore distance increased, kinetochore-associated Ipl1-GFP diminished (Fig. 8, A and B). As reported, the kinetochore localization of Ipl1 was partially dependent on Rts1 (Fig. 8 B; Peplowska et al., 2014). In contrast, the levels of kinetochore-localized Ipl1-GFP were significantly increased in *siz1Δ siz2Δ* and *sgo1-4R* cells, for almost all interkinetochore distances (Fig. 8 B). We independently verified this result by ChIP and found a modest but statistically significant increase in chromatin-associated Ipl1 when the sister chromatids were under tension (Fig. 8, C and D). The similar impact of *sgo1-4R* and *siz1Δ siz2Δ* on Ipl1-GFP dissociation from kinetochores (Fig. 8, B and C) suggests that Sgo1 SUMOylation is the key role of Siz1/Siz2 in reducing the levels of kinetochore-associated CPC upon biorientation.

### Sgo1 SUMOylation reduces PP2A-Rts1 binding

A simple explanation for the effect of SUMOylation on Ipl1 kinetochore association could be that Ipl1 is stabilized in the absence of SUMOylation, leading to its hyperactivity. However, total Ipl1 levels were unchanged throughout mitosis in the wild-type or SUMO-deficient strains (Fig. 8 E), arguing against the idea that Ipl1 stability is regulated by SUMOylation. In budding yeast, the phospho-H3T3–dependent pathway of CPC recruitment is absent, and instead, CPC recruitment occurs both through direct physical interactions with kinetochore components (Fischböck-Halwachs et al., 2019; García-Rodríguez et al.,

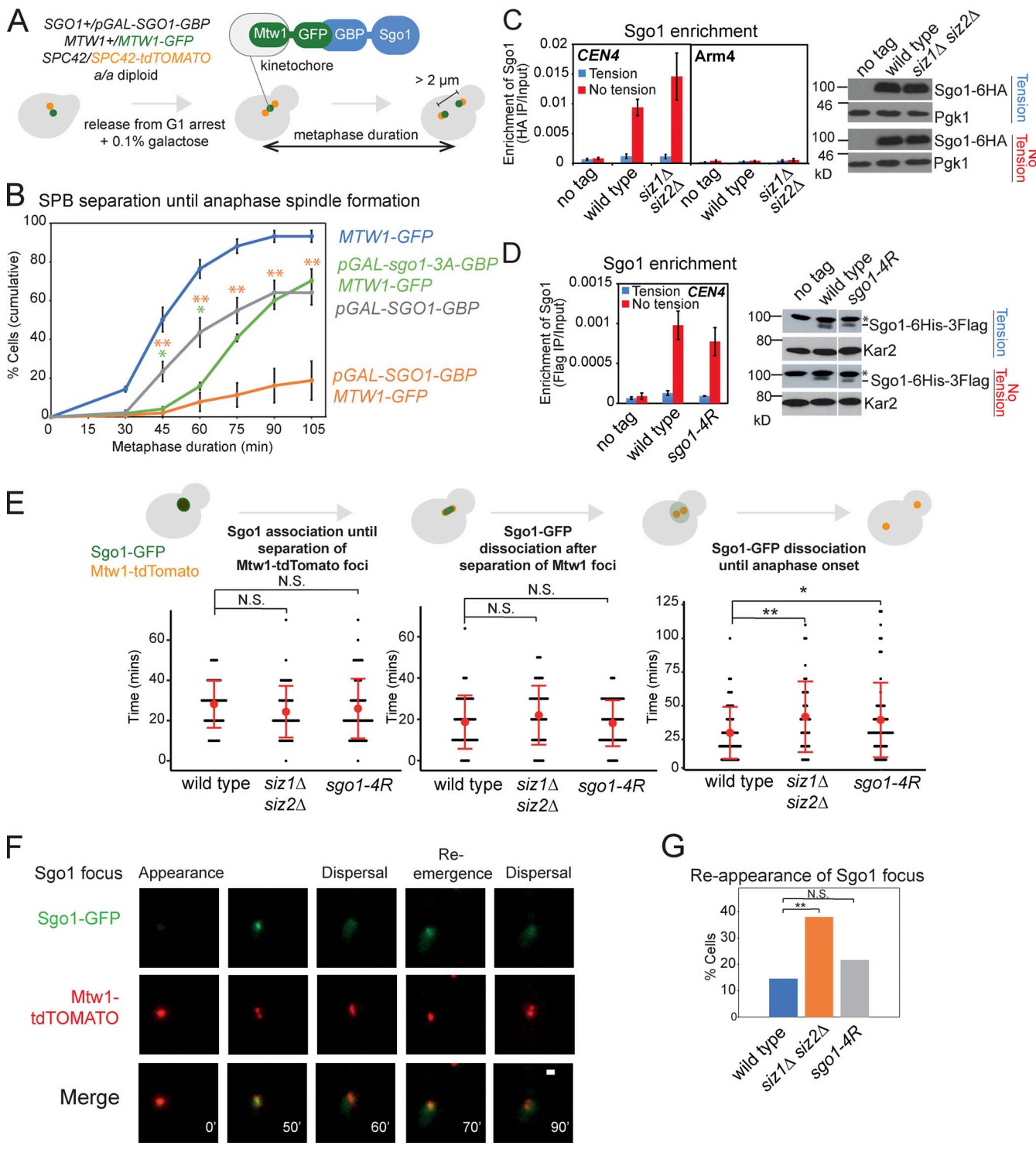

Figure 5. **Sgo1 is released under tension in the absence of SUMOylation but reassociation is increased in *siz1Δ siz2Δ*. (A and B)** Tethering Sgo1 to the kinetochore component Mtw1 delays anaphase onset. **(A)** Scheme of the experiment. Live-cell imaging was performed in an a/a diploid synchronized by release from G1. Low overexpression of *pGAL-SGO1-GBP* was induced with 0.1% galactose. Metaphase duration was estimated as the time between the emergence of two Spc42-tdTOMATO dots until they separated >2 µm apart. **(B)** Metaphase duration was measured in the following strains: *MTW1-GFP* (AMy26682), *pGAL-SGO1-GBP* (AMy26679), *pGAL-SGO1-GBP MTW1-GFP* (AMy26568), and *pGAL-sgo1-3A-GBP MTW1-GFP* (AMy26570) cells carrying *SPC42-tdTOMATO*. Shown are the average values of three to four independent experiments, with error bars representing standard errors. Statistics: two-tailed Student's *t* test comparing to the *pGAL-SGO1-GBP* strain (*, P < 0.05; **, P < 0.01). **(C)** SUMOylation is not required for bulk Sgo1 dissociation from the pericentromeres under tension. No tag control (AMy2508), *SGO1-6HA* (AMy6390), and *siz1Δ siz2Δ SGO1-6HA* (AMy8115) cells carrying *pMET-CDC20* were arrested in metaphase in the presence (DMSO) or absence (benomyl/nocodazole) of spindle tension, and Sgo1 association with the indicated site was measured by ChIP-qPCR. Shown are the average values of three independent experiments, with error bars representing standard errors. Metaphase-arrested *siz1Δ siz2Δ* cells showed a level of Sgo1 similar to wild-type cells. Protein extracts were analyzed by anti-HA or loading control (Pgk1) immunoblots. **(D)** Sgo1-4R shows similar localization to wild-type Sgo1. Sgo1

association with *CEN4* was measured by ChIP-qPCR using *SGO1-6HIS-3FLAG* (AMy25141) and *sgo1-4R-6HIS-3FLAG* (AMy26696) strains carrying *pMET-CDC20*. The experiment was performed as described in C. Metaphase-arrested *sgo1-4R* cells showed a level of Sgo1 similar to wild-type cells. Protein extracts were analyzed by anti-Flag or loading control (Kar2) immunoblots. The nonspecific bands are marked with an asterisk. **(E–G)** Evaluating Sgo1 localization and dynamics in the SUMO mutants. Wild-type (AMy9233), *siz1Δ siz2Δ* (AMy15604), and *sgo1-4R* (AMy23811) strains carrying *pMET-CDC20*, *SGO1-yeGFP*, and *MTW1-tdTOMATO* were grown in methionine dropout medium, followed by live-cell imaging. At least 50 cells from two independent experiments were analyzed for each strain. **(E)** *siz1Δ siz2Δ* and *sgo1-4R* delay in metaphase after the initial bulk Sgo1 removal from the pericentromere. Anaphase onset was estimated as the time when the two Mtw1-tdTOMATO dots were >2 µm apart. Error bars represent standard deviation, with the red dots indicating mean values. Statistics: two-tailed Student's *t* test (*, P < 0.05; **, P < 0.01; N.S., not significant). **(F)** Representative image of reemerged Sgo1 focus after its initial dispersal. Images were acquired as described in Materials and methods and were maximum-intensity projected for Sgo1-GFP and Mtw1-tdTOMATO. Scale bar = 1 µm. **(G)** The occurrence of Sgo1 focus reemergence increased in *siz1Δ siz2Δ*. Statistics: Fisher's exact test (**, P < 0.01).

2019) and through an Sgo1-dependent pathway, to which PP2A-Rts1 also contributes (Fig. 8 B; Verzijlbergen et al., 2014; Peplowska et al., 2014).

Interestingly, the coiled-coil domain of Sgo1 is required for both PP2A-Rts1 binding (Verzijlbergen et al., 2014; Xu et al., 2009) and maximum Sgo1 SUMOylation. A cocrystal structure of human Sgo1 coiled coil with PP2A-Rts1 found an extensive interaction surface between the two binding partners (Xu et al., 2009). Therefore, the attachment of SUMO (as predicted in Fig. S3 A) on the coiled coil, or the binding of the SUMO machinery to this region, may impact Sgo1–PP2A–Rts1 interaction.

To test this hypothesis, we first examined the effect of Rts1 binding on Sgo1 SUMOylation. Sgo1-3A, which does not bind Rts1, is highly SUMOylated (Fig. S3 B), suggesting that Sgo1 SUMOylation occurs independently of Rts1 binding and is possibly enhanced in its absence. We next used our robust in vitro Sgo1 SUMOylation system to examine the impact of Sgo1 SUMOylation on Rts1 binding. First, we subjected purified V5-tagged Sgo1 on beads to in vitro SUMOylation (alongside a –ATP unmodified control). Beads were then washed extensively to remove SUMO enzymes before incubating with yeast extracts containing Myc-tagged Rts1. While unSUMOylated Sgo1 bound Rts1-9Myc robustly, the presence of in vitro SUMOylated Sgo1 markedly reduced the amount of coimmunoprecipitated Rts1 (Fig. 9 A). Reciprocally, Rts1-9Myc coupled to beads coimmunoprecipitated unSUMOylated Sgo1, but not SUMOylated Sgo1 (Fig. 9 B). Hence, SUMOylation disrupts Sgo1–Rts1 interaction in vitro.

We subsequently analyzed global Sgo1–Rts1 in vivo interaction by two methods: immunoprecipitation (IP) of Sgo1-6His-3Flag followed by label-free quantitative MS (IP-MS; Table S5) and coimmunoprecipitating Rts1-9MYC using TAP-tagged Sgo1 (coimmunoprecipitation [co-IP]). Rts1 binding was abolished in the *sgo1-3A* mutant as reported (Xu et al., 2009; Fig. S3, C and D). In contrast, Rts1 binding was increased in *siz1Δ siz2Δ* (Fig. S3, C and D). Consistently, overexpressing *SIZ2* reduced Sgo1–Rts1 interaction in the co-IP (Fig. S3 D), recapitulating the in vitro analysis (Fig. 9, A and B). The interaction studies with Sgo1-4R were less clear: while IP-MS showed that it partially reduced binding to Rts1 and condensin (Fig. S3 C), Rts1 binding in the co-IP was unaffected (Fig. S3 D). The cause of this discrepancy is unclear, but one possibility is that Sgo1-4R structure is modestly impaired by the tag and growth temperature used in IP-MS. Nevertheless, and in sharp contrast to loss-of-function (*sgo1Δ* or *sgo1-3A*) mutations, untagged Sgo1-4R was proficient in biorientation, error correction, and chromosome segregation (Fig. 6),

suggesting that it retains sufficient binding to its effectors to perform these functions.

IP cannot distinguish between chromatin-bound and freely diffusible Sgo1, and Rts1 is capable of interacting with Sgo1 that is not bound to chromatin (Verzijlbergen et al., 2014). We therefore used live imaging to examine the kinetochore-localized Rts1 pool. The intensity of Rts1-GFP colocalizing with Mtw1-tdTOMATO was significantly increased in both *siz1Δ siz2Δ* and untagged *sgo1-4R* just before kinetochore separation (corresponding to early metaphase, when Sgo1 SUMOylation is maximal; Fig. 9, C and D), but was mildly increased only in *siz1Δ siz2Δ* in G1 (Fig. 9 D). Therefore, SUMOylation is a negative regulator of Sgo1-dependent recruitment of Rts1 to centromeres/kinetochores in vivo.

Previous work has indicated that forced Rts1 association with kinetochores can substitute for Sgo1 in achieving biorientation (Eshleman and Morgan, 2014; Peplowska et al., 2014). If the role of SUMOylation is to disrupt the Sgo1–Rts1 interaction, forced interaction between the two proteins is expected to ablate such a mode of regulation, thus mimicking the absence of SUMOylation. Consistently, artificial tethering of Sgo1-GBP to Rts1 fused to nonfluorescent GFP (Rts1-nfGFP) increased the switching of *CEN4*-mNeonGreen (*CEN4-mNG*) foci (Fig. 9 F), similar to *sgo1-4R* and *siz1Δ siz2Δ* mutants (Fig. 6 F). Neither initial biorientation nor chromosome segregation was affected in the tethering strain (Fig. 9, E and G). We also tethered Sgo1-4R-GBP to Rts1-nfGFP and found that switching levels were comparable to tethering of wild-type Sgo1 (Fig. 9 F), consistent with the idea that SUMOylation antagonizes the Sgo1–Rts1 interaction to stabilize biorientation. We noted a slight increase in chromosome missegregation in the *sgo1-4R-GBP* tethering strain, potentially caused by reduced levels of Sgo1-4R-GBP in *RTS1-nfGFP* cells (Fig. S3 E), for reasons that are unclear. Tethering Sgo1 or Sgo1-4R to Rts1 delayed metaphase mildly but significantly (Fig. 9 H). Together, these results demonstrate that SUMO-mediated disruption of the Sgo1–Rts1 interaction is important for stabilizing biorientation and timely anaphase onset.

## Sgo1 and Bir1 SUMOylation cooperate to promote anaphase onset

We find that *siz1Δ siz2Δ* and *sgo1-4R* share common phenotypes, including futile cycles of microtubule–kinetochore attachment and persistence of kinetochore-localized Ipl1 and Rts1. However, compared with *sgo1-4R*, *siz1Δ siz2Δ* exhibited a stronger metaphase delay, stabilized Sgo1, and possibly as a result, increased

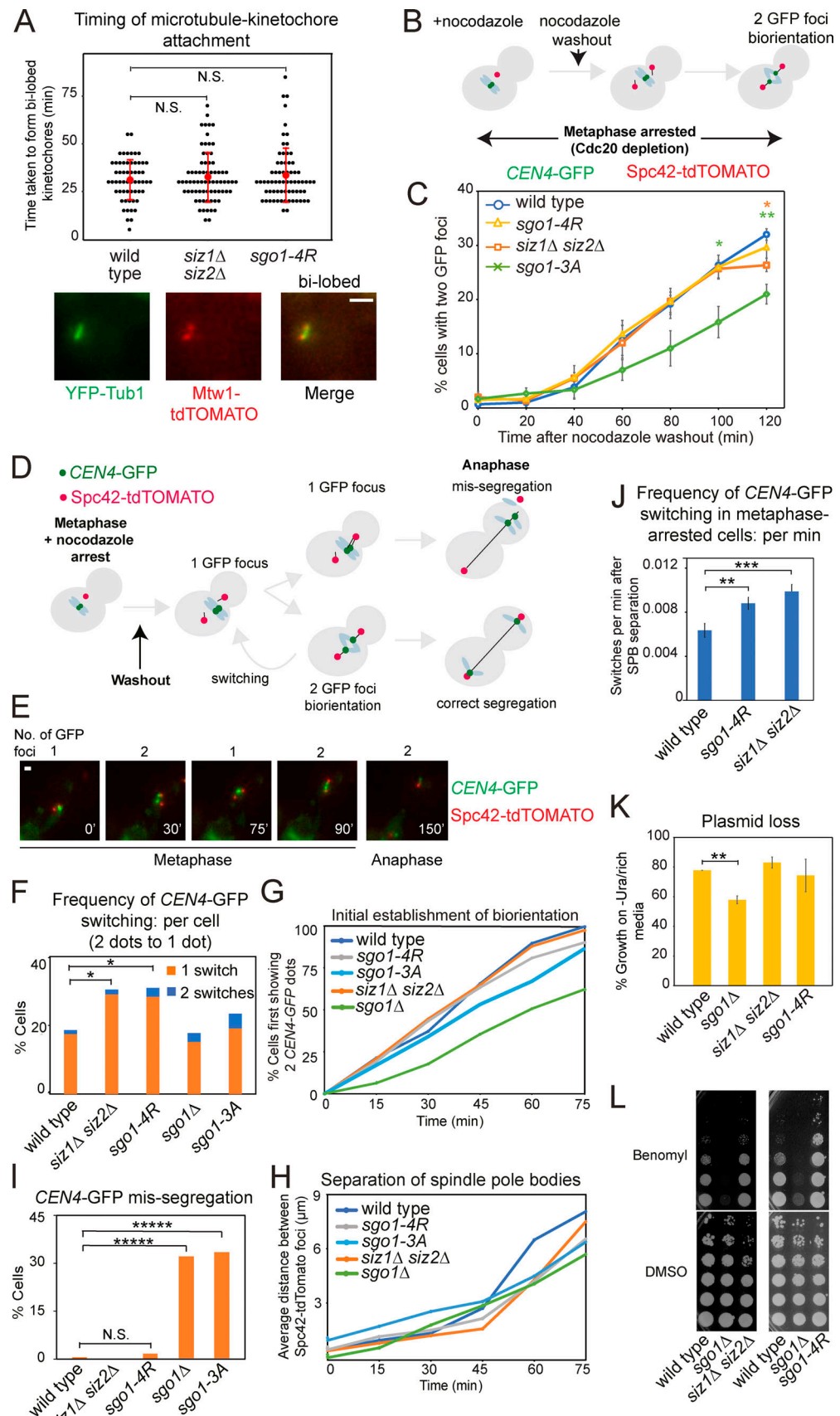

Figure 6. **SUMOylation of Sgo1 facilitates stable microtubule–kinetochore attachments. (A)** Siz1/Siz2-mediated SUMOylation does not impair the initial microtubule–kinetochore attachment. Strains used carried *pMET-CDC20 YFP-TUB1* and *MTW1-tdTOMATO*, and were wild type (AMy29568), *siz1Δ siz2Δ*

(AMy29784), and *sgo1-4R* (AMy29782). Cells were arrested in metaphase by Cdc20 depletion and nocodazole treatment. After nocodazole washout, cells were imaged every 5 min while maintaining Cdc20 depletion. Images were acquired as described in Materials and methods and were maximum-intensity projected for Mtw1-tdTOMATO and YFP-Tub1. Scale bar = 1 μm. The timing of bilobed kinetochore formation (as shown in the representative image) was measured. At least 50 cells from two independent experiments were analyzed for each strain. Error bars represent standard deviation, with the red dots indicating mean values. Statistics: two-tailed Student's *t* test. N.S., not significant. **(B and C)** SUMO-deficient Sgo1 is proficient for sister kinetochore biorientation. **(B)** Experimental scheme for evaluating sister kinetochore biorientation after nocodazole washout. Briefly, cells were arrested in metaphase by Cdc20 depletion in the presence of nocodazole. After drug washout, cells remained arrested in metaphase and were fixed and visualized as described in Materials and methods. **(C)** *sgo1-4R* and *siz1Δ siz2Δ* cells are proficient in the initial establishment of sister kinetochore biorientation. Strains used carried *pMET-CDC20*, *CEN4-GFP*, and *SPC42-tdTOMATO* and were wild type (AMy4643), *sgo1-3A* (AMy8923), *siz1Δ siz2Δ* (AMy27803), and *sgo1-4R* (AMy26278). Typically, 200 cells were analyzed for each strain in each experiment. Shown are the average values of three independent experiments, with error bars representing standard error. Statistics: two-tailed Student's *t* test comparing to the wild-type strain (*, P < 0.05; **, P < 0.01). **(D–I)** Unstable kinetochore–microtubule attachments in SUMOylation mutants. Strains used in C, together with *sgo1Δ* (AMy6117), were monitored by live imaging on a microfluidics device. **(D)** Scheme describing biorientation assay after nocodazole washout. Briefly, nocodazole was washed out from metaphase-arrested cells, and state of *CEN4-GFP* was tracked by live-cell imaging. **(E)** Representative images of reassociation of two *CEN4-GFP* dots. Images were acquired as described in Materials and methods and were maximum-intensity projected for *CEN4*-GFP and Spc42-tdTOMATO. Scale bar = 1 μm. **(F)** Biorientation is unstable in *siz1Δ siz2Δ* and *sgo1-4R* cells. At least 50 cells from three independent experiments were analyzed for each strain. Shown are the percentages of cells with the indicated number of switchings from two *CEN4*-GFP dots back to one dot. Statistics: Fisher's exact test (*, P < 0.05). **(G)** The initial establishment of biorientation is unaffected in *siz1Δ siz2Δ* and *sgo1-4R* cells. The time point at which a cell first displayed two *CEN*4-GFP dots was defined as the timing of the initial establishment of biorientation. **(H)** The distance between Spc42-tdTOMATO dots was measured by ImageJ, and the average distance was calculated for each time point. **(I)** *siz1Δ siz2Δ* and *sgo1-4R* cells do not show increased missegregation of chromosomes. The inheritance of *CEN4-GFP* by the daughter cells was assessed. Statistics: Fisher's exact test (*****, P < 0.00001). **(J)** Switching frequency was increased in cells arrested in metaphase. Wild-type, *sgo1-4R*, and *siz1Δ siz2Δ* strains used in Fig. 6, D–I, were released from α-factor into methionine-containing medium (metaphase arrest). Cells were imaged every 15 min, and switching frequency was calculated as number of switches observed per minute cells spent after spindle pole body (SPB) separation. At least 100 cells from two independent experiments were evaluated for each strain. Error bars represent standard error. Statistics: two-tailed Student's *t* test comparing to the wild-type strain (**, P < 0.01; ***, P < 0.001). **(K)** *siz1Δ siz2Δ* and *sgo1-4R* cells retain a *CEN*-containing plasmid to a similar extent as the wild-type cells. Wild type (AMy1176), *sgo1Δ* (AMy827), *siz1Δ siz2Δ* (AMy7625), and *sgo1-4R* (AMy21705) were transformed with a *CEN*-containing plasmid pRS316. Plasmid loss was evaluated in three independent transformants of each strain as described in Materials and methods. Error bars represent standard error. Statistics: one-tailed Student's *t* test comparing to the wild-type strain (**, P < 0.01). **(L)** *siz1Δ siz2Δ* and *sgo1-4R* cells grew similarly as wild type or showed mildly improved growth on benomyl. Serially diluted cells of wild type (AMy1176), *sgo1Δ* (AMy827), *siz1Δ siz2Δ* (AMy7625), and *sgo1-4R* (AMy21705) were plated on medium containing 10 μg/ml benomyl or DMSO (solvent).

Sgo1 reassociation with chromatin during the metaphase–anaphase transition.

Because the metaphase delay of *siz1Δ siz2Δ* largely depends on Sgo1 (Fig. 1, D and G), the moderate effect of *sgo1-4R* on metaphase timing could be because Sgo1 degradation contributes to its inactivation, or because of the existence of other SUMOylation targets, on either Sgo1 itself or its effectors. To discern these possibilities, we generated the *sgo1-4R-Δdb* mutant, which reduced SUMOylation compared with *sgo1-Δdb* (Fig. S4 A) but did not delay metaphase further than *sgo1-4R* (Fig. S4 B), indicating that Siz1/Siz2 must have other targets. To identify other relevant targets, we analyzed SUMOylation of Sgo1 effectors and regulators in metaphase-arrested cells in either the presence or absence of tension and found that Ipl1, Bir1, Ycs4, and Brn1 were SUMOylated, although only Bir1 SUMOylation was affected by tension (Fig. S4 C). SUMOylation on Rts1 or the Mps1 kinase was not observed in our experimental conditions (Fig. S4 C), despite the latter being SUMOylated in human cells (Restuccia et al., 2016).

Bir1 plays a crucial role in CPC recruitment to centromeres, through both Sgo1- and Ndc10-dependent pathways (Cho and Harrison, 2011; Yoon and Carbon, 1999; Kawashima et al., 2010). Bir1 SUMOylation is dependent on Siz1/Siz2, and Bir1 is stabilized in *siz1Δ siz2Δ* cells (Fig. 10 A), where slower-migrating forms, likely phosphorylated, accumulate (Fig. 10 B).

To understand whether Bir1 SUMOylation might regulate the metaphase–anaphase transition, we mutated three lysine residues (Lys707, 732, and 785) that were reported to be SUMOylated in a proteome-wide study (Esteras et al., 2017) to generate *bir1-3R*, which drastically reduced Bir1 SUMOylation (Fig. 10 A).

Like *siz1Δ siz2Δ* and *sgo1-4R*, *bir1-3R* was insensitive to benomyl, suggesting that CPC function is intact (Fig. S4 D). *bir1-3R* exhibited a mild metaphase delay, which was not exacerbated by *sgo1-4R* or *sgo1-Δdb* (Fig. 10, C and D). Strikingly, the *sgo1-4R-Δdb bir1-3R* triple mutant showed a strong metaphase delay without causing benomyl sensitivity, similar to *siz1Δ siz2Δ* (Figs. 10 D and S4 D). Hence, the requirement of Siz1/Siz2 for timely anaphase can be explained by the combinatorial effects of direct Sgo1 and Bir1 SUMOylation, together with indirect effects on Sgo1 turnover.

## Discussion

### Identification of shugoshin regulators

Starting with an unbiased genetic screen, we have identified SUMO ligases as negative regulators of the pericentromeric hub that responds to a lack of tension between kinetochores. We identify Sgo1, the central pericentromeric adaptor protein, as a key new target of the Siz1/Siz2 SUMO ligases. Kinetochore-microtubule interactions are unstable in Sgo1 SUMO–deficient cells (*siz1Δ siz2Δ* and *sgo1-4R*). Persistent cycles of kinetochore detachment and reattachment engage the SAC, explaining why a failure to SUMOylate Sgo1 results in a metaphase delay. Consistently, inactivation of components of the SAC or error correction pathways (Mad2 or Bir1) advance anaphase timing in *siz1Δ siz2Δ* cells.

Sgo1 inactivation at kinetochores is essential for timely anaphase entry (Clift et al., 2009; Fig. 5 B), and here we have identified one mechanism that contributes to this inactivation. Sgo1 also prevents anaphase onset by inhibiting separase

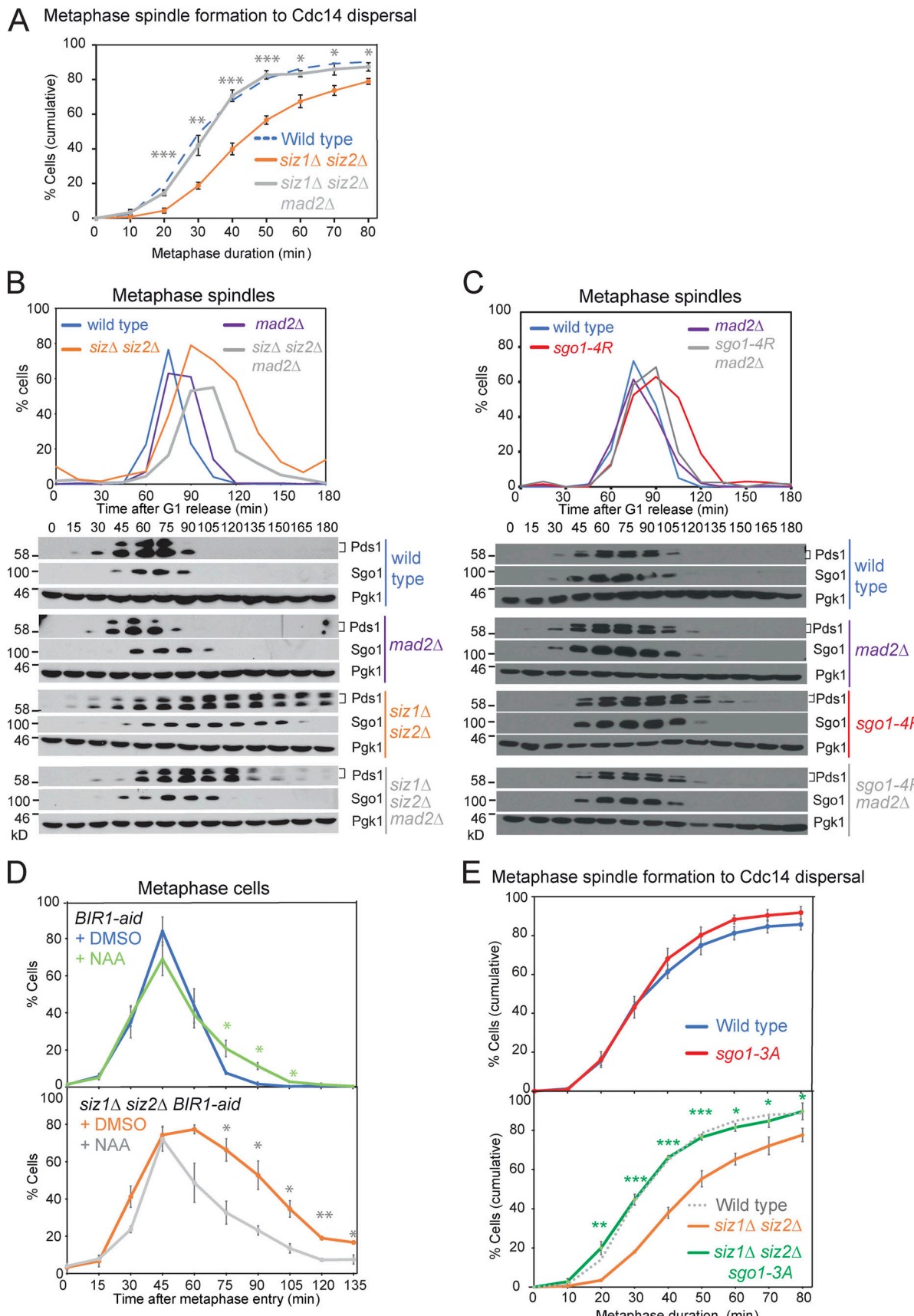

Figure 7. **Siz1/Siz2 and Sgo1 SUMOylation promote the metaphase–anaphase transition in a SAC- and CPC-dependent manner. (A and B)** Deletion of the SAC component, *MAD2,* rescued the metaphase delay of *siz1Δ siz2Δ* mutant. **(A)** Metaphase duration was determined by live imaging for the following

strains carrying *YFP-TUB1* and *CDC14-GFP*: wild type (AMy24174), *siz1Δ siz2Δ mad2Δ* (AMy29479), and *siz1Δ siz2Δ* (AMy24313). Shown are the average values of three independent experiments, with error bars representing standard error. Statistics: two-tailed Student's *t* test comparing to the *siz1Δ siz2Δ* strain (*, $P < 0.05$; **, $P < 0.01$; ***, $P < 0.001$). **(B)** A mitotic time course was performed including the following strains: wild type (AMy8467), *mad2Δ* (AMy9635), *siz1Δ siz2Δ* (AMy8452), and *siz1Δ siz2Δ mad2Δ* (AMy9634). **(C)** Deletion of *MAD2* rescued the metaphase delay of *sgo1-4R* mutant. Mitotic time course was performed for the following strains: wild type (AMy18500), *mad2Δ* (AMy22555), *sgo1-4R* (AMy22165), and *sgo1-4R mad2Δ* (AMy22739). **(D)** Conditional degradation of Bir1 partially rescued the metaphase delay in *siz1Δ siz2Δ BIR1-aid*. *BIR1-aid* (AMy29537) and *BIR1-aid siz1Δ siz2Δ* (AMy29536) strains carried *YFP-TUB1* and *CDC14-GFP*. 0.5 mM naphthalene-1-acetic acid (NAA) was added when >80% cells showed small buds, and 0.25 mM NAA was re-added every hour. Cells were fixed and analyzed as described in Fig. 1 G. Metaphase cells were identified based on spindle morphology and sequestered Cdc14. 100–200 cells were analyzed for each time point, and shown are the average results of three independent experiments, with the error bars representing standard error. Statistics: one-tailed Student's *t* test (*, $P < 0.05$; **, $P < 0.01$). **(E)** The *sgo1-3A* mutant rescued the metaphase delay in *siz1Δ siz2Δ*. Metaphase duration was determined by live-cell imaging as described in Fig. 1 F. Strains used carried *TUB1-YFP* and *CDC14-GFP* and were wild type (AMy24174), *siz1Δ siz2Δ* (AMy24313), *sgo1-3A* (AMy24433), and *siz1Δ siz2Δ sgo1-3A* (AMy24471). Shown are the average values of three to four independent experiments, with error bars representing standard error. Statistics: two-tailed Student's *t* test comparing to the wild-type strain (*, $P < 0.05$; **, $P < 0.01$; ***, $P < 0.001$).

independently of securin (Clift et al., 2009). PP2A-Cdc55–dependent dephosphorylation of separase and, potentially, also cohesin itself are likely to underlie this inhibition (Lianga et al., 2018; Yaakov et al., 2012). Notably, *ZDS2*, a negative regulator of PP2A-Cdc55 (Queralt and Uhlmann, 2008), was also isolated in our screen along with *HOS1*, the cohesin deacetylase (Xiong et al., 2010; Beckouët et al., 2010; Borges et al., 2010; Table S1), indicating that further mechanisms await discovery.

## SUMOylation of Sgo1 and Bir1 ensure efficient entry into anaphase

How does SUMOylation regulate anaphase entry? We found that Sgo1 SUMOylation is dispensable for its tension-dependent release from pericentromeres, and that, although SUMOylation promotes Sgo1 degradation indirectly, this is not sufficient for efficient anaphase entry. Instead, our work suggested that Sgo1 SUMOylation likely promotes anaphase entry by dampening the error-correction machinery as microtubules establish stable interactions with microtubules in prometaphase (Fig. 10 E). Consistent with this interpretation, sister kinetochore biorientation was established on time, but was highly unstable in Sgo1 SUMO–deficient mutants (Fig. 6 F). Furthermore, Sgo1 showed increased reassociation with kinetochores in *siz1Δ siz2Δ* mutants as tension was established at metaphase (Fig. 5, F and G). Remarkably, we found that Ipl1 removal from kinetochores was incomplete in the Sgo1 SUMO mutants (Fig. 8, A and B), suggesting a key role of this modification in modulating the subcellular localization of Ipl1. Interestingly, the CPC component and Ipl1 regulator, Bir1, is also SUMOylated (Fig. 10 A; Montpetit et al., 2006) and we found that Bir1-SUMO, like Sgo1 SUMO, is important for timely anaphase onset. Meanwhile, we showed that Rts1 binds preferentially to unSUMOylated Sgo1, and that tethering Sgo1 to Rts1 destabilized biorientation in a way similar to the Sgo1 SUMO mutants (Fig. 9).

Our work suggests a model (Fig. 10 E) wherein, in prometaphase, Sgo1 associates with pericentromeric chromatin dynamically and recruits PP2A-Rts1 to promote CPC maintenance. We envisage that a fraction of pericentromere-associated Sgo1 and Bir1 are SUMOylated, which both reduces PP2A-Rts1 binding and diminishes CPC levels at kinetochores. This dampening of error-correction activities increases the lifetime of microtubule attachment, providing an opportunity for bipolar attachments to form. The resultant tension triggers the release of Sgo1

and its effectors from the pericentromere, stabilizing the bioriented state and sharpening the transition into anaphase. SUMOylation also likely acts indirectly to reduce the pool of Sgo1 and Bir1 available for persistence at, or reassociation with, pericentromeres. Upon commitment to anaphase, Sgo1 is irreversibly degraded in an APC/C-dependent manner.

Several questions need to be addressed to understand the underlying mechanisms. Key among them is how does Sgo1/PP2A and SUMOylation regulate CPC recruitment and removal? While our MS analysis showed robust binding between Sgo1 and PP2A-Rts1, we did not identify any coprecipitating CPC components (Table S5). The association between Sgo1 and CPC is therefore likely to be transient, or indirect. Interestingly, in cells lacking Rts1, CPC subunits show decreased phosphorylation, while Ipl1 substrates show increased phosphorylation, supporting the idea that PP2A-Rts1 regulates CPC (Zapata et al., 2014; Touati et al., 2018). Furthermore, CPC relocalization to the spindle midzone was found to be mediated by the Cdc14 phosphatase (Mirchenko and Uhlmann, 2010), raising the possibility that a complex PP2A-Rts1/Cdc14 regulatory network exists to control the levels of CPC around the kinetochores. In fission yeast, Sgo2 delays anaphase onset by preventing relocalization of CPC to the spindle midzone by the Klp9 kinesin (Meadows et al., 2017), and Sgo2 is also SUMOylated (Fig. 2 C) on its coiled-coil domain in a proteomics study (Nie et al., 2015). Therefore, error-correction silencing and CPC localization appear to be governed by an interplay between shugoshins, kinases, phosphatases, and kinesins. How Sgo1 and Bir1 SUMOylation participate in this complex regulatory network awaits future studies.

## SUMOylation: A generalized mechanism of centromere regulation with implications for disease?

Accumulating evidence indicates that SUMOylation might play a specific role at centromeres to fine-tune chromosome segregation. The SUMO isopeptidase Ulp2/Smt4 is important for maintenance of cohesion specifically at centromeres, in part through regulating Topoisomerase II (Stephens et al., 2015). The Pds5 subunit of cohesin is also known to prevent poly-SUMOylation of cohesin (D'Ambrosio and Lavoie, 2014), and centromeric cohesin may be particularly susceptible since it lacks Pds5 (Petela et al., 2017). Protein inhibitor of activated STAT SUMO ligases are known to be localized at centromeres in vertebrate mitotic cells and oocytes (Azuma et al., 2005; Ban

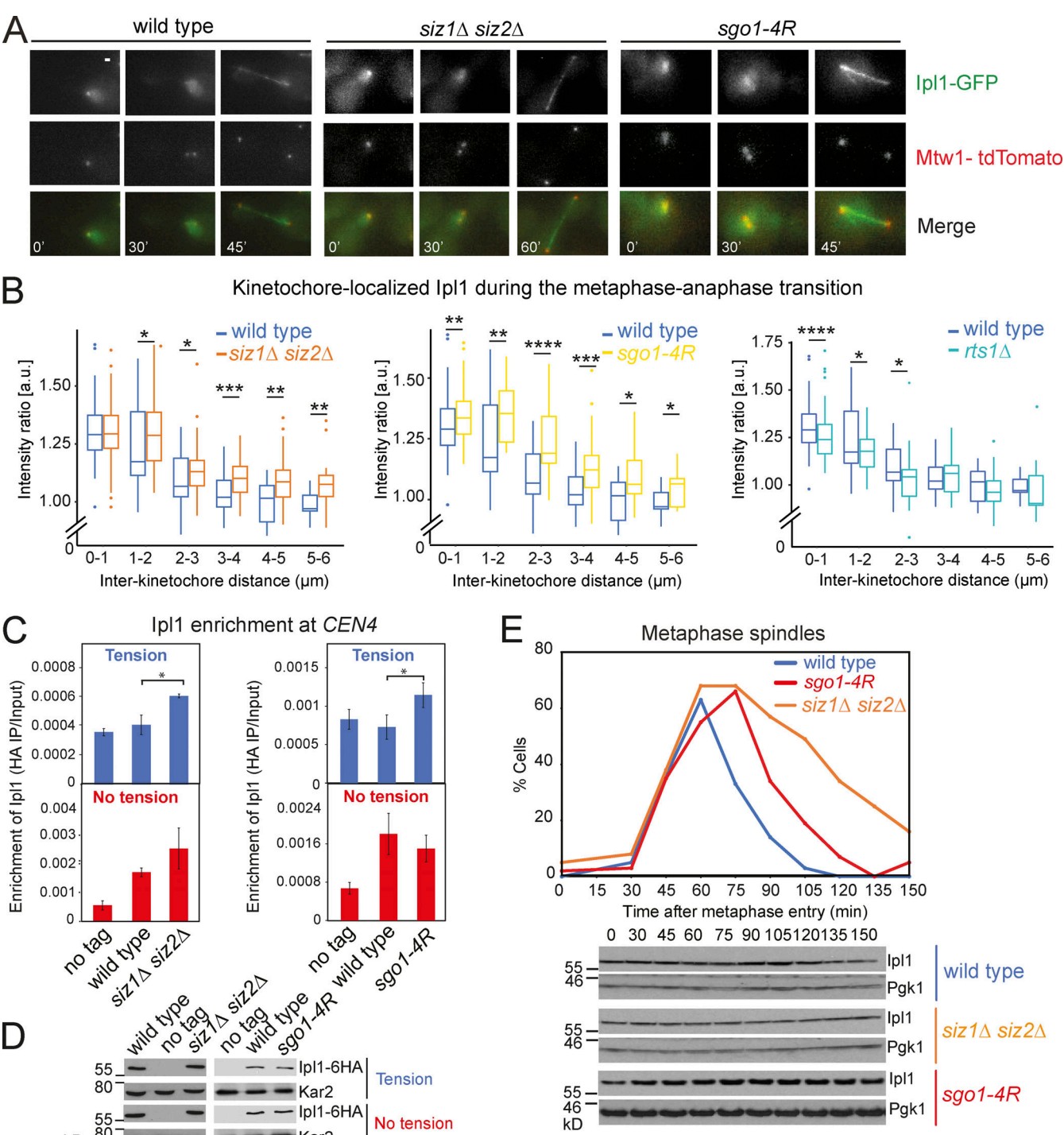

Figure 8. **Ipl1 stays longer on kinetochores during the metaphase–anaphase transition in Sgo1 SUMO-deficient mutants. (A and B)** Strains used contained *IPL1-yeGFP*, *MET-CDC20*, and *MTW1-tdTOMATO*, and were wild type (AMy9231), *siz1Δ siz2Δ* (AMy15602), *sgo1-4R* (AMy24143), and *rts1Δ* (AMy13180). Cells were released from G1 into medium lacking methionine and were imaged on a microfluidics device. **(A)** Representative images. Images were acquired as described in Materials and methods and were maximum-intensity projected for Ipl1-GFP and Mtw1-tdTOMATO. Scale bar = 1 µm. **(B)** Ipl1 removal from kinetochores is incomplete in SUMO-deficient mutants, in contrast to *rts1Δ*, in which Ipl1 loading was reduced. Line scans were performed across kinetochore foci of single cells, which measured the distance between the two Mtw1-tdTOMATO foci, as well as the Ipl1-GFP intensities colocalizing with the Mtw1 foci (https://doi.org/10.5281/zenodo.3442325). 200–300 line scans compiled from two independent experiments were performed for each strain. Intensity ratio = average peak intensity of the two Ipl1-GFP signals/average peak intensity of the two Mtw1-tdTOMATO signals. Box plots were generated for each distance interval. Statistics: one-tailed Student's *t* test (*, P < 0.05; **, P < 0.01; ***, P < 0.001). **(C and D)** In Sgo1 SUMO-deficient cells, Ipl1 is not completely removed when the cells are under tension. **(C)** Ipl1 association with *CEN4* was measured by ChIP-qPCR using wild type (AMy26686), *siz1Δ siz2Δ* (AMy23194), and *sgo1-4R* (AMy26692) carrying *IPL1-6HA*, together with a no-tag control (AMy2508). Cells were arrested in metaphase by depletion of Cdc20 in the presence or absence of spindle tension. Error bars represent standard error calculated from three to five independent experiments. Statistics: one-tailed Student's *t* test (*, P < 0.05). **(D)** Ipl1 protein levels are unchanged in Sgo1 SUMO-deficient mutants. Protein extracts from C were analyzed by anti-HA and anti-Kar2 (loading control)

Western blotting. **(E)** Ipl1 levels are unchanged during the metaphase–anaphase transition. Strains used contained Ipl1-6HA and were wild type (AM8975), *siz1Δ siz2Δ* (AM21934), and *sgo1-4R* (AM22182). Cells were released from G1 arrest, and spindles and protein extracts were analyzed as described in Materials and methods.

et al., 2011; Ding et al., 2018). Moreover, the SUMO pathway is required to prevent cohesion loss during meiosis II, which centrally requires Sgo2-PP2A, and it is conceivable that modulation of the PP2A–Sgo1 interaction as we find in yeast underlies these observations in mouse oocytes (Ding et al., 2018). Indeed, global studies found that shugoshins are SUMOylated in fission yeast and human cells, though the function has yet to be examined (Schimmel et al., 2014; Nie et al., 2015). Furthermore, there is ample evidence from other organisms that CPC function is subject to regulation by SUMOylation. Aurora B SUMOylation in *Xenopus* and human cells was shown to promote its enrichment at centromeres, and it was proposed that this modification may serve as a reversible mechanism to dampen Aurora B kinase activity (Fernández-Miranda et al., 2010), while in *Caenorhabditis elegans* meiosis, the localizations of both Aurora B and Bub1 kinase are influenced by SUMOylation (Pelisch et al., 2019, 2017; Davis-Roca et al., 2018). This suggests that SUMO may have a general role in CPC/error-correction pathways, and we speculate that multilateral SUMO–SUMO-interacting motif interactions (Jentsch and Psakhye, 2013) enable the coordinated relocalization of surveillance factors. Notably, misregulation of the SUMO pathway is widespread in different cancers (Rabellino et al., 2017; Seeler and Dejean, 2017). Potentially, reductions in chromosome segregation fidelity caused by SUMO malfunction at centromeres, as we show here, could be a contributing factor in driving these malignant states.

## Materials and methods
### Yeast strains and plasmids
Yeast strains are derivatives of W303 and are listed in Table S2. Plasmids and primers are listed in Table S3 and Table S4, respectively. StuI-digested AMp1239 was transformed into a *CDC14-GFP* strain to make the *YFP-TUB1 CDC14-GFP* parent strain. Genes were deleted or tagged using PCR-based transformation. *SGO1* K-R mutant plasmids were generated using Quikchange II XL site-directed mutagenesis kit (Agilent), with primers listed in Table S4. K-R mutants were PCR amplified from the resulting plasmid using primers AMo16 and AMo3177 and were integrated into an *sgo1Δ* strain (AMy827). Wild-type *BIR1* together with its endogenous promoter and terminator sequence was amplified from genomic DNA using primers AMo9476 and AMo9477 and cloned into YIplac128 by Gibson assembly (AMp1948). The *bir1-3R* plasmid (AMp1950) was generated by Gibson assembly, using a fragment containing *bir1-3R* mutations (GeneArt). PCR product of primers AMo9466 and AMo9467 on plasmid AMp1950 was transformed into a *bir1Δ/+* heterozygous diploid. *7HIS-SMT3* and *HIS-UBI* plasmids were kind gifts from Dr. Helle Ulrich (Institute of Molecular Biology gGmbH, Mainz, Germany). *RTS1-GFP* was a kind gift from Richard Hallberg (Syracuse University, Syracuse, NY; Gentry and Hallberg, 2002).

### Yeast growth and synchronization
Unless otherwise stated, yeast strains were grown in YEP supplemented with 2% glucose and 0.3 mM adenine (YPDA). To prepare cells for G1 arrest, overnight cultures were inoculated to $OD_{600}$ = 0.2 and grown for 1 h at room temperature, before rediluting to $OD_{600}$ = 0.2. Cells were arrested in 5 µg/ml α-factor for 90 min and additional 2.5 µg/ml α-factor for 90 min, until >95% cells exhibited shmooing morphology. To release cells from G1, α-factor was washed out using 10× volume YEP. For *pMET-CDC20*–containing strains, cells were arrested in G1 in methionine dropout medium. After α-factor washout, cells were released into YPDA (+ DMSO or + 30 µg/ml benomyl and 15 µg/ml nocodazole) + 8 mM methionine for 1 h. 4 mM methionine and DMSO/15 µg/ml nocodazole were re-added for a further 1 h before samples were collected. We routinely combine benomyl and nocodazole to ensure a robust arrest.

### Metaphase duration measurements by live-cell imaging and mitotic time course
Synthetic complete/dropout media were used for growing and washing cells for live-cell imaging. Cells released from G1 arrest were loaded onto µ-slide dishes (Ibidi) coated with concanavalin A (Sigma-Aldrich). All live images in this study were acquired by a 100×/1.46 objective on a Zeiss AxioObserver Z1 microscope coupled to an ORCA Flash 4.0 camera (Digital C11440; Hamamatsu), in a temperature-controlled chamber (25°C for glucose-based growth and 30°C for raffinose-based growth). Images were acquired using the Zen software, and analyses were performed using ImageJ.

Mitotic time course experiments were performed at room temperature. For every time point, samples were either treated with 5% TCA for protein extraction or fixed in 3.7% formaldehyde diluted in 0.1 M potassium phosphate buffer, pH 6.4, for immunofluorescence. TCA-treated pellets were snap frozen in liquid nitrogen, washed in acetone, and air-dried. Protein samples were prepared by bead beating and boiling in SDS sample buffer. For immunofluorescence, fixed cells were spheroplasted using zymolyase (AMS Biotechnology) and glusulase (Perkin Elmer) and fixed in methanol for 3 min, followed by 10-s incubation in acetone. Rat anti-α-tubulin antibody (#MCA77G; Abd Serotec) was used at 1:50, and preabsorbed (x6 diluted) anti-rat FITC (#712-095-153; Jackson ImmunoResearch) was used at 1:16.7. Spindle morphology of ≥50 cells was analyzed for each time point. For fixed-cell analysis of YFP-Tub1 Cdc14-GFP strains, 900 µl cells was added to 100 µl of 37% formaldehyde and incubated for 9 min, before washing once in 80% ethanol and resuspending in 50 µl PBS. Fixed cells were analyzed by an 100×/1.46 objective on a Zeiss Imager.Z1 microscope coupled to an ORCA-ER camera (Digital C4742-80; Hamamatsu).

### Western blotting
Proteins were separated in 8% bis-Tris acrylamide gels and were transferred to nitrocellulose membranes (except for

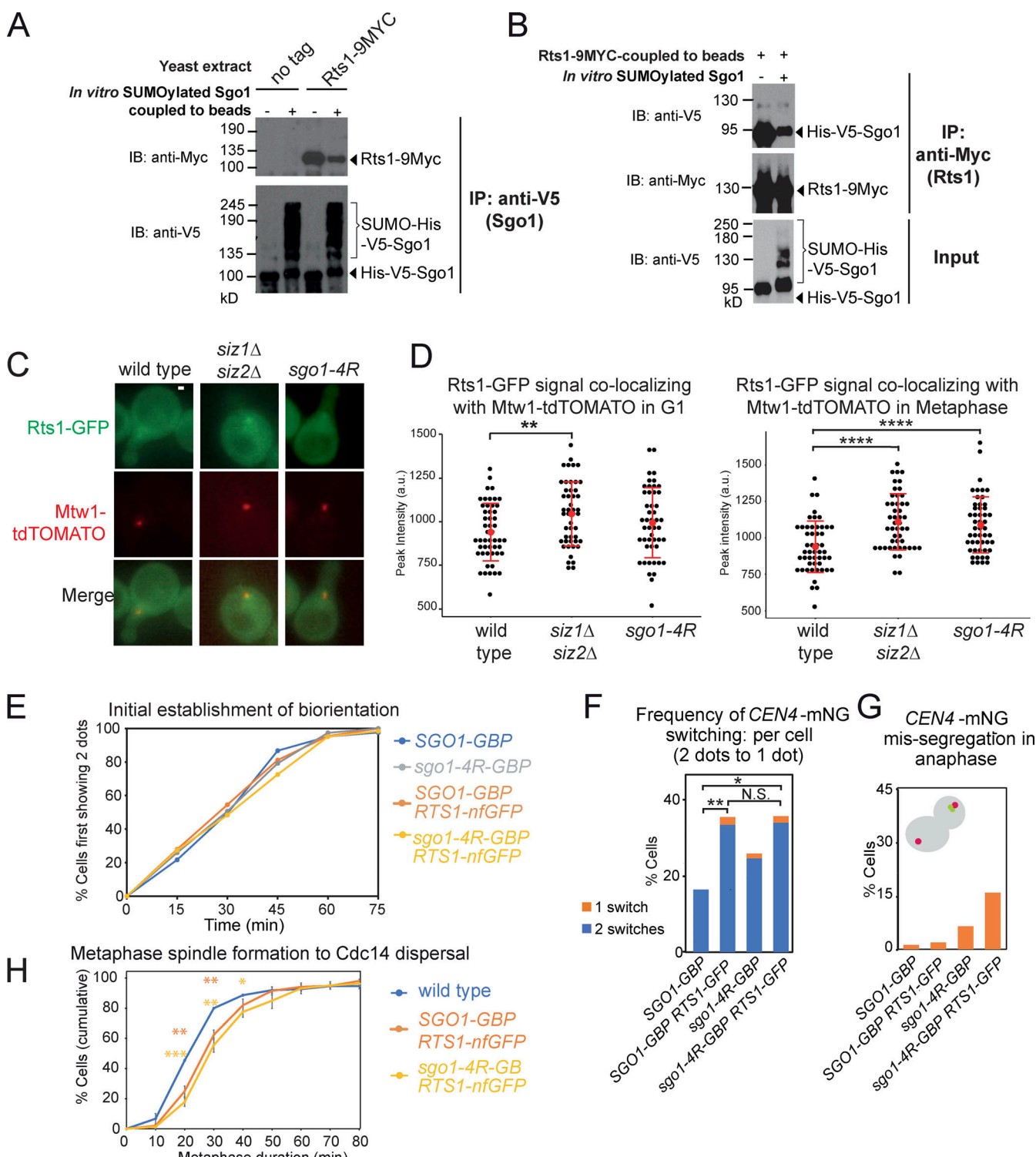

Figure 9. **SUMOylation blocks Rts1 binding to Sgo1, and release of this interaction is important for stable biorientation. (A)** SUMOylated Sgo1 has reduced binding affinity for Rts1. Recombinant V5-tagged Sgo1 was mixed with components of the SUMOylation pathway in the presence or absence of ATP. Anti-V5 antibody coupled to Protein G Dynabeads was added to the mixture, washed thoroughly, and incubated with extract from *sgo1Δ* (AM827) or *sgo1Δ RTS1-9MYC* (AMy8832) cells. Levels of Sgo1 and Rts1 bound to beads were probed by anti-V5 and anti-Myc Western blotting, respectively. IB, immunoblot. **(B)** Rts1 preferentially binds to unSUMOylated Sgo1. Rts1-9Myc was immunoprecipitated from *sgo1Δ RTS1-9MYC* (AMy8832) using anti-Myc antibody coupled to Protein G Dynabeads. Beads were incubated with in vitro SUMOylation reaction mixture containing Sgo1. Levels of Sgo1 and Rts1 bound to beads were probed by anti-V5 and anti-Myc Western blotting, respectively. **(C and D)** The kinetochore-proximal pool of Rts1 was increased in the absence of SUMOylation in metaphase. Strains used carried Rts1-GFP and Mtw1-tdTOMATO and were wild type (AMy29076), *siz1Δ siz2Δ* (AMy29074), and *sgo1-4R* (AMy29083). Measurements were made in shmooing cells (G1) or right before the emergence of two Mtw1-tdTOMATO kinetochore foci (metaphase). **(C)** Representative images of Rts1-GFP and Mtw1-tdTOMATO signals from the same z-positions. Images were acquired as described in Materials and methods. Scale bar = 1 μm.

**(D)** Quantification of Rts1-GFP peak intensity that occupied the same area as the kinetochore (Mtw1-tdTOMATO). ImageJ was used to measure the intensities. For each cell, the area occupied by the kinetochore was selected by outlining the boundaries of Mtw1-tdTOMATO signals. The same area in the same z-slice was restore selected in the Rts1-GFP channel, the peak green value was measured, and background was subtracted. 48 cells from two independent experiments were analyzed for each strain. Red dots represent mean values, and error bars represent standard deviation. Statistics: one-tailed Student's $t$ test (**, $P < 0.01$; ****, $P < 0.0001$). **(E–G)** Forced interaction between Rts1 and Sgo1 destabilizes biorientation. Strains used carry *CEN4-mNeonGreen pMET-CDC20* and *SPC42-tdTOMATO* and were *SGO1-GBP* (AMy28389), *SGO1-GBP RTS1-nonfluorescent GFP* (AMy28092), *sgo1-4R-GBP* (AMy28417), and *sgo1-4R-GBP RTS1-nonfluorescent GFP* (AMy28416). The assay was performed as described in Fig. 6 D. **(E)** Tethering Rts1 to wild-type Sgo1 or Sgo1-4R does not affect the initial establishment of biorientation. **(F)** Increased reassociation of *CEN4-mNeonGreen* dots was observed when Rts1 was tethered to wild-type Sgo1 or Sgo1-4R. At least 50 cells from two independent experiments were analyzed for each strain. Shown are the percentages of cells with the indicated number of switchings from two *CEN4-GFP* dots back to one dot. Statistics: Fisher's exact test (*, $P < 0.05$; **, $P < 0.01$; N.S., not significant). **(G)** Missegregation is modestly increased when Rts1 is tethered to wild-type Sgo1 or Sgo1-4R. **(H)** Tethering Sgo1 to Rts1 mildly delays metaphase. Strains used carried *CDC14-mRuby YFP-TUB1* and were wild type (AM29672), *SGO1-GBP-3MYC RTS1-nfGFP* (AM29711), and *sgo1-4R-GBP-3MYC RTS1-nfGFP* (AM29715). Shown are the average of three independent experiments, and error bars represent standard error. Statistics: one-tailed Student's $t$ test comparing to the wild-type strain (*, $P < 0.05$; **, $P < 0.01$; ***, $P < 0.001$).

---

ubiquitination analysis, for which PVDF membranes were used). Membranes were blocked in 3–5% milk in PBS + 0.1% Tween-20 (PBST) or TBS + 0.1% Tween-20 (TBST). All antibodies were diluted in 2% milk PBST (except for ubiquitination analysis, where 2% milk TBST was used). Anti-c-Myc (9E10, #626802; BioLegend), anti-V5 (#MCA1360; Bio-Rad AbD Serotec), anti-Flag (M2, #F1804; Sigma-Aldrich), anti-HA (12CA5, #1666606001; Roche), anti-HA (HA11, #901501; BioLegend), and peroxidase anti-peroxidase (#1291; Sigma-Aldrich) antibodies were used at 1:1,000 dilution. HRP-coupled anti-mouse or anti-rabbit secondary antibodies (#NXA931 and #NA934, respectively; GE Healthcare) were used as appropriate. Anti-Pgk1 (laboratory stock) and anti-Kar2 (laboratory stock) loading controls were used at 1:10,000 and 1:100,000 dilutions, respectively. Signals were detected by ECL (Thermo Fisher Scientific) and autoradiograms. 20% Femto-ECL (Thermo Fisher Scientific) diluted in Pico-ECL was used to detect SUMOylated-Sgo1 signals.

**Analysis of in vivo SUMOylation**
For cycling cells, cultures were inoculated to $OD_{600} = 0.2$ in 200 ml synthetic dropout medium and grown for 4 h at room temperature. Equal-OD cells were collected for samples in the same experiment. For time course experiments, cells were synchronized and arrested in metaphase as described above. 200 ml culture was collected for each time point.

Cell pellets were resuspended in 20 ml cold $H_2O$ and incubated with 3.2 ml solution containing 1.85 M sodium hydroxide and 7.5% β-mercaptoethanol. After 20-min incubation on ice, 1.65 ml of 100% trichloroacetic acid was added, and cells were incubated on ice for a further 20 min. Cell pellets were drop-frozen in liquid nitrogen and lysed by bead-beating in 300 μl buffer A (6 M guanidine hydrochloride, 100 mM sodium phosphate buffer, pH 8.0, and 10 mM Tris-HCl, pH 8.0). Lysed cells were centrifuged at 13,000 rpm for 20 min, and the supernatant was diluted with 600 μl buffer A and recentrifuged at 13,000 rpm for 10 min. 10 μl supernatant was added to buffer B (8 M urea, 100 mM sodium phosphate buffer, pH 6.8, and 10 mM Tris-HCl, pH 6.8) as input control. The remaining lysate was loaded onto a column packed with 600 μl of 50% slurry Ni-NTA agarose beads (Qiagen) and washed twice with buffer A + 0.05% Tween-20, twice with buffer B + 0.05% Tween-20, and once with buffer B + 0.05% Tween-20 + 20 mM imidazole.

SUMOylated proteins were eluted by buffer B + 0.05% Tween-20 + 200 mM imidazole. Input and elute samples were concentrated by centrifugation using Vivaspin columns (Sartorius) and boiled in SDS sample buffer.

**Cloning, expression, and purification of recombinant Sgo1**
Full-length wild-type *SGO1* was amplified from plasmid AMp899, to replace *SMT3* in plasmid AMp773 by Gibson assembly using primers AMo8849–AMo8852. V5 tag was inserted by Gibson assembly using plasmid AMp970 and primers AMo8866–AMo8869 to generate N-terminal tobacco etch virus protease cleavable $His_{x7}$-V5 tag-tagged *SGO1* (AMp1738) under the control of a *pCUP1* promoter. *sgo1-4R* was PCR amplified from plasmid AMp1340 and ligated into AMp1738 by Gibson assembly using primers AMo8850, AMo8851, AMo9124, and AMo9125 to generate N-terminal tobacco etch virus protease cleavable $His_{x7}$-V5 tag-tagged *sgo1-4R* (AMp1783).

A protease-deficient yeast strain (AMy8184) was transformed with the resulting plasmids (AMp1738 or AMp1783), and the transformants were inoculated into 8 liters uracil dropout medium. When $OD_{600}$ reached 0.5–0.7, 0.5 mM $CuSO_4$ was added to induce $His_{x7}$-V5-SGO1 (or *sgo1-4R*) expression. Cells were harvested 6 h after induction. Cell pellets were snap frozen in liquid nitrogen and ground to powder in a ball breaker machine (Retsch). All purification steps were performed at 4°C or on ice. Cell powder was resuspended in lysis buffer (25 mM Tris-HCl, pH 7.5, 150 mM NaCl, 1 mM $MgCl_2$, 10% glycerol, 0.1% NP-40, 0.05 mM EDTA, 0.05 mM EGTA, 1 mM DTT, 1× CLAAPE protease inhibitors [chymostatin, leupeptin, aprotinin, antipain, pepstatin, and E-64], 1 mM Pefabloc, 0.4 mM Na orthovanadate, 0.1 mM microcystin, 1 mM N-ethylmaleimide, 2 mM β-glycerophosphate, 1 mM Na pyrophosphate, 5 mM NaF, and complete EDTA-free protease inhibitor [Roche]). Cell lysates were treated with 40 U/ml benzonase for 1.5 h. The crude lysate was diluted with 25 mM sodium phosphate buffer, pH 7.5, 500 mM NaCl, 10% glycerol, 0.1% NP-40, 10 mM imidazole, 1 mM DTT, and 0.25 mM PMSF; cleared by ultracentrifugation (50,000 $g$ at 4°C for 30 min); filtered through a 0.22-μm filter; and loaded onto a HiTrap IMAC FF 1-ml column (GE Healthcare) charged with $Ni^{2+}$ and attached to an AKTA system. The column was washed with 25 mM sodium phosphate buffer, pH 7.5, 500 mM NaCl, 10% glycerol, 0.1% NP-40, 25 mM imidazole, 1 mM DTT, and 0.25 mM PMSF. Recombinant Sgo1 protein was

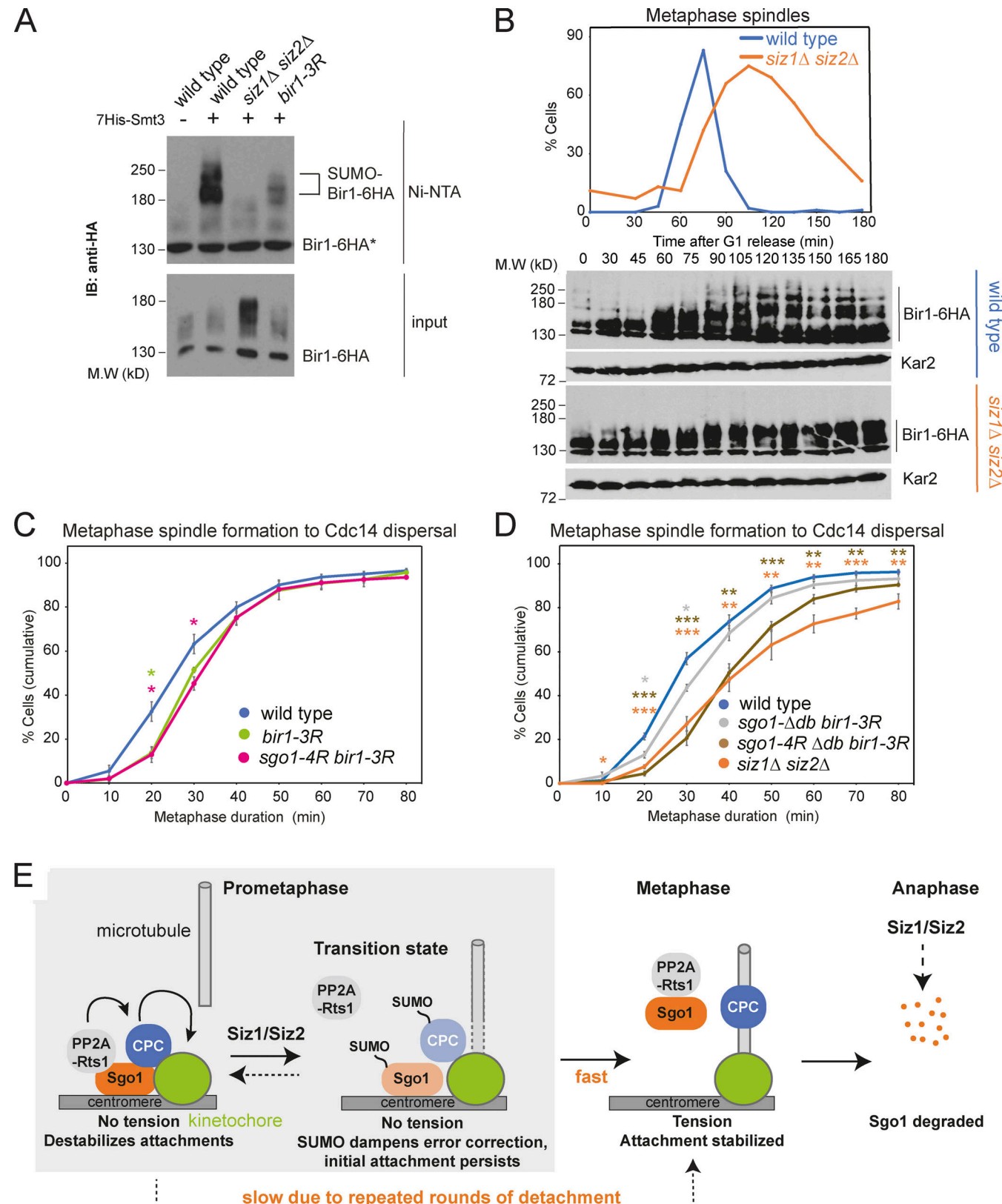

Figure 10. **Bir1 SUMOylation cooperates with Sgo1 SUMOylation to promote anaphase onset. (A)** Bir1 is SUMOylated in a Siz1/Siz2-dependent manner. Strains were *BIR1-6HA* (vector control, AMy23696; *HIS-SMT3*, AMy 23697), *BIR1-6HA siz1Δ siz2Δ* (AMy23698), and *bir1-K707R K732R K785R-6HA* (AMy29757). SUMO pull-down was performed as described in Fig. 2 A. IB, immunoblot; M.W, molecular weight. **(B)** Siz1/Siz2 changes posttranslational modifications on Bir1. *BIR1-6HA* (AMy3533) and *siz1Δ siz2Δ BIR1-6HA* (AMy7826) were released from α-factor arrest. Spindle morphology was analyzed as described in Fig. 1 D. Bir1 levels were analyzed by anti-HA(HA11) Western blot. Kar2 was a loading control. **(C)** *bir1-3R* caused a modest delay in metaphase. Strains carried

*CDC14-GFP* and *YFP-TUB1* and were wild type (AMy24174), *bir1-3R* (AMy29736), and *sgo1-4R bir1-3R* (AMy29734). Cells were imaged and quantified as described in Fig. 1, E and F. Shown are the average values of three to four independent experiments, with error bars representing standard error. Statistics: two-tailed Student's *t* test comparing to the wild-type strain (*, P < 0.05). **(D)** *bir1-3R sgo1-4R-Δdb* delayed metaphase to a similar extent to *siz1Δ siz2Δ*. Strains carried *CDC14-GFP* and *YFP-TUB1* and were wild type (AMy24174), *siz1Δ siz2Δ*(AMy24313), *sgo1-Δdb bir1-3R* (AMy30108), and *sgo1-4R-Δdb bir1-3R* (AMy30064). Shown are the average values of three to four independent experiments, with error bars representing standard error. Statistics: two-tailed Student's *t* test comparing to the wild-type strain (*, P < 0.05; **, P < 0.01; ***, P < 0.001). **(E)** Schematic model of how Siz1/Siz2-dependent SUMOylation might promote the metaphase–anaphase transition. In prometaphase, before kinetochore–microtubule attachment, a pool of Sgo1 and CPC is SUMOylated, promoting the release of PP2A-Rts1 and dampening CPC activity. The SUMO-dependent turnover of PP2A-Rts1/CPC at kinetochores prevents microtubule release at newly attached kinetochores, thereby stabilizing the attachment and allowing tension to be generated. In response to tension, Sgo1 is released from chromatin and is subsequently degraded by APC/C to fully turn off metaphase signaling pathways. In the absence of Siz1/Siz2, disassembly and turnover of pericentromeric complexes is inefficient, so PP2A-Rts1 and CPC remain on the kinetochore as microtubule capture occurs. Persistent CPC results in microtubule detachment and unattached kinetochores, which cause spurious SAC activity and delay anaphase entry.

eluted with an increasing imidazole gradient (25–500 mM) over 40 column volumes and then loaded onto a gel filtration Superose 6 10/300 column in 50 mM Tris-HCl, pH 7.5, 500 mM NaCl, 10% glycerol, 0.1 mM EDTA, 0.1 mM EGTA, 1 mM DTT, and 0.25 mM PMSF. Fractions containing the eluted protein were run on an SDS-PAGE gel, and the purity of recombinant Sgo1 was assessed by InstantBlue (Expedeon) staining.

### Cloning, expression, and purification of SUMOylation enzymes

SUMO E1 enzymes Uba2 and Aos1 (from vectors pET11a-*UBA2* and pET28a-*AOS1*; Windecker and Ulrich, 2008) were coexpressed in BL21 CodonPlus (DE3) RIL (Agilent Technologies) in the presence of 0.05 mM IPTG at 18°C for 20 h. Expression of SUMO E2 enzyme Ubc9 and SUMO Smt3 (pET21b-*UBC9* and pET21a-*SMT3*; Windecker and Ulrich, 2008) was induced by 0.1 mM IPTG at 18°C for 20 h in BL21 CodonPlus (DE3) RIL. E1, E2, and SUMO were purified using Ni-NTA agarose resin (Qiagen) with 20 mM sodium phosphate, pH 7.5, 500 mM NaCl, and 10–200 mM imidazole, followed by Superdex 200 16/600 column (hand poured using prep grade resin from GE Healthcare) in 50 mM Tris-HCl, pH 7.5, 150 mM NaCl, 1 mM DTT, and 10% glycerol.

*siz1-167-508* and *siz2-130-490* were amplified from plasmids pGEX-4T-1-*SIZ1-HF* (Windecker and Ulrich, 2008) and AMp1528 (YEPlac112-*SIZ2*; this work), respectively. The respective PCR product was cloned into pMAL-c2X to generate N-terminal maltose-binding protein (MBP)-tagged *siz1-167-508* (AMp1679) and N-terminal MBP-tagged *siz2-130-490* (AMp1680). Siz1 (167–508) or Siz2 (130–490) were expressed in Luria–Bertani broth containing 10 μM ZnCl₂, 0.1 mM IPTG, and appropriate antibiotics at 16°C for 20 h using BL21 CodonPlus (DE3) RIL (Agilent Technologies). Siz1 (167–508) or Siz2 (130–490) was purified from MBPTrap HP 1-ml column (GE Healthcare) with 50 mM Tris-HCl, pH 7.5, 150 mM NaCl, 10 μM ZnCl₂, and 0–10 mM maltose gradient, followed by gel filtration with a Superdex 200 16/600 column (hand-poured using preparation-grade resin from GE Healthcare) in 50 mM Tris-HCl, pH 7.5, 150 mM NaCl, 10 μM ZnCl₂, and 10% glycerol.

### In vitro SUMOylation assays and SUMOylated Sgo1 pull-down assays

Purified Sgo1 (1 μM) was mixed with 5 mM ATP, 15 μM SUMO, 0.5 μM E1 (unless otherwise indicated), 0.5 μM E2 (unless otherwise indicated), and 0.1 μM (unless otherwise indicated)

Siz1 (167–508) or Siz2 in a reaction buffer consisting of 25 mM Hepes, pH 7.5, 150 mM KCl, 10 mM MgCl₂, 15% glycerol, 0.1% NP-40, 0.1 mM DTT, and 0.25 mM PMSF. The reaction was incubated at 30°C for 2 h (unless otherwise indicated). For detection of SUMOylated Sgo1, the reaction was boiled in SDS sample buffer before analysis by anti-V5 Western blotting.

For Rts1 binding assay, the product of in vitro SUMOylation was incubated with anti-V5 (#MCA1360; Bio-Rad AbD Serotec)–coupled protein G magnetic Dynabeads (Thermo Fisher Scientific) in binding buffer A (25 mM Hepes, pH 7.5, 150 mM KCl, 2 mM MgCl₂, 15% glycerol, 0.1% NP-40, 0.1 mM EDTA, 0.5 mM EGTA, and 0.25 mM PMSF) for 2.5 h at 4°C. After washing three times with binding buffer A, beads were incubated with extract from strain AMy8832 for 2 h at 4°C. Beads were washed five times with binding buffer A + 10 mM *N*-ethylmaleimide and were heated at 65°C for 15 min to elute bound proteins. Sgo1 binding assay was performed similarly, except that Rts1-9Myc was immunoprecipitated from lysate of strain AM8832 by anti-Myc (9E10, #626802; BioLegend)–coupled protein G Dynabeads and subsequently incubated with products of in vitro SUMOylation (with or without ATP).

### Sgo1 half-life measurement

Cultures were inoculated to OD₆₀₀ = 0.2 in YEP + 2% raffinose + 0.3 mM adenine. 5 μg/ml α-factor was added for 1.5 h and re-added to 2.5 μg/ml every hour until the end of the experiment. *SGO1* overexpression was induced by the addition of 2% galactose for 30 min. De novo synthesis of Sgo1 protein was quenched by the addition of 2% glucose and 1 mg/ml cycloheximide (Acros Organics).

### Analysis of ubiquitination in vivo

For ectopically expressed *SGO1*, cells were grown in –Ura + 2% raffinose and arrested in α factor until >95% cells were in G1. 2% galactose was added to induce *GAL-SGO1* expression for 30 min before harvesting. For endogenously expressed *SGO1*, 400 ml cycling cells grown at room temperature was harvested. Cell harvest, lysis, and dilution were performed as described in Analysis of the in vivo SUMOylation. 10 μl was saved as input control, and the remaining supernatant was incubated with 80 μl of 50% slurry Ni-NTA agarose beads (Qiagen) with rotation at 4°C overnight. The beads were washed twice with buffer A + 0.05% Tween-20 and four times with buffer C (8 M urea, 100 mM sodium phosphate buffer, pH 6.3, and 10 mM Tris-HCl,

pH 6.3) + 0.05% Tween-20. Beads and input were heated at 60°C for 10 min in HU buffer (8 M urea, 200 mM Tris-HCl, pH 6.8, 1 mM EDTA, 5% SDS, 0.1% [wt/vol] bromophenol blue, and 1.5% DTT).

## ChIP
Metaphase-arrested cells were fixed in 1.1% formaldehyde for ≥30 min. Cells were washed twice with TBS (20 mM Tris-HCl, pH 7.5, and 150 mM NaCl), once with FA lysis buffer (50 mM Hepes-KOH, pH 7.5, 150 mM NaCl, 1 mM EDTA, 1% Triton X-100, and 0.1% sodium deoxycholate) + 0.1% SDS, and drop frozen in liquid nitrogen. Cells were lysed by bead beating in FA lysis buffer + 0.5% SDS + 1 mM PMSF + EDTA-free protease inhibitor cocktail (Roche). The resulting pellets were sonicated in a Bioruptor machine. Sheared chromatin was incubated with the relevant antibody and protein G Dynabeads (Thermo Fisher Scientific) at 4°C overnight. Anti-HA (12CA5, #1666606001; Roche Diagnostics) or anti-Flag M2 (# F1804; Sigma-Aldrich) antibodies were used as appropriate. Beads were washed once with FA lysis buffer + 0.1% SDS + 275 mM NaCl, once with FA lysis buffer + 0.1% SDS + 500 mM NaCl, once with wash buffer (10 mM Tris-HCl, pH 8, 0.25 M LiCl, 1 mM EDTA, 0.5% NP-40, and 0.5% sodium deoxycholate), and once with TE (10 mM Tris-HCl, pH 8, and 1 mM EDTA). Chromatin was eluted by boiling in the presence of 10% Chelex beads, treating with proteinase K (Life Technologies) for 30 min, and boiling for a further 10 min. qPCR was done using Express SybrGreenER (Thermo Fisher Scientific) on a Lightcycler machine (Roche). Primers used are listed in Table S4. ChIP enrichment was determined using the formula $E^{-\Delta Ct}$, where Ct = cycle threshold; $\Delta Ct = Ct_{(ChIP)} - [Ct_{(input)} - \log E(\text{input dilution factor})]$; and $E$ = primer efficiency.

## Biorientation assay in fixed and live cells
Cells carrying *pMET-CDC20*, *SPC42-tdTOMATO*, and *CEN4-GFP* were arrested in metaphase in the presence of nocodazole and benomyl as described in Yeast growth and synchronization. For analyzing biorientation in metaphase-arrested cells (Fig. 6, B and C), drugs were washed out by filtering with 10 times the volume of YEP, and cells were released into YPDA + 8 mM methionine (to maintain Cdc20 depletion). Samples were taken at the indicated time points, fixed in 3.7% formaldehyde for 9 min, washed in 80% ethanol, and resuspended in 1 µg/ml DAPI in PBS. 200 cells were analyzed for each time point.

For analyzing biorientation in cells going into anaphase (Fig. 6, E–I; and Fig. 7, E–G), nocodazole-arrested cells were loaded onto the ONIX Microfluidic Perfusion System (CellAsic). Imaging started as soon as cells were released into –Met medium.

## Plasmid loss assay
To measure the rate of plasmid loss, yeast strains transformed with plasmid pRS316 were grown overnight in –Ura medium, before diluting to $A_{600} = 0.2$ in 3 ml YPDA. After 3 h, cells were plated on –Ura or YPDA plates. 1,000–2,000 cells were counted for each transformant. The percentage of cells grown on YPDA over that grown on –Ura was calculated, and shown are the average percentages from three independent transformants of each strain background.

## Ipl1-GFP intensity measurement
Cells were released from G1 arrest in methionine dropout medium and imaged using the microfluidics device described above. Line scans were performed across kinetochore foci of single cells, using an ImageJ plug-in designed for this study: https://doi.org/10.5281/zenodo.3442325.

## Sgo1-6His-3Flag IP for MS
The following procedures were performed in triplicate for each strain analyzed. Cultures were inoculated to $OD_{600} = 0.2$ in 2 liters YPDA and grown at 30°C until $OD_{600}$ reached 1.8–2. Benomyl was dissolved in YPDA by boiling. Cooled medium was added to cells to achieve a final benomyl concentration of 30 µg/ml. Approximately 20 g of cell pellets were harvested and drop frozen in liquid nitrogen. Cells were ground with a Retsch RM100 electric mortar-grinder. The resulting powders were lysed and treated with benzonase as described in Purification of recombinant protein. Cell debris was removed by centrifugation at 3,600 rpm for 10 min, and the soluble fraction was incubated with Protein G Dynabeads, previously conjugated to mouse anti-Flag (M2, #F1804; Sigma-Aldrich) for 2 h at 4°C. Beads were washed four times in lysis buffer before elution at 50°C for 15 min in NuPAGE LDS sample buffer (Thermo Fisher Scientific) supplemented with 5% β-mercaptoethanol.

Protein samples were run on a bis-Tris gel and stained with Imperial Protein Stain (Thermo Fisher Scientific). Excised gel pieces were cut into 1-mm³ cubes and destained with 50 mM ammonium bicarbonate and acetonitrile (ACN). Proteins were reduced in 10 mM DTT or 30 min at 37°C and alkylated in 55 mM iodoacetamide for 20 min at room temperature. Digestion was performed in 12.5 ng/µl trypsin for 16 h at 37°C. Peptides were eluted with 80% ACN + 0.1% trifluoroacetic acid (TFA) solution and dried in a SpeedVac vacuum. Peptides were resuspended in 100 µl of 0.1% TFA and passed through a StageTip containing C18 discs. Peptides were eluted from StageTips in 40 µl of 80% ACN in 0.1% TFA and concentrated down to 1 µl by vacuum. All samples were then prepared for liquid chromatography–MS/MS analysis by diluting them to 5 µl with 0.1% TFA. Liquid chromatography–MS-analyses were performed on an Orbitrap Fusion Lumos Tribrid Mass Spectrometer (Thermo Fisher Scientific) coupled online to an Ultimate 3000 RSLCnano Systems (Dionex; Thermo Fisher Scientific). Peptides were separated on a 50-cm EASY-Spray column (Thermo Fisher Scientific) assembled in an EASY-Spray source (Thermo Fisher Scientific) and operated at a constant temperature of 50°C. Mobile phase A consisted of 0.1% formic acid in water; mobile phase B consisted of 80% ACN and 0.1% formic acid. Peptides were loaded onto the column at a flow rate of 0.3 µl/min and eluted at a flow rate of 0.25 µl/min according to the following gradient: 2–40% buffer B in 150 min, then to 95% in 11 min. Survey scans were performed at 120,000 resolution (scan range 350–1,500 m/z) with an ion target of $4.0 \times 10^5$. The RF lens was set to 30%, and the maximum injection time to 50 ms. The cycle time was set to 3 s, and dynamic exclusion to 60 s. MS2 was performed in the

ion trap on rapid scan mode with ion target of $2.0 \times 10^4$ and higher-energy collisional dissociation fragmentation with normalized collision energy of 27 (Olsen et al., 2007). The isolation window in the quadrupole was set at 1.4 Thomson, and the maximum injection time was set to 35 ms. Only ions with charge between 2 and 7 were selected for MS2. The MaxQuant software platform v1.6.1.0 (Cox and Mann, 2008) was used to process raw files, and search was conducted against the *Saccharomyces cerevisiae* (strain S288C) complete/reference proteome set of Saccharomyces Genome Database (released in June 2019), using the Andromeda search engine (Cox et al., 2011). The first search peptide tolerance was set to 20 ppm, and the main search peptide tolerance was set to 4.5 ppm. Isotope mass tolerance was 2 ppm and maximum charge of 7. A maximum of two missed cleavages were allowed. Fixed modifications, cysteine carbamidomethylation; variable modifications, oxidation of methionine, acetylation of lysine and the N-terminus, and phosphorylation of serine, threonine, and tyrosine. Label-free quantitation (LFQ) analysis was performed by using the MaxLFQ algorithm (Cox et al., 2014). False discovery rate was set to 1%. Flow-through sample data were used to identify Sgo1 interactors. DEP R package was used to analyze the LFQ data (Zhang et al., 2018). The R script used is accessible via GitHub: https://github.com/BXSu/JCB_mass_spec. Raw data are available on the PRIDE database http://www.ebi.ac.uk/pride, with the accession no. PXD019287.

### Sgo1-Rts1 co-IP for Western blot analysis
800 ml of cycling cells were grown at room temperature and harvested when $OD_{600}$ reached 0.8–1. Cells were ground in a ball-breaker machine (Retsch), lysed, and treated with benzonase as described in Purification of recombinant protein. Soluble cell lysate was incubated with rabbit IgG coupled to Dynabeads M-270 epoxy beads for 1.5 h at 4°C, then washed and eluted as described for MS.

### Rts1 intensity measurement
Cells were released from G1 arrest in methionine dropout medium and imaged on Ibidi dishes as described in Metaphase duration measurements. Peak Rts1-GFP intensity colocalizing with Mtw1-tdTOMATO signal was measured, and background signal was subtracted.

### Statistical analyses
For each time point of time course experiments, the average ± SEM is shown. Dot plots display mean (red dot) ± SD (red line). Upper and lower hinges of box plots represent the first and third quartiles, respectively, and the middle lines represent median values. Upper and lower whiskers of box plots extend to 1.5 times interquartile range. Comparisons of two groups were performed by one- or two-tailed Student's *t* test, as appropriate for the experiment. Comparisons of data pooled from independent experiments were performed by Fisher's exact tests. P values < 0.05 were considered significant. Significant values were shown by *, P < 0.05; **, P < 0.01; and ***, P < 0.001.

### Online supplemental material
Fig. S1 outlines the high copy suppressor screen for *SGO1* overexpression phenotype and shows that the metaphase delay of *siz1Δ siz2Δ* is Cdc55 dependent. Fig. S2 shows Coomassie stains of recombinant Sgo1 and in vivo SUMOylation of Sgo1 truncation mutants. Fig. S3 shows data supporting the idea that SUMOylation disrupts Sgo1–Rts1 interaction. Fig. S4 shows SUMOylation of Sgo1-associated proteins in metaphase-arrested cells and phenotypic analyses of *sgo1-Δdb*, *sgo1-4R*, and *bir1-3R* single, double, and triple mutants. Table S1 is a complete list of high copy suppressors of *GAL-SGO1* sickness identified by the screen shown in Fig. S1 A. Table S2 lists the yeast strains used in the study. Table S3 provides the plasmids used in the study. Table S4 identifies the oligonucleotides used in the study. Table S5 is a list of the proteins interacting with immunoprecipitated Sgo1-4R or Sgo1 in wild-type or *siz1Δ siz2Δ* cells as measured by label-free quantitative MS.

## Acknowledgments
We are grateful to Liz Bayne (University of Edinburgh, Edinburgh, UK), Richard Hallberg, Yoshiko Kikuchi (University of Tokyo, Tokyo, Japan), and Helle Ulrich for yeast strains and plasmids and to Weronika Borek for help with R programming. We thank Andrew Goryachev, Lori Koch, Federico Pelisch, Emily Petty, and Lorraine Pillus for helpful discussions and comments on the manuscript. We are grateful for support from the Wellcome Centre for Cell Biology Proteomics core facility, the Centre Optical Imaging Laboratory, and the Edinburgh Protein Purification Facility.

This work was funded by the Wellcome Trust through Senior Research Fellowships to A.L. Marston (107827), A.A. Jeyaprakash (202811), and J. Rappsilber (103139), an instrument grant (108504), and core funding for the Wellcome Centre for Cell Biology (203149).

The authors declare no competing financial interests.

Author contributions: Conceptualization: A.L. Marston, X.B. Su, C. Schaffner, M. Wang, O.O. Nerusheva, D. Clift, A. Wallek, and Z. Storchova. Methodology: X.B. Su, M. Wang, C. Schaffner, C. Spanos, O.O. Nerusheva, M. Tatham, and R. Hay. Investigation: X.B. Su, M. Wang, C. Schaffner, C. Spanos, D. Clift, O.O. Nerusheva, A. Wallek, and Y. Wu. Software: D.A. Kelly. Writing – original draft: X.B. Su and A.L. Marston. Writing – review and editing: all authors. Visualization: X.B. Su, A.A. Jeyaprakash, and A.L. Marston. Supervision: A.L. Marston, J. Rappsilber, R. Hay, and Z. Storchova; Funding acquisition, A.L. Marston.

Submitted: 19 May 2020

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

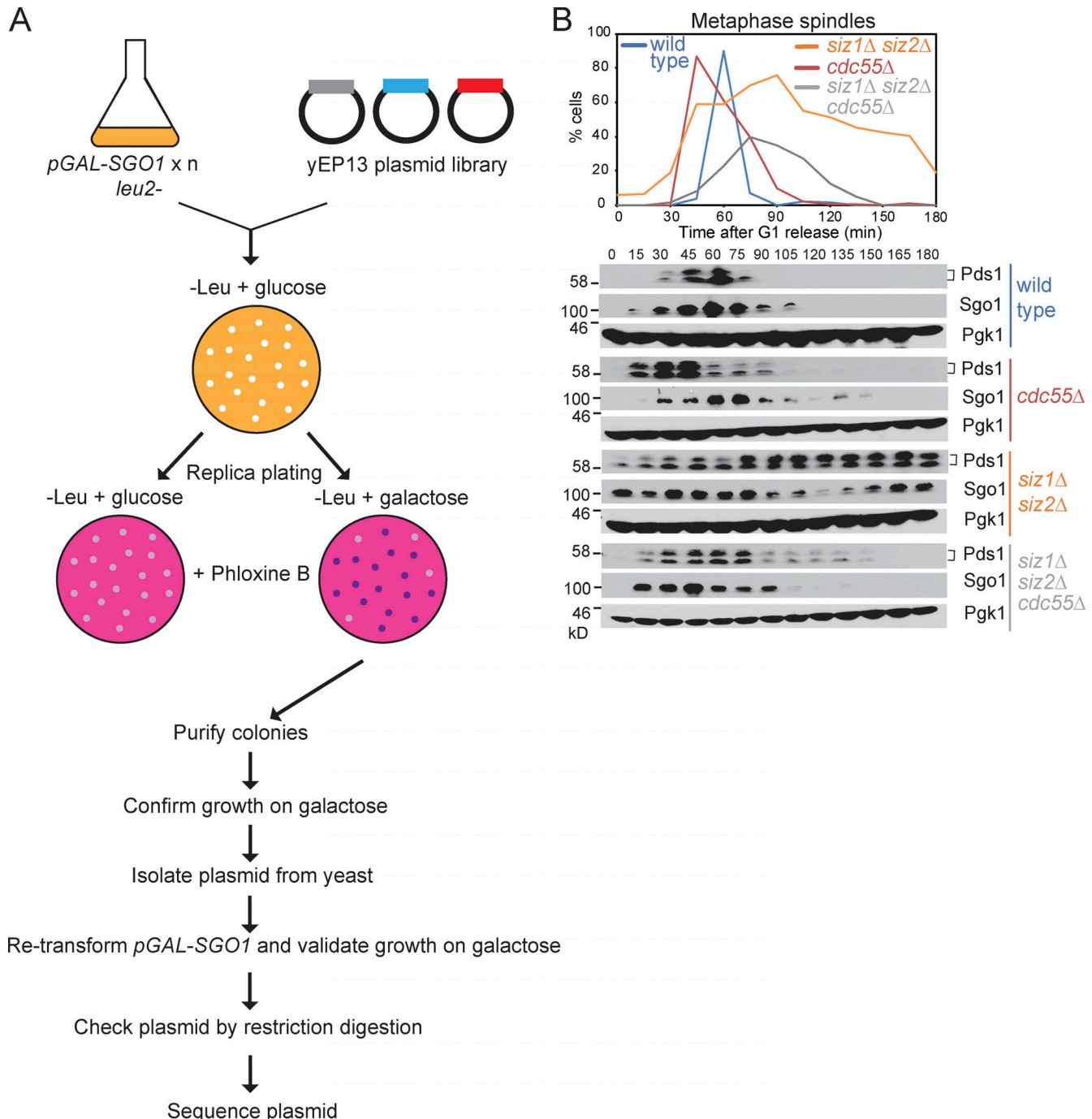

Figure S1. **Identification of SUMO ligases as Sgo1 regulators. (A)** Overview of multicopy suppressor screen to identify Sgo1 antagonists. **(B)** Deletion of *CDC55* partially alleviates the metaphase delay phenotype of the *siz1Δ siz2Δ* mutant. Mitotic time course analysis was performed as described in Fig. 1 D for the following strains: wild type (AMy8467), *cdc55Δ* (AMy8779), *siz1Δ siz2Δ* (AMy8452), and *siz1Δ siz2Δ cdc55Δ* (AMy8637).

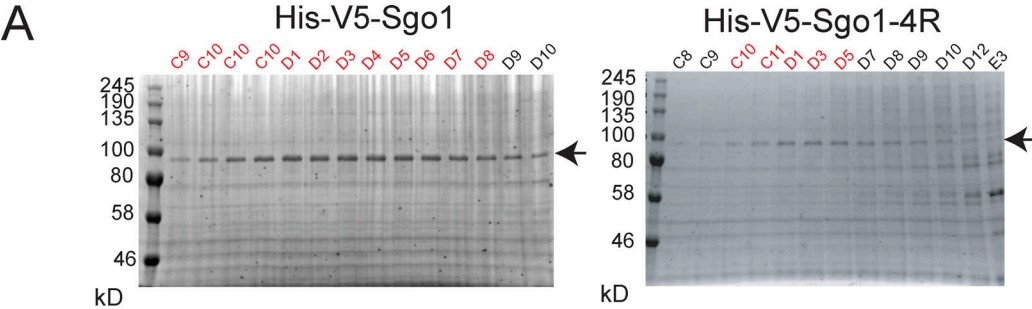

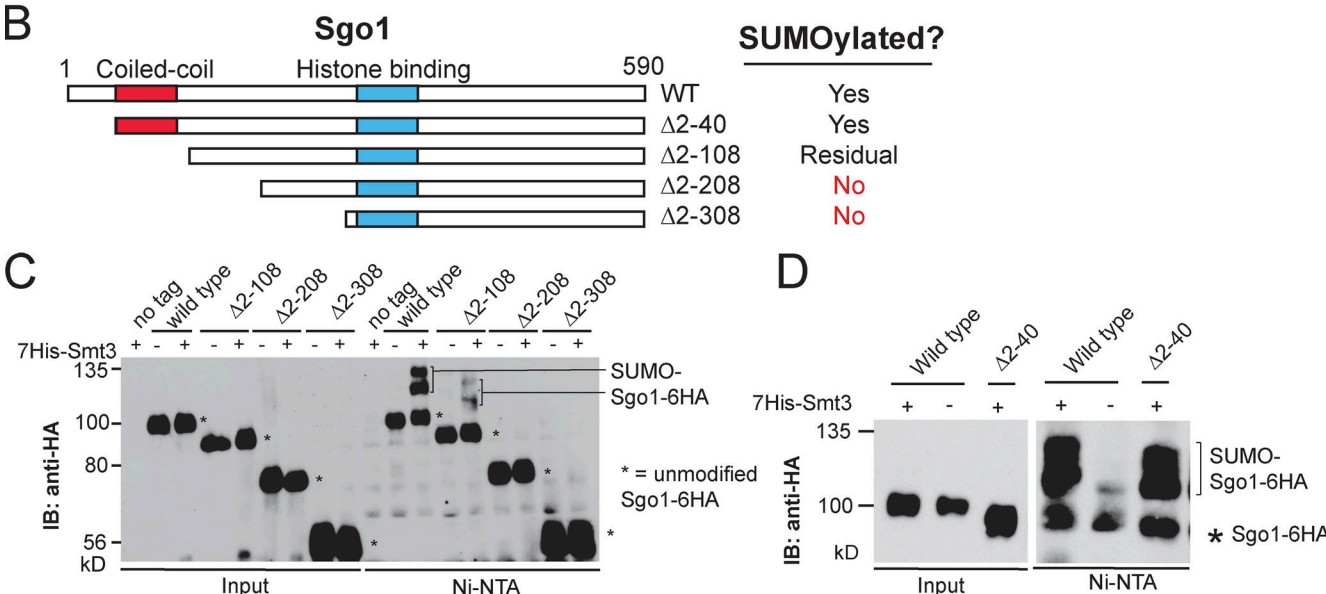

Figure S2. **Recombinant Sgo1 was successfully purified and maximum Sgo1 SUMOylation required the coiled-coil domain. (A)** Purification of Sgo1. Coomassie staining confirmed the successful purification of wild-type and Sgo1-4R proteins. **(B)** Schematics describing the truncation mutants generated for Sgo1. The conserved coiled-coil and basic domains are highlighted in red and blue, respectively. Results from C and D are summarized on the right. **(C)** The first 208 amino acids are essential for Sgo1 SUMOylation. In vivo SUMOylation was assessed for the following Sgo1-6HA tagged strains as described in Fig. 2 A, together with the indicated negative controls: wild type (AMy7654), *sgo1Δ2-108* (AMy14764), *sgo1Δ 2–208* (AMy14765), and *sgo1Δ2-308* (AMy14766). Un-modified Sgo1 bands are marked with asterisks. IB, immunoblot. **(D)** The coiled-coil domain is required for maximum Sgo1 SUMOylation. In vivo SUMOylation was assessed for the following Sgo1-6HA tagged strains: wild type (AMy7654) and *sgo1Δ2-40* (AMy18194).

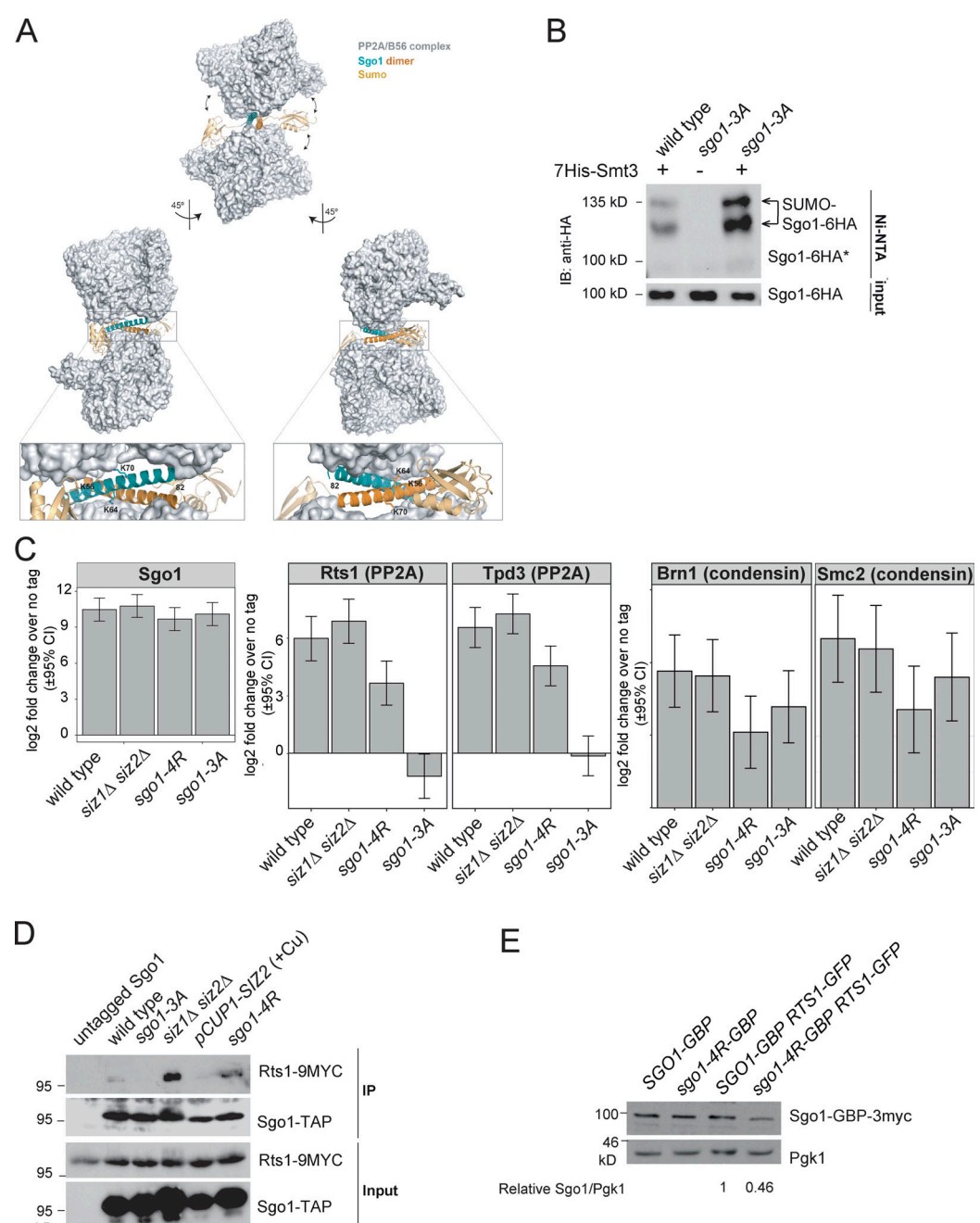

Figure S3. **SUMOylation disrupts Sgo1–Rts1 interaction. (A)** Structural modeling predicts that SUMOylation on the coiled-coil domain of Sgo1 is incompatible with Sgo1–PP2A interaction. *S. cerevisiae* Sgo1–PP2A interaction was modeled based on structural information obtained from cocrystallized human Sgo1(51–96) and PP2A using Phyre2 web portal (Kelley et al., 2015). Potential consequences of SUMOylation were modeled using the molecular graphic program PyMOL (v2.0; Schrödinger). According to this model, Lys64 and Lys70 are critically positioned at the binding surface with no room to accommodate a bulkier modification such as SUMOylation. Lys56 is exposed to the solvent, but the attachment of SUMO (highlighted in gold) is expected to result in steric clashes with PP2A and weaken Sgo1-PP2A binding. Structural information is unavailable beyond Leu82, so Lys85 could not be included in this model. **(B)** Rts1 binding is not required for Sgo1 SUMOylation. 6-HA–tagged wild type (AMy906) and Sgo1-3A(AMy25988) were analyzed as described in Fig. 2 A. IB, immunoblot. **(C)** Global Sgo1–Rts1 interaction was mildly increased in *siz1Δ siz2Δ* and moderately reduced in *sgo1-4R*. Global Sgo1–Brn1 interaction was intact in *siz1Δ siz2Δ* and moderately reduced in *sgo1-4R*. Strains used carried 6His-3Flag–tagged Sgo1: wild type (AMy23137), *siz1Δ siz2Δ* (AMy29146), *sgo1-4R* (AMy26329), and *sgo1-3A* (AMy29203). Cells were arrested in benomyl at 30°C, and proteins copurifying with Sgo1-6His-3Flag were analyzed by label-free quantitative MS in triplicate. Data were analyzed by the DEP R package, and error bars represent standard deviation. CI, confidence interval. **(D)** Global Sgo1–Rts1 interaction was increased in *siz1Δ siz2Δ* and reduced in the *SIZ2*-overexpressing strain. *sgo1-4R* did not show a strong impact on the interaction. Cycling cells grown at room temperature were used for co-IP analysis as described in Materials and methods. Strains used carried Sgo1-TAP and Rts1-9Myc and were wild type (AMy9144), *sgo1-3A* (AMy9145), *siz1Δ siz2Δ* (AMy21943), *pCUP1-SIZ2* (AMy27970), and *sgo1-4R* (AMy26090) or untagged Sgo1 (AMy4721). Sgo1-TAP and Rts1-9Myc were detected by PAP and anti-Myc Western blotting, respectively. **(E)** Sgo1-4R expression level was reduced when Rts1-GBP was also present. Protein extracts from strains used for Fig. 9, E–G, were analyzed by Western blot. Signal intensities were measured using ImageJ. Ratios of Sgo1-GBP-3myc to Pgk1 were calculated, and the intensity for *SGO1-GBP-3MYC RTS1-nfGFP* was set to 1.

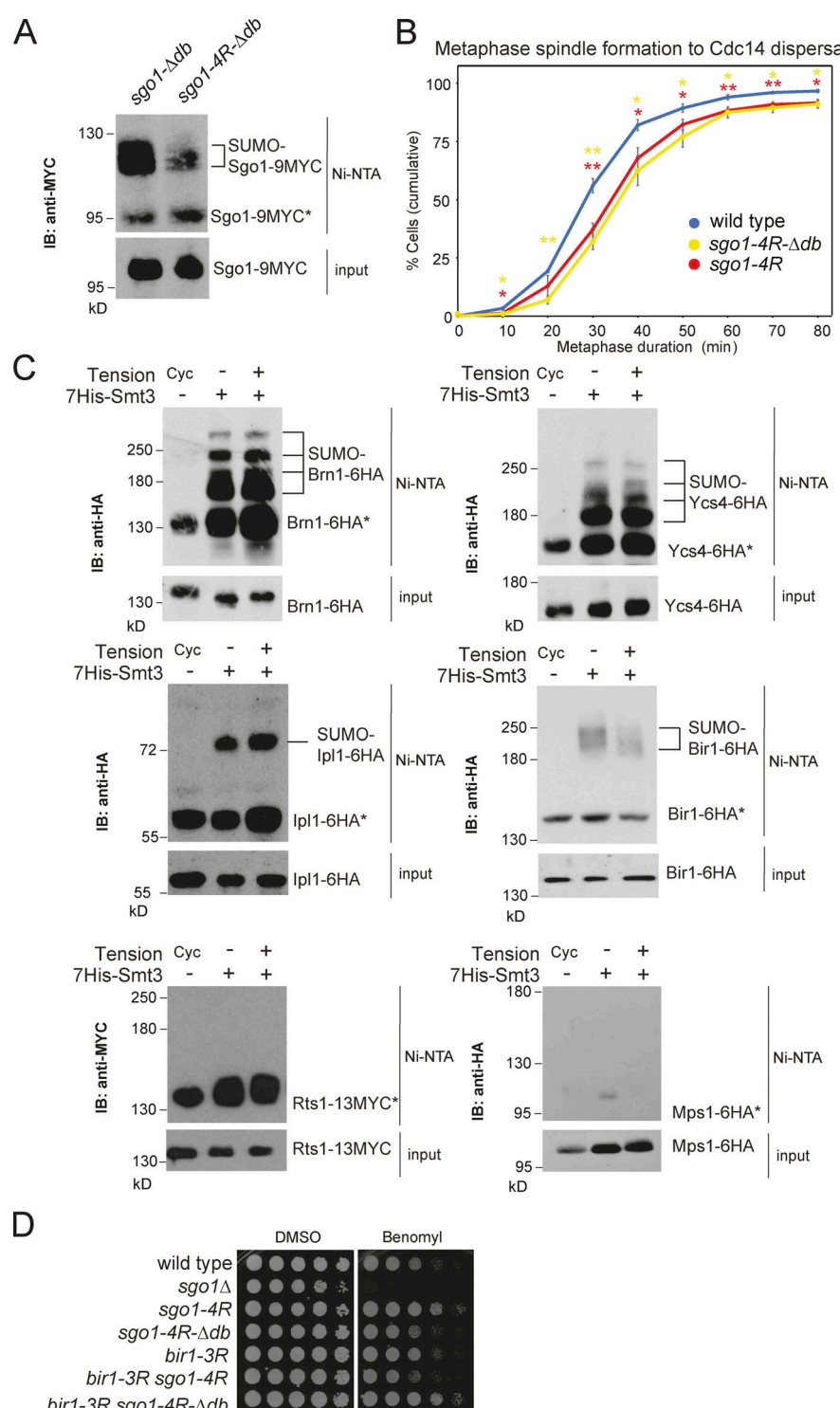

Figure S4. **Combinatorial effects of SUMOylation on pericentromeric proteins. (A)** Sgo1-4RΔdb showed reduced SUMOylation compared with Sgo1-Δdb. SUMO pull-down was performed for *sgo1-Δdb-9MYC* (AMy29896) and *sgo1-4R-Δdb-9MYC* (AMy29996). **(B)** *sgo1-4RΔdb* showed a metaphase delay similar to *sgo1-4R*. Strains imaged contained *CDC14-GFP* and *YFP-TUB1* and were wild type (AMy24174), *sgo1-4R* (AMy29305), and *sgo1-4RΔdb* (AMy30024). Shown are the average of three to four independent experiments, and error bars represent standard error. Statistics: one-tailed Student's *t* test (*, P < 0.05; **, P < 0.01). IB, immunoblot. **(C)** Components of condensin and CPC complex are SUMOylated in metaphase, but Rts1 and Mps1 are not SUMOylated in metaphase. Strains used carried *pMET-CDC20* and were transformed with *His-SMT3* or empty vector: *BRN1-6HA* (AMy8955), *YCS4-6HA* (AMy8953), *IPL1-6HA* (AMy6937), *BIR1-6HA* (AMy6941), *RTS1-13MYC* (AMy8951), and *MPS1-6HA* (AMy7450). Vector controls were harvested as cycling cells (Cyc). *HIS-SMT3* transformants were metaphase arrested in the presence or absence of tension. SUMO pull-down was performed as described in Materials and methods. **(D)** *bir1-3R* and *sgo1-4RΔdb* single- and double-mutant cells grew similarly to wild type or with mildly improved growth on benomyl. Serially diluted cells of wild type (AMy1176), *sgo1Δ* (AMy827), *sgo1-4R* (AMy21705), *sgo1-4RΔdb* (AMy29901), *bir1-3R* (AM29717), *bir1-3R sgo1-4R* (AM29735), and *bir1-3R sgo1-4RΔdb* (AMy30060) were plated on medium containing 10 µg/ml benomyl or DMSO (solvent).

Provided online are five tables. Table S1 is a complete list of high copy suppressors of *GAL-SGO1* sickness. Table S2 lists the yeast strains used in this study. Table S3 provides the plasmids used in this study. Table S4 identifies the oligonucleotides used in this study. Table S5 is a list of the proteins interacting with immunoprecipitated Sgo1-4R or Sgo1 in wild-type or *siz1Δ siz2Δ* cells as measured by label-free quantitative MS.

