## [Peer Review File · The Journal of Cell Biology]

SUMOylation stabilizes sister kinetochore biorientation to allow timely anaphase

Xue Su, Menglu Wang, Claudia Schaffner, Olga Nerusheva, Dean Clift, Christos Spanos, David Kelly, Michael Tatham, Andreas Wallek, Yehui Wu, Juri Rappsilber, A. Arockia Jeyaparakash, Zuzana Storchova, Ronald Hay, and Adele Marston

Corresponding Author(s): Adele Marston, University of Edinburgh

Review Timeline:

Submission Date:	2020-05-19
Editorial Decision:	2020-06-12
Revision Received:	2021-02-18
Editorial Decision:	2021-03-12
Revision Received:	2021-03-22

Monitoring Editor: Arshad Desai

Scientific Editor: Melina Casadio

Transaction Report:

DOI: <https://doi.org/10.1083/jcb.202005130>

June 13, 2020

Re: JCB manuscript #202005130

Prof. Adele L Marston
University of Edinburgh
The Wellcome Trust Centre for Cell Biology School of Biological Sciences
Michael Swann Building
Edinburgh EH9 3BF

Dear Prof. Marston,

Thank you for submitting your manuscript entitled "SUMOylation inactivates shugoshin-PP2ARts1 to stabilize sister kinetochore biorientation" and thank you for your patience with the peer review process. Your manuscript has been assessed by three expert reviewers, whose comments are appended below. Although the reviewers express potential interest in this work, significant concerns raised by them unfortunately preclude publication of the current version of the manuscript in JCB.

You will see that all reviewers find the results you present of interest. They however raise significant issues that undermine many aspects of the conclusions and the presented model. The referees are not convinced that the data exclude SUMOylation of Sgo1 at anaphase onset promoting degradation, rather than promoting biorientation and reducing PP2A-Rts1 binding. Significant revisions seem necessary to address the reviewer comments, in particular to explain the difference in severity of the phenotypes of Sgo1 versus Siz1/2 mutants. While we editorially agree that the reviewers all make valid points, following the reviewer suggestions has the potential to significantly improve the study and make the findings more impactful. Although we appreciate that this will require new experimental effort, if you are able to address all the reviewer points to the best of your ability, we would be willing to consider a revision. Note that any revision received will need to be re-assessed by all three reviewers.

Please let us know if you are able to address the reviewer criticisms and wish to submit a revised manuscript to JCB. As you know, the typical timeframe for revisions is three to four months. However, we at JCB realize that the implementation of social distancing and shelter-in-place measures that limit spread of COVID-19 also pose challenges to scientific researchers. Therefore, JCB has waived the revision time limit. We recommend that you reach out to the editors once your lab has reopened to decide on an appropriate time frame for resubmission. Please note that papers are generally considered through only one revision cycle, so any revised manuscript will likely be either accepted or rejected.

If you choose to revise and resubmit your manuscript, please also attend to the following editorial points. Please direct any editorial questions to the journal office.

GENERAL GUIDELINES:

Text limits: Character count is < 40,000, not including spaces. Count includes title page, abstract, introduction, results, discussion, acknowledgments, and figure legends. Count does not include materials and methods, references, tables, or supplemental legends.

Figures: Your manuscript may have up to 10 main text figures. To avoid delays in production, figures must be prepared according to the policies outlined in our Instructions to Authors, under Data Presentation, <http://jcb.rupress.org/site/misc/ifora.xhtml>. All figures in accepted manuscripts will be screened prior to publication.

IMPORTANT: It is JCB policy that if requested, original data images must be made available. Failure to provide original images upon request will result in unavoidable delays in publication. Please ensure that you have access to all original microscopy and blot data images before submitting your revision.

Supplemental information: There are strict limits on the allowable amount of supplemental data. Your manuscript may have up to 5 supplemental figures. Up to 10 supplemental videos or flash animations are allowed. A summary of all supplemental material should appear at the end of the Materials and methods section.

If you choose to resubmit, please include a cover letter addressing the reviewers' comments point by point. Please also highlight all changes in the text of the manuscript.

Regardless of how you choose to proceed, we hope that the comments below will prove constructive as your work progresses. We would be happy to discuss them further once you've had a chance to consider the points raised. You can contact the journal office with any questions, cellbio@rockefeller.edu or call (212) 327-8588.

Thank you for thinking of JCB as an appropriate place to publish your work.

Sincerely,

Arshad Desai, PhD
Editor, Journal of Cell Biology

Melina Casadio, PhD
Senior Scientific Editor, Journal of Cell Biology

Reviewer #1 (Comments to the Authors (Required)):

In the manuscript by Su, Marston and colleagues, the authors identify a role of SUMOylation in the regulation of the yeast shugoshin Sgo1. The authors determine that SUMOylation of Sgo1 negatively regulates its activity in chromosome biorientation. They use a variety of cell biological and biochemical techniques to thoroughly and convincingly tease apart a fairly subtle phenotype, identify and characterize mutants that partially prevent Sgo1 SUMOylation, and determine a mechanism of action in preventing the binding of the phosphatase Rts1. There is an impressive amount of work that has gone into this study, and the experiments appear to have been performed and interpreted in an extremely rigorous manner. Of course, the subtlety of the phenotypes and the lack of evidence that the mechanism is conserved in other species lessen the impact of the study a bit. Furthermore, I am a bit confused about the temporal aspects of how the authors envision the regulation of Sgo1 by SUMOylation increasing the fidelity of chromosome segregation.

Major points:

1a. I do not understand how the model for the function of Sgo1 SUMOylation agrees with the data. The authors set up the key question in the field in the intro as: "with the initiation of cohesin cleavage at anaphase onset, tension between sister kinetochores is lost, which could lead to re-activation of the error correction and biorientation pathways." Since SUMOylation appears to go away completely at the onset of anaphase (Fig. 2D), it seems unlikely that it could contribute to error correction silencing at this stage. In the discussion, the authors conclude "Sgo1 SUMOylation likely promotes anaphase entry by silencing the error correction process." In addition, their model figure (Fig. 7H) shows SUMOylation of stable attachments whereas unstable attachments are not SUMOylated. However, in Figure 2F, the authors demonstrate that Sgo1 has very little SUMOylation at metaphase-arrested cells with stable attachments. Furthermore, they see an increase in SUMOylation when they destabilize attachments with microtubule poisons. To me, these data appear to directly contradict their model.

Could it be that instead SUMOylation occurs predominantly at the inner centromeres of unattached kinetochores to allow for attachments to occur more easily? This could prevent kinetochores from having too much "error correction" activity prior to MT attachment, which might prevent the efficient capture of MTs in the first place. With this model, a lack of SUMOylation would also cause a metaphase delay and would better fit with the observed timing and attachment states at which SUMOylation is observed. Perhaps the increased CEN-GFP dot switching comes from an inability to convert attachments on the sides of MTs to end-on attachments.

A time course of Sgo1 SUMOylation from G1 to Ccd20-depleted metaphase arrest might help clarify some of these issues. Perhaps SUMOylation in metaphase arrest is only extremely low in comparison to unattached kinetochores, but would be relatively high in comparison to prometaphase and anaphase cells.

1b. On a similar note, the authors observe an increase in Ipl1-GFP levels at centromeres in mutants that prevent SUMOylation at inter-kinetochore distances that strongly indicate the cells are well into anaphase (Figure 6D). In wild-type cells, both the SUMOylation and Sgo1 itself would be gone at this stage. This suggests that, while the effect of SUMOylation on Sgo1 degradation "is not required for efficient anaphase entry", it does still have a significant effect on this assay.

Minor points:

2. I think it would be helpful to see the upper area of the blots for the Input in the SUMOylation detection assays (Figs 2B,D,E,F). I assume that if there were any visible bands in the input corresponding to the size of the SUMOylation products present without enrichment that the authors would show them, but it would be nice for the readers to be able to look for themselves. Perhaps those could be shown in the supplement.

3. Some of the graphs show significance with asterisks (e.g. Figs. 1F, 3E-F) and others do not (e.g. Figs. 1G, 5B, S1C). It is unclear if in those cases there are no significant differences or if it was simply not noted in these cases.

4. I don't understand the purpose of the experiment shown in Fig. 4B. The authors show that artificially and indirectly targeting a phosphatase to the outer-kinetochore prevents cells from entering metaphase. It feels like the experiment that would connect this to SUMOylation is missing.

5. On page 9, the authors write "likely due to the mild overexpression from the galactose-inducible

promoter." Expression from the galactose-inducible promoter is notoriously extremely strong.

6. On page 10-11, the authors write "siz1 Δ siz2 Δ ipl1-as behaved similarly as ipl1-as (Figure 6B)", however the metaphase timing for siz1 Δ siz2 Δ ipl1-as in the graph looks more similar to siz1 Δ siz2 Δ than ipl1-as. The authors should clarify whether this statement is referring to the microscopy or the western blots and attempt to explain the discrepancy between the two.

7. On page 11, the authors write "suggesting that Sgo1 SUMOylation itself may be sufficient for reducing the levels of CPC upon the satisfaction of biorientation." The word "sufficient" here is unclear, as CPC levels are reduced over time even in the presence of the mutants. Do the authors mean "Sgo1 SUMOylation is sufficient for the function of SUMOylation in reducing the levels of CPC"?

Reviewer #2 (Comments to the Authors (Required)):

In this study Xue and colleagues discover a novel role for SUMOylation in stabilizing kinetochore biorientation by modulating the interactions of Shugoshin (Sgo1), a key adaptor protein and its regulatory partners PP2A-Rts1 phosphatase and its effectors including Ipl1 kinase, all localized to the yeast pericentromere. Through an unbiased genetic screen, the authors identify the SUMO E3 ligase Siz2 as a high copy suppressor of a metaphase delay caused by overexpression of Sgo1. Following this they perform a series of in vitro and in vivo experiments to show that Sgo1 is SUMOylated specifically at the pericentromere during metaphase by Siz2 and / or Siz1. The authors follow a narrative where Sgo1 SUMOylation leads to reduced binding of PP2A-Rts1 which in turn leads to destabilization of the chromosome passenger complex and timely removal of Ipl1 thereby facilitating anaphase entry. Through these results, this study teases apart a role for SUMO mediated signalling of Sgo1 which is distinct from the role of SUMO and Ubiquitin mediated regulation of Sgo1 protein stability.

While the overall findings and claims are interesting the study suffers from the general problem that whereas deletion of the candidate SUMO E3 ligases Siz1/2 has a large impact on mitotic delay, mutations that are predicted to impact SUMOylation of Sgo1 specifically do not recapitulate the effects to the same extent and in several experiments the impact is not obvious. This leads to doubts as to whether the proposed mechanism can be explained by Sgo1 regulation or by a broader role for the SUMO machinery in chromosome segregation. Below, I outline specific concerns that would require additional attention. In all, I think the work needs significant revision before being considered for publication:

Major comments:

Fig 1D. The effect of siz1/siz2 deletion on metaphase delay is much greater than that of sgo1 deletion alone. The authors claim that the sgo1/siz1/2 triple mutant partially rescues the siz1/2 double mutant but another way of looking at this figure is that siz1/2 mutation greatly exacerbates the sgo1 defect. Clearly, this indicates that Siz1/Siz2 likely have other more prominent targets than Sgo1 through which it is exerting its effect on the timing of anaphase onset. This should be addressed. Further it is not clear to me whether the suppression of siz1/2 by sgo1 deletion is significant in both figure 1D and 1G. Good stats are needed here.

Fig 2A and text. The authors argue that only a small pool of Sgo1 is SUMOylated. This could be a

consequence of a large amount of deSUMOylase activity in the extracts. To better assess this experiments need to be performed in absence of the SUMO protease ULP1/2 or using a pan SUMO protease inhibitor such as NEM.

In figure 2D as the time course proceeds from the release of G1, the levels of Sgo1 peak in metaphase and then fall (presumably due to degradation by APC). The authors imply that SUMOylation is maximal before anaphase onset but it is more likely that SUMOylation simply follows Sgo1 abundance.

Fig 2F. Considering the above, if Sgo1 would be maximally SUMOylated in metaphase, the time of maximal tension, this is inconsistent with the experiment in 2F that indicates that in metaphase arrested cells (by depletion of Cdc20) SUMOylated Sgo1 is only observed under 'no tension' conditions. Further, more details are needed here. How is CDC20 inhibited? Do the authors have any control for an efficient mitotic arrest? Why use both benomyl and nocodazole combined?

Fig 3E again the phenotype of sgo1-4R does not recapitulate siz1/siz2 mutant indicating Siz1 and 2 play mitotic roles elsewhere. This should be more clearly acknowledged.

On Sgo1 lysine mutants: Figure S3E is not convincing, all strains looks to the same. The used of the deltaDB, (destruction box) is not explained in the text. Furthermore, it's not clear from 3F that the over expression of SGO1-4R rescues Siz2 overexpression less well than the wild type Sgo1 over expression. Comparing 1C with 3F, they look the same, it is mostly the single mutants that are different. This should be compared within a single experimental setup.

Fig S4. Deletion of Siz1/2 does not affect ubiquitylation. This means some other factors are likely responsible. However this would not be inconsistent with over expression of Siz1 and 2 suppressing ubiquitylation? This would be a trivial explanation for suppression seen in Fig 1A.

Fig 5E. Is the frequency of switching increased in siz1/2 and sgo1-3R mutant simply because the cells spend longer time in metaphase?

Figure 6A. The SAC dependency is unconvincing. Siz1/2 mutants have an effect on biorientation (Fig 5E) yet MAD2 deletion has barely any effect on the delay. Is this significant? Clearly there is a non-SAC dependent component to the metaphase delay in Siz1/2 mutants.

Fig 6B, here the ipl1 and ipl1 suz1/2 mutants are deemed similar. While indeed, Pds1 levels appear similar, the frequency of metaphase spindles seem very different and more so than in other instances in the paper where authors claimed differences. It appears that the suz1/2 mutations are suppressed by the additional ipl1 mutation indicating that some of the siz1/2 delay is caused by error correction in a Siz1/2 independent manner.

Fig 6D. Given that Ipl1 levels are increased and Ipl1 homolog AurkB has been reported to be SUMOylated which was found to regulate its levels (Fernández-Miranda et al., 2010), it should be tested whether the metaphase delay could be a result of direct SUMOylation and stabilization of Ipl1.

Similarly, 7A/B. Rts1 shows reduced binding to Sgo1 when Sgo1 is SUMOylated. One reason for this might be NOT because RTS1 cannot bind to SGO1-SUMO but because Rts1 is itself SUMOylated. Are there any higher molecular weight species seen that could indicate this?

Fig 7C. The model that SUMOylation interferes with binding of PP2A-Rts1 to Sgo1 is central to this paper. It is supported by the immunofluorescence data in this figure, and *in vitro* SUMOylation assays. The native mass spec data in Figure S7 is less clear as it indicates general binding defects of the Sgo1-4R mutant, possibly due to tagging as the authors indicate making it difficult to determine if there is indeed a loss in Rts1 binding. It would strengthen this paper greatly if a more direct *in vivo* co-IP could be performed in WT and *sgo1-4R* conditions to detect interaction between Sgo1 and PP2A-Rts1, e.g. using different tags and by direct blotting for PPs1/Rts1.

Fig 7F,G. While switching frequency is used as a measure of stable biorientation, it would help to determine whether Rts1 tethering also lead to a metaphase delay?

Figure S5A. (A) The SPB separation appears to happen faster in the *sgo1* mutant although the initial establishment of biorientation is delayed as shown in (B). This appears add odds. Can this be explained?

Minor comments

Top of page 6. CDC55 is not explained or referenced.

Fig S3A does not seem useful to show as the experiment did not work. It would be better to simply explain in the lack of SUMO MS data in the text

Figure 7A labelling of panels is shifted

Figure S5A. Typo in figure legend "...and initial establishment of biorientation for the experiment (missing "B") shown in Figure 5C-F..."

Top of page 11: "...suggesting that SUMOylation and CPC work in the same pathway to regulate the metaphase-anaphase transition."

The argumentation here is a bit confusing. I think it makes more sense to state that the anaphase delay by *siz1/2* is in part imposed by *ipl1* i.e. error correction.

Page 11 middle Figure S6D and E should be C and D

Reviewer #3 (Comments to the Authors (Required)):

Sister chromatids biorientation during mitosis is crucial for accurate chromosome segregation. Surveillance pathways correct erroneous kinetochore-microtubule attachments through tension sensing and arrest cell cycle until biorientation is established. The budding yeast Shugoshin (Sgo1) protein promotes biorientation by recruiting condensin and PP2A-Rts1 and maintaining the CPC at centromere as well as pericentromere regions. Once tension is achieved, Sgo1 and its associated effectors are removed from pericentromere to promote cell cycle progression. Consistent with this, SGO1 overexpression results in a pronounced metaphase delay. This work deals with Sgo1 turnover during mitotic progression. Su et al. started from an unbiased genetic screen for negative regulators of Sgo1 and identified the SUMO ligase Siz2. They showed that Siz1/Siz2 promotes efficient anaphase onset partially by antagonizing Sgo1-mediated pathways but not affecting Sgo1 localization. Specifically, Siz1/Siz2 sumoylates Sgo1 to reduce its interaction with PP2A-Rts1, and this may in turn lower centromeric CPC, although mechanism for CPC removal has not been

investigated here.

Overall, this study provides a number of new findings to address how protein sumoylation may regulate Shugoshin to modulate the kinase-phosphatase network. However, a number of concerns should be addressed, as summarized below:

Major points:

1. Figure 1B and 1C, since Siz1 over-expression was not analyzed, it is improper to assume that its over-expression can mimic that of Siz2. Despite that Siz1 and Siz2 are considered paralogs, they do possess differences. In the same vein, siz1 and siz2 single deletion mutants should have been analyzed in Figure 1D/E/F to demonstrate their redundant role in this process.
2. Figure 2A is trivial, consider removal. Figure 2B should include the effect of deleting either Siz1 or Siz2 to verify their possible redundancy towards Sgo1. Why is the contaminant Sgs1-HA (marked by asterisk) in the Ni-elution not the same in the same experiment, given the same amount of Sgs1-HA in the input? Reproducibility issue?
3. The fluctuation of Sgo1 protein levels during the cell cycle may complicate interpretation of the timing of Sgo1 SUMOylation (Figure 2D). This issue could be addressed by the sgo1- Δ db mutant, which does not impair the metaphase-anaphase transition.
4. Figure 3: a total of six lysines exist in the region (41-108), why not mutate all of them? It is clear from Figure 3B/3C, sgo1-4R mutant still retains significant amount of sumoylation and a rather modest phenotype.
5. Sgo1 is stabilized in siz1 Δ siz2 Δ , ubc9-1 and slx5 Δ (Fig. S1B, S4A and S4B), suggesting the SUMOylation and ubiquitination catalyzed by Slx5-Slx8 of Sgo1 are required for its efficient degradation. The possibility that Slx5 may target Sgo1 cannot be excluded, since the ubiquitination assay analyzed overexpressed Sgo1 (Figure S4C). To address this concern, chromosomal tagged Sgo1 should be analyzed instead. The result shown in Figure S3E (sgo1-2R/ Δ db) is inconclusive since sumoylation of Sgo1 is not completely eliminated. To test whether SUMOylation of Sgo1 is a degradation-independent mechanism of Sgo1 inactivation, consider combining sgo1- Δ db mutant and sgo1 sumoylation-deficient or siz1/siz2 mutant.
6. Figure 5: if sgo1 mutant has an unstable biorientation, is this mutant sensitive to nocodazole? Does sgo1 sumoylation-defective mutant show elevated chromosome/plasmid loss?
7. In Figure 7F, artificial tethering of Sgo1 to Rts1 increased association of CEN4-GFP, regardless of wild type Sgo1 or sgo1-4R. However, mis-segregation is increased when Rts1 is tethered to Sgo1-4R, to a greater extent than tethered to wild type Sgo1 (Figure 7G), how to explain this observation?

Minor points:

1. The abstract should be re-written to clarify the authors' model of how sumoylation of Sgo1 may stabilize biorientation supported by results. The finding of CPC removal defect is modest with no mechanistic insight provided.
2. The legend of Figure 1D: 'siz1 Δ siz2 Δ cells are delayed in metaphase and sgo1 Δ had no additive effect' does not match the main text 'sgo1 Δ reduced the duration of metaphase in siz1 Δ siz2 Δ cells' on Page 5. It seems that the effect of sgo1 Δ is subtle. Another note: page and line number should be included in the text.
3. 'We independently verified this.....under tension (Figure S6D and S6E).....(Figure 6D and S6D)'. The corresponding figures should be Figure S6C and S6D, Figure 6D and S6C. Overall, supplementary figures are arranged in a haphazard way with no particular coordination with the main text, making it difficult for readers to follow.

Reviewer #1 (Comments to the Authors (Required)):

In the manuscript by Su, Marston and colleagues, the authors identify a role of SUMOylation in the regulation of the yeast shugoshin Sgo1. The authors determine that SUMOylation of Sgo1 negatively regulates its activity in chromosome biorientation. They use a variety of cell biological and biochemical techniques to thoroughly and convincingly tease apart a fairly subtle phenotype, identify and characterize mutants that partially prevent Sgo1 SUMOylation, and determine a mechanism of action in preventing the binding of the phosphatase Rts1. There is an impressive amount of work that has gone into this study, and the experiments appear to have been performed and interpreted in an extremely rigorous manner. Of course, the subtlety of the phenotypes and the lack of evidence that the mechanism is conserved in other species lessen the impact of the study a bit. Furthermore, I am a bit confused about the temporal aspects of how the authors envision the regulation of Sgo1 by SUMOylation increasing the fidelity of chromosome segregation.

We thank the reviewer for their support of our work and their suggestions which have been extremely valuable in clarifying some key points in our manuscript.

Although we do not know if the details of the mechanism we describe is conserved, a proteome-wide study (Nie et al., 2015) found *Schizosaccharomyces pombe* shugoshin, Sgo2, to be SUMOylated and we confirmed this to be the case (new Figure S2B). Human Sgo1 was also reported to be SUMOylated in a global study (Schimmel et al., 2014).

In our revision we also looked more broadly into SUMOylation of pericentromere components and included a new figure and supplementary figure showing this data (Figures 8 and S8). We now provide evidence that both Ipl1 (Aurora B) and Bir1 (survivin) are SUMOylated (Figure S8C) and that SUMOylation of the CPC subunit, Bir1, works together with SUMO-Sgo1 to allow timely anaphase (Figure 8D). Previous studies have described SUMOylation of various CPC subunits, including Aurora B itself in a range of systems, including *C. elegans*, *Xenopus* and human cells (Fernández-Miranda et al., 2010; Pelisch et al., 2017, 2019; Davis-Roca et al., 2018). Together, these observations suggest that the mechanisms we describe will be generally important in ensuring the fidelity of chromosome segregation.

While the phenotype of cells with reduced Sgo1 SUMOylation (*sgo1-4R*) is subtle, it is highly reproducible. Mutants lacking the SUMO ligases, Siz1 and Siz2, show a stronger phenotype, indicating that either additional Sgo1 SUMOylation sites exist (indeed Sgo1-4R has a low level of residual SUMOylation) or that there are other key target proteins. As described above, in our revised manuscript we present new data showing that Bir1 is an important Siz1 Siz2 substrate for the mechanism we describe (Figure 8). Furthermore, the combination of reduced Sgo1 and Bir1 SUMOylation, together with stabilization of Sgo1 (*sgo1-4R Δdb bir1-3R*) results in a metaphase delay comparable to cells lacking Siz1 and Siz2 (Figure 8D). Finally, it is important to note that subtle changes in surveillance pathways monitoring chromosome segregation are more likely to result in aneuploidy than major insults that would block proliferation. We would therefore argue that the identification of such mechanisms is highly relevant to our understanding of how aneuploidy arises e.g. in cancer.

We agree that, as originally presented, the temporal aspects of our model were not clear and we apologise for the confusion. We have now clarified this point with new data – this is addressed in response to this reviewer's point 1 below.

Major points:

1a. I do not understand how the model for the function of Sgo1 SUMOylation agrees with the data. The authors set up the key question in the field in the intro as: "with the initiation of cohesin cleavage at anaphase onset, tension between sister kinetochores is lost, which could lead to re-activation of the error correction and biorientation pathways." Since SUMOylation appears to go away completely at the onset of anaphase (Fig. 2D), it seems unlikely that it could contribute to error correction silencing at this stage. In the discussion, the authors conclude "Sgo1 SUMOylation likely promotes anaphase entry by silencing the error correction process." In addition, their model figure (Fig. 7H) shows SUMOylation of stable attachments whereas unstable attachments are not SUMOylated. However, in Figure 2F, the authors demonstrate that Sgo1 has very little SUMOylation at metaphase-arrested cells with stable attachments. Furthermore, they see an increase in SUMOylation when they destabilize attachments with microtubule poisons. To me, these data appear to directly contradict their model.

Could it be that instead SUMOylation occurs predominantly at the inner centromeres of unattached kinetochores to allow for attachments to occur more easily? This could prevent kinetochores from having too much "error correction" activity prior to MT attachment, which might prevent the efficient capture of MTs in the first place. With this model, a lack of SUMOylation would also cause a metaphase delay and would better fit with the observed timing and attachment states at which SUMOylation is observed. Perhaps the increased CEN-GFP dot switching comes from an inability to convert attachments on the sides of MTs to end-on attachments.

A time course of Sgo1 SUMOylation from G1 to Ccd20-depleted metaphase arrest might help clarify some of these issues. Perhaps SUMOylation in metaphase arrest is only extremely low in comparison to unattached kinetochores, but would be relatively high in comparison to prometaphase and anaphase cells.

The reviewer is correct. The time at which SUMOylation is important was not resolved in our previous submission and we apologise for the confusion. Following the suggestion of the reviewer, we analysed Sgo1 SUMOylation in a time course as cells progressed from G1 into a metaphase arrest with tension (Cdc20 depletion). This data (Figure 2C) shows that SUMOylation is highest prior to the establishment of the arrest in metaphase. Taken together with the fact that we observe Sgo1 SUMOylation in metaphase-arrested cells with nocodazole (Figure 2D), this finding indicates that Sgo1 SUMOylation occurs when kinetochores are unattached/not under tension and when Sgo1 is associated with the chromatin (note Sgo1 is released from the chromatin under tension). We therefore agree with the reviewer that Sgo1-SUMO likely plays its most critical role in pro-metaphase, as kinetochore-microtubule attachments are being made. However, we found no evidence that Sgo1-SUMO or Siz1/Siz2 are important for the initial establishment of attachments both from our biorientation assays where we observe a single chromosome (Figure 5C) or in a new experiment where we imaged all kinetochores simultaneously (Figure 5A). Instead, we believe that SUMOylation is important to stabilize these attachments by dampening the activity of CPC and thereby preventing premature detachment before they have had the opportunity to generate tension. Two key pieces of evidence support this interpretation. First, we observe increased re-association of GFP-labelled centromeres in metaphase-arrested SUMO mutants (termed "switching" in our manuscript; Figure S5C). Second, our revised manuscript also presents the new observation that Sgo1 reassociates with kinetochores

after its initial release in a larger fraction of *siz1Δ siz2Δ* cells than wild type (Figure 4H-I). These observations are consistent with futile cycles of error correction in SUMO mutants that extend metaphase and delay commitment to anaphase.

In addition to inclusion of the new experiments in support of this hypothesis we have revised the model to reflect this interpretation and to include the cell cycle stage (now shown in Figure 8E). We have also revised the text accordingly, including removal of the sentence related to anaphase in the introduction, and revision of the statement in the discussion which now reads "Instead, we propose that SUMOylation promotes anaphase entry by dampening the error correction machinery as microtubules establish stable interactions with kinetochores in prometaphase.

1b. On a similar note, the authors observe an increase in Ipl1-GFP levels at centromeres in mutants that prevent SUMOylation at inter-kinetochore distances that strongly indicate the cells are well into anaphase (Figure 6D). In wild-type cells, both the SUMOylation and Sgo1 itself would be gone at this stage. This suggests that, while the effect of SUMOylation on Sgo1 degradation "is not required for efficient anaphase entry", it does still have a significant effect on this assay.

Again, the reviewer makes a good point. Although preventing Sgo1 degradation is not on its own sufficient to delay cells in metaphase (Figure S4F), we have now obtained evidence that Sgo1 stabilization does contribute to the delayed anaphase entry in *siz1Δ siz2Δ* mutants. In new Figure 8D, we find that preventing Sgo1 degradation (by deletion of its destruction box) exacerbates the metaphase delay of cells with reduced Sgo1 and Bir1 SUMOylation, resulting in a phenotype similar to *siz1Δ siz2Δ*. This data shows that stabilization of Sgo1 contributes to the metaphase delay in SUMO-deficient cells and adds further support to the idea that SUMOylation is important for commitment to anaphase.

Minor points:

2. I think it would be helpful to see the upper area of the blots for the Input in the SUMOylation detection assays (Figs 2B,D,E,F). I assume that if there were any visible bands in the input corresponding to the size of the SUMOylation products present without enrichment that the authors would show them, but it would be nice for the readers to be able to look for themselves. Perhaps those could be shown in the supplement.

We included the upper area of the blots for the inputs in Figure 2A, which shows that SUMOylated products are not readily detectable without enrichment, confirming that only a small fraction of the Sgo1 pool is SUMOylated at any one time.

3. Some of the graphs show significance with asterisks (e.g. Figs. 1F, 3E-F) and others do not (e.g. Figs. 1G, 5B, S1C). It is unclear if in those cases there are no significant differences or if it was simply not noted in these cases.

In the revised manuscript, where differences are significant, we have included asterisks.

4. I don't understand the purpose of the experiment shown in Fig. 4B. The authors show that artificially and indirectly targeting a phosphatase to the outer-kinetochore prevents cells from entering metaphase. It feels like the experiment that would connect this to SUMOylation is

missing.

The reviewer is correct that this figure was not directly related to SUMOylation in our previous manuscript. However, we believe it is important because although our previous work (Nerusheva et al., 2014) demonstrated the tension-dependent release of Sgo1, it did not confirm that this was biologically important. Our demonstration that Sgo1 association with kinetochores hinders the metaphase-anaphase transition is important knowledge for interpretation of the remainder of the manuscript where we explore the effect of SUMOylation on persistence of the Sgo1-dependent signalling pathway that delays cells in metaphase. We also added a new observation in which Sgo1 reassociation with kinetochores was increased in the SUMO mutants (Figure 4H-I), which could contribute to their delay in the metaphase-anaphase transition.

5. On page 9, the authors write "likely due to the mild overexpression from the galactose-inducible promoter." Expression from the galactose-inducible promoter is notoriously extremely strong.

These experiments were performed using a very low concentration of galactose (0.1%, rather than the typical 1%), to keep expression as close to that of endogenous Sgo1 as possible. We clarified this point in the revised manuscript.

6. On page 10-11, the authors write "*siz1Δ siz2Δ ipl1-as* behaved similarly as *ipl1-as* (Figure 6B)", however the metaphase timing for *siz1Δ siz2Δ ipl1-as* in the graph looks more similar to *siz1Δ siz2Δ* than *ipl1-as*. The authors should clarify whether this statement is referring to the microscopy or the western blots and attempt to explain the discrepancy between the two.

We agree with the reviewer that there was a discrepancy in the western and spindle counting in the *ipl1-as* experiment. We believe that the western blotting most accurately reflects the phenotype since, for reasons that are unclear, *ipl1-as* cells displayed aberrant spindle morphology, confounding accurate scoring. Due to the difficulties with interpreting this experiment we have removed it from the manuscript.

Instead, we took a different approach to examine the requirement for CPC for the metaphase delay in *siz1Δ siz2Δ* cells, focusing on the Ipl1/Aurora B targeting subunit, Bir1. In this experiment, we degraded Bir1 using the auxin-inducible degron after cells had entered the cell cycle (bud emergence in >80% cells) and visualized Cdc14-GFP and YFP-Tub1 to distinguish metaphase and anaphase (Cdc14 is released from the nucleolus at anaphase). The new data (Figure 6C) clearly show that Bir1 degradation advances the timing of anaphase in *siz1Δ siz2Δ* cells.

7. On page 11, the authors write "suggesting that Sgo1 SUMOylation itself may be sufficient for reducing the levels of CPC upon the satisfaction of biorientation." The word "sufficient" here is unclear, as CPC levels are reduced over time even in the presence of the mutants. Do the authors mean "Sgo1 SUMOylation is sufficient for the function of SUMOylation in reducing the levels of CPC"?

We have changed the text based on the reviewer's recommendation.

Reviewer #2 (Comments to the Authors (Required)):

In this study Xue and colleagues discover a novel role for SUMOylation in stabilizing kinetochore biorientation by modulating the interactions of Shugoshin (Sgo1), a key adaptor protein and its regulatory partners PP2A-Rts1 phosphatase and its effectors including Ipl1 kinase, all localized to the yeast pericentromere. Through an unbiased genetic screen, the authors identify the SUMO E3 ligase Siz2 as a high copy suppressor of a metaphase delay caused by overexpression of Sgo1. Following this they perform a series of in vitro and in vivo experiments to show that Sgo1 is SUMOylated specifically at the pericentromere during metaphase by Siz2 and / or Siz1. The authors follow a narrative where Sgo1 SUMOylation leads to reduced binding of PP2A-Rts1 which in turn leads to destabilization of the chromosome passenger complex and timely removal of Ipl1 thereby facilitating anaphase entry. Through these results, this study teases apart a role for SUMO mediated signalling of Sgo1 which is distinct from the role of SUMO and Ubiquitin mediated regulation of Sgo1 protein stability.

While the overall findings and claims are interesting the study suffers from the general problem that whereas deletion of the candidate SUMO E3 ligases Siz1/2 has a large impact on mitotic delay, mutations that are predicted to impact SUMOylation of Sgo1 specifically do not recapitulate the effects to the same extent and in several experiments the impact is not obvious. This leads to doubts as to whether the proposed mechanism can be explained by Sgo1 regulation or by a broader role for the SUMO machinery in chromosome segregation. Below, I outline specific concerns that would require additional attention. In all, I think the work needs significant revision before being considered for publication:

Thank you to the reviewer for their constructive comments which have helped us substantially revise our manuscript. Of particular note, in response to the comment above, we have explored the role of SUMO in the metaphase-anaphase transition more broadly and now present data showing that combinatorial SUMOylation of Sgo1 and Bir1 regulate anaphase onset. This is described in more detail below.

Major comments:

Fig 1D. The effect of siz1/siz2 deletion on metaphase delay is much greater than that of sgo1 deletion alone. The authors claim that the sgo1/siz1/2 triple mutant partially rescues the siz1/2 double mutant but another way of looking at this figure is that siz1/2 mutation greatly exacerbates the sgo1 defect. Clearly, this indicates that Siz1/Siz2 likely have other more prominent targets than Sgo1 through which it is exerting its effect on the timing of anaphase onset. This should be addressed. Further it is not clear to me whether the suppression of siz1/2 by sgo1 deletion is significant in both figure 1D and 1G. Good stats are needed here.

There may be a mis-understanding here. The purpose of this figure is to test whether the metaphase delay caused by the absence of SUMOylation (i.e. in *siz1Δ siz2Δ*) could be caused by Sgo1. The *sgo1Δ* mutant is a null loss of function mutation and any effect of this mutation on the cell cycle must be unrelated to SUMOylation (Note: *sgo1Δ* are delayed in metaphase entry, rather than duration). Therefore, comparison of *sgo1Δ* to *siz1Δ siz2Δ* is not a useful comparison in addressing our question. Rather, the useful comparison is *siz1Δ siz2Δ* vs. *siz1Δ siz2Δ sgo1Δ*, where it can be observed that metaphase duration is shorter in *siz1Δ siz2Δ sgo1Δ* compared to *siz1Δ siz2Δ*, both as judged by reduced persistence of

Pds1 and reduced accumulation of metaphase spindles (area under the graph is smaller). Nevertheless, we take the point of the reviewer that the rescue is not complete because *sgo1Δ siz1Δ siz2Δ* show a slightly greater delay than *sgo1Δ* alone, which suggests that Siz1/Siz2 may have other important targets. We explore this in new Figure 8.

We have also clarified the rescue of *siz1Δ siz2Δ* by *sgo1Δ* in two independent ways in the revised manuscript. First, following comments from reviewer 1, we addressed the discrepancy between Pds1 western blotting and spindle morphology as methods to measure the duration of metaphase in Figure 1D. We realised that alteration of spindle morphology in *siz1Δ siz2Δ* cells, where cells carry thick bundles extending well beyond 2 μm, confounded this analysis. Therefore, we re-analysed the immunofluorescence slides and set a more stringent criteria where metaphase spindle morphology is defined as those with lengths < 2 μm. After this re-evaluation, Pds1 western blotting and spindle morphology both lead to the same conclusion: metaphase duration is shorter in *siz1Δ siz2Δ sgo1Δ* than in *siz1Δ siz2Δ*. Second, we replaced original Figure 1G, which used the auxin-induced degron to degrade Sgo1 in live cell imaging, with a more rigorous experiment performed in triplicate in fixed cells. After repeated live cell imaging attempts, it became apparent that the auxin analog NAA caused toxicity even to wild type cells under our imaging conditions. We therefore performed time course analysis in flasks and analysed fixed samples carrying Cdc14-GFP and YFP-Tub1 by fluorescence microscopy at defined timepoints. This also allowed us to analyse many cells (100-200) at each time point in three biological replicates. The new data (shown in Figure 1G) shows that the rescue of the *siz1Δ siz2Δ* metaphase delay by Sgo1 degradation is statistically significant. Together, the experiments in Figure 1D and 1G show that the presence of Sgo1 is at least partially responsible for the metaphase delay in *siz1Δ siz2Δ* cells.

Fig 2A and text. The authors argue that only a small pool of Sgo1 is SUMOylated. This could be a consequence of a large amount of deSUMOylase activity in the extracts. To better assess this experiments need to be performed in absence of the SUMO protease ULP1/2 or using a pan SUMO protease inhibitor such as NEM.

In budding yeast, Ulp1 is also required for generating SUMO precursors for the SUMO E1 reaction and so inactivation of Ulp1 would confound our analysis. Following the reviewer's suggestion, we depleted Ulp2, using the auxin-inducible degron system and assessed the effects on Sgo1 SUMOylation. However, we did not observe increased Sgo1 SUMOylation in these cells (See Figure R1 below), arguing that Ulp2-dependent de-SUMOylation does not have a major effect on steady state Sgo1-SUMO levels. As this does not extend the conclusions of our study, we chose not to include this experiment in the manuscript but provide it here for the reviewers' reference.

Moreover, all of the SUMOylation pulldown experiments were performed under strong denaturing conditions, using a well-established protocol that has been used in many studies (e.g. (Psakhye et al., 2019)). Yeast cells were rapidly harvested on ice in the presence of trichloroacetic acid which precipitates all proteins and thus inhibits enzymatic activities. NEM is therefore not necessary and also incompatible with the protocol because it active towards sulfhydryls only at pH 6.5- 7.5. The SUMO pulldown was then performed in the presence of 6M guanidine hydrochloride and 8M urea, also strongly denaturing conditions.

Therefore, we believe that our conclusion that only a small fraction of Sgo1 is SUMOylated is valid.

Sgo1 SUMOylation in metaphase-arrested, no tension, *ULP2*-depleted cells

Figure R1: Sgo1 SUMOylation is not increased in the absence of Ulp2. Cells were released from G1 and arrested in metaphase by Cdc20 depletion (*pMET-CDC20* in the presence of methionine) in the absence of tension (addition of nocodazole and benomyl). Ulp2-aid degradation was induced by the addition of auxin (NAA) upon the emergence of small budded cells (which is also the time at which Sgo1 starts to be synthesised). Sgo1 SUMOylation pulldown assay was performed as described in Figure 2.

In figure 2D as the time course proceeds from the release of G1, the levels of Sgo1 peak in metaphase and then fall (presumably due to degradation by APC). The authors imply that SUMOylation is maximal before anaphase onset but it is more likely that SUMOylation simply follows Sgo1 abundance.

The reviewer is correct that it was not possible in our prior experiment to determine whether the alterations in Sgo1-SUMO forms were due to cell cycle-dependent changes in Sgo1 SUMOylation or Sgo1 abundance. We have addressed this in two different experiments where Sgo1-SUMOylation levels were altered independent of changes in Sgo1 abundance. In the experiment shown in new Figure S4G we analysed SUMOylation of non-degradable Sgo1 (lacking its destruction box) as cells progressed from G1 into anaphase. This clearly shows that Sgo1 SUMOylation is maximum just prior to the metaphase peak (45 mins), and declines at anaphase onset (60 mins), while Sgo1 abundance does not change over this transition. Furthermore, we examined SUMOylation of Sgo1 as cells progressed from G1 into a metaphase arrest in the presence of spindle tension. This is shown in new Figure 2C where we observed maximum SUMOylation prior to maximum metaphase arrest, followed by a substantial decrease in SUMOylation in the metaphase arrest, without changes in protein abundance. Therefore, Sgo1 SUMOylation is maximal just prior to metaphase. Taken with the data shown in Figure 2D, we conclude that Sgo1 SUMOylation is maximal in prometaphase, i.e. in the presence of unattached kinetochores and the absence of tension. We are grateful to the reviewer for raising this point as addressing it has greatly helped clarify our conclusions.

Fig 2F. Considering the above, if Sgo1 would be maximally SUMOylated in metaphase, the

time of maximal tension, this is inconsistent with the experiment in 2F that indicates that in metaphase arrested cells (by depletion of Cdc20) SUMOylated Sgo1 is only observed under 'no tension' conditions. Further, more details are needed here. How is CDC20 inhibited? Do the authors have any control for an efficient mitotic arrest? Why use both benomyl and nocodazole combined?

We agree with the reviewer that this was not clarified in the original manuscript. As described above, our new experiments (Figure 2C and S4G) revealed that Sgo1 SUMOylation is maximal prior to the establishment of tension in metaphase, likely in prometaphase. This is fully consistent with the observations in Figure 2D where cells are arrested by depletion of Cdc20, either in the presence or absence of spindle tension. Cdc20 was depleted using a strain which carries *CDC20* under control of the methionine-repressible promoter (*pMET3-CDC20*). Cells were released from G1 (alpha factor wash-out) into medium containing methionine to repress *CDC20*. The "no tension" condition also contains benomyl and nocodazole in the release medium, while the "tension" condition contains only DMSO. This is now written in the figure legend. Tubulin immunofluorescence and nuclear staining was used to confirm efficient mitotic arrest (see example of scoring in Figure 2C). Although not necessary for the 1.5 hour incubation used in these experiments, we combine benomyl and nocodazole routinely as we find it ensures the most robust arrest. We have briefly mentioned this in the methods section.

Fig 3E again the phenotype of *sgo1-4R* does not recapitulate *siz1/siz2* mutant indicating Siz1 and 2 play mitotic roles elsewhere. This should be more clearly acknowledged.

Throughout our revised manuscript, we have highlighted the differences in phenotypes between *sgo1-4R* and *siz1Δ siz2Δ* more strongly. Excitingly, our revisions have also provided a potential explanation for these differences. We found that several pericentromere proteins are SUMOylated (shown in new Figure S8C) and, among these, one of the CPC subunits, Bir1, showed tension-dependent changes in SUMOylation. Using previously identified SUMOylation sites on Bir1 (Esteras et al., 2017), we were able to generate a Bir1 protein with reduced SUMOylation (new Figure 8A). Our analysis of *sgo1-4R*, *bir1-3R* and non-degradable Sgo1 revealed that Siz1 and Siz2 are likely to regulate timely anaphase through SUMOylation of both Sgo1 and Bir1 and, in addition, regulating the stability of Sgo1 (Figure 8D). Based on these findings we have revised our model to indicate that SUMOylation plays a broad role in regulating anaphase onset and that although Sgo1 and Bir1 are two key effectors, other activities of Siz1/Siz2 are likely to contribute.

On Sgo1 lysine mutants: Figure S3E is not convincing, all strains look the same. The use of the deltaDB, (destruction box) is not explained in the text. Furthermore, it's not clear from 3F that the over expression of SGO1-4R rescues Siz2 overexpression less well than the wild type Sgo1 over expression. Comparing 1C with 3F, they look the same, it is mostly the single mutants that are different. This should be compared within a single experimental setup.

For clarity, we removed the experiments with Sgo1-2R from the manuscript completely. As highlighted by the reviewer, the *sgo1-2R* mutant is not particularly informative as it only slightly reduces Sgo1 SUMOylation and, presumably as a result, does not cause any obvious phenotypes on their own.

In the revised manuscript, the experiments with the Sgo1 destruction box now come only later where they are fully explained in the text. We also replaced the previous time course with a live cell imaging experiment where we additionally examined a *siz1Δ siz2Δ sgo1Δdb*

mutant. We observed that *sgo1-Δdb* did not delay metaphase on its own or exacerbate the metaphase delay in *siz1Δ siz2Δ* (Figure S4F).

As requested by the reviewer, we now compared the metaphase delay caused by overexpression of *SGO1* and *sgo1-4R*, together with the ability of overexpressed *SIZ2* to rescue this delay in the same experimental set up. This experiment is now shown in Figure S3D. In the previous experiment, there was an endogenous copy of *SGO1* in the *pGAL-sgo1-4R* strain (*pGAL-sgo1-4R* was integrated ectopically at the *LEU2* locus). In the revised manuscript, we generated a new yeast strain in which both the endogenous and ectopically expressed Sgo1 carried the 4R mutation. The side by side comparison led to the same conclusion that *SIZ2* overexpression more efficiently rescued *SGO1* overexpression than *sgo1-4R* overexpression, and the difference in timing was only significant for *SGO1* overexpression (Figure S3D).

Fig S4. Deletion of Siz1/2 does not affect ubiquitylation. This means some other factors are likely responsible. However this would not be inconsistent with over expression of Siz1 and 2 suppressing ubiquitylation? This would be a trivial explanation for suppression seen in Fig 1A.

Sgo1 is known to be a ubiquitination substrate of the APC/C (Eshleman and Morgan, 2014). We agree with the reviewer that it is possible that Siz2 overexpression regulates Sgo1 partly by causing its degradation, potentially through indirect effects on the APC/C. However, we would expect that Siz1/Siz2 increase, rather than suppress Sgo1 ubiquitination, contrary to the point made by the reviewer. Furthermore, even if Siz1/Siz2 does regulate Sgo1 stability, there is strong evidence that Siz2 does more. First, *Sgo1-dbΔ* does not show a metaphase delay on its own (Figure S4F). Second, high dosage of *SIZ2* does not efficiently rescue overexpressed *sgo1-4R* (Figure S3D).

Fig 5E. Is the frequency of switching increased in *siz1/2* and *sgo1-3R* mutant simply because the cells spend longer time in metaphase?

We thank the reviewer for this excellent point. To address this, we measured dot switching in metaphase-arrested cells (Figure S5C) and measured number of switches per min spent in metaphase. We found that switching per min in metaphase was significantly increased in *sgo1-4R* and *siz1Δ siz2Δ*. Hence, unstable biorientation is more likely to be a cause, rather than a consequence of prolonged metaphase.

Figure 6A. The SAC dependency is unconvincing. Siz1/2 mutants have an effect on biorientation (Fig 5E) yet *MAD2* deletion has barely any effect on the delay. Is this significant? Clearly there is a non-SAC dependent component to the metaphase delay in Siz1/2 mutants.

We respectfully disagree. In the *mad2Δ* time course experiment referred to (Figure 6B), Pds1 degradation is advanced in *siz1Δ siz2Δ mad2Δ* compared to *siz1Δ siz2Δ* by at least one hour: this represents a substantial rescue. The accumulation of metaphase spindles (area under the curve) is also reduced in *siz1Δ siz2Δ mad2Δ* cells compared to *siz1Δ siz2Δ*, even though, as mentioned above, *siz1Δ siz2Δ* mutants have aberrant spindle morphology which reduces the confidence in spindle scoring. Therefore, as an independent confirmation we used our live cell imaging assay (new Figure 6A), which shows that deletion of *MAD2* shortens the duration of metaphase in *siz1Δ siz2Δ* cells to a length comparable to wild type, and that this is statistically significant.

Fig 6B, here the *ipl1* and *ipl1 suz1/2* mutants are deemed similar. While indeed, Pds1 levels appear similar, the frequency of metaphase spindles seem very different and more so than in other instances in the paper where authors claimed differences. It appears that the *suz1/2* mutations are suppressed by the additional *ipl1* mutation indicating that some of the *siz1/2* delay is caused by error correction in a *Siz1/2* independent manner.

As described in the response to reviewer 1, point 6, *ipl1-as* cells displayed aberrant spindle morphology, confounding accurate scoring of metaphase. We removed this experiment and replaced it with an experiment in which Bir1 was conditionally degraded and which clearly shows that functional CPC contributes to the metaphase delay of *siz1Δ siz2Δ* cells (Figure 6C).

Fig 6D. Given that Ipl1 levels are increased and Ipl1 homolog AurkB has been reported to be SUMOylated which was found to regulate its levels (Fernández-Miranda et al., 2010), it should be tested whether the metaphase delay could be a result of direct SUMOylation and stabilization of Ipl1.

As suggested, we tested whether Ipl1 is SUMOylated *in vivo* in budding yeast and found this to be the case, as in other organisms. However, Ipl1 SUMOylation status was unchanged with tension (Figure S8C). We also assessed Ipl1 levels in a time course but found that Ipl1 levels were unchanged during the metaphase-anaphase transition in both wild type and SUMO-deficient mutants (Figure S6F). Therefore, it is unlikely that SUMOylation regulates Ipl1 levels during the metaphase-anaphase transition in budding yeast. Instead, excitingly, we found that SUMOylation of the CPC subunit Bir1 contributes to promoting timely anaphase, in conjunction with the effects on Sgo1. This new data is shown in Figure 8.

Similarly, 7A/B. Rts1 shows reduced binding to Sgo1 when Sgo1 is SUMOylated. One reason for this might be NOT because RTS1 cannot bind to SGO1-SUMO but because Rts1 is itself SUMOylated. Are there any higher molecular weight species seen that could indicate this?

We apologise for the lack of clarity in our original explanation of Figure 7A and B. In this experiment, Sgo1-V5 purified from yeast cells and immobilized on beads was subjected to *in vitro* SUMOylation reaction by addition of E1, E2, E3, SUMO and ATP (as in Figure 2B). The components of the SUMO reaction were removed by stringent washes of Sgo1-V5-SUMO-bound beads. Only then was yeast lysate containing tagged Rts1 incubated with Sgo1-SUMO-beads. Hence, Rts1 could not be SUMOylated in this experimental set-up. We improved the labelling in Figure 7A and clarified this point in the text which now reads “*First we subjected purified V5-tagged Sgo1 on beads to in vitro SUMOylation (alongside a -ATP unmodified control). Beads were then washed extensively to remove SUMO enzymes before incubating with yeast extracts containing Myc-tagged Rts1.*” We also now show the upper portion of the Rts1 gel where, as expected, no SUMO bands were apparent (Figure 7A).

Incidentally, we also probed for SUMOylated Rts1 in the His-SMT3 pull-down and found no evidence that Rts1 is an *in vivo* SUMO substrate in metaphase-arrested cells (Figure S8C).

Fig 7C. The model that SUMOylation interferes with binding of PP2A-Rts1 to Sgo1 is central to this paper. It is supported by the immunofluorescence data in this figure, and *in vitro* SUMOylation assays. The native mass spec data in Figure S7 is less clear as it indicates

general binding defects of the Sgo1-4R mutant, possibly due to tagging as the authors indicate making it difficult to determine if there is indeed a loss in Rts1 binding. It would strengthen this paper greatly if a more direct *in vivo* co-IP could be performed in WT and sgo1-4R conditions to detect interaction between Sgo1 and PP2A-RTS1, e.g. using different tags and by direct blotting for PPs1/Rts1.

We performed the Co-IP experiment as suggested (Figure S7D). This experiment uses TAP-tagged Sgo1 and cells were grown at a lower temperature than in the mass spectrometry experiment. We observed an increase in Rts1 co-purifying with Sgo1 in *siz1Δ siz2Δ*, although the level of Sgo1 pulled down was also increased due to the stabilizing effect of Siz1/2 on the protein. *sgo1-4R*, on the other hand, did not reproducibly increase Rts1 binding in our two replicates. We think this could be because only a small pool of Sgo1 is SUMOylated, and therefore it is unlikely that global Sgo1-Rts1 binding could be impacted by the mutations. Interestingly, overexpression of *SIZ2* (using *pCUP1-SIZ2* +Cu as described in Figure 1C) reduced Sgo1-Rts1 binding in the Co-IP experiment. This data, shown with the mutants in Figure S7D is consistent with our *in vitro* pulldown which showed that SUMOylation disrupts Sgo1-Rts1 interaction. It is also consistent with the observation that Rts1 levels at kinetochores are elevated in both *siz1Δ siz2Δ* and *sgo1-4R* cells (Figure 7D).

Fig 7F,G. While switching frequency is used as a measure of stable biorientation, it would help to determine whether Rts1 tethering also lead to a metaphase delay?

Thank you for this excellent suggestion. We performed this experiment (now shown as new Figure S7G) and found that both Sgo1 (and Sgo1-4R) tethered to RTS1 caused a moderate but significant delay in metaphase, similar to that of sgo1-4R mutant.

Figure S5A. (A) The SPB separation appears to happen faster in the sgo1 mutant although the initial establishment of biorientation is delayed as shown in (B). This appears add odds. Can this be explained?

SPB separation is shown in this experiment as a control to ensure that any observations of slow biorientation (separation of *CEN4-GFP* foci) was not a result of slow entry into metaphase, therefore fast entry into metaphase in *sgo1Δ* does not alter the conclusions. Nevertheless, this is unexpected and we are grateful to the reviewer for pointing it out. On closer inspection, we realised that the fast SPB separation was caused by a small number of cells skewing the mean value of SPB distance. We analysed more cells and the average SPB separation is now comparable with the other strains, though *CEN4-GFP* separation is still delayed and reduced in the *sgo1Δ*, as expected and as has been well-established by other groups in addition to our own (Indjeian et al., 2005; Verzijlbergen et al., 2014; Peplowska et al., 2014).

Minor comments

Top of page 6. CDC55 is not explained or referenced.

This section now reads “Deletion of *CDC55*, a PP2A-regulatory subunit that was previously shown to rescue the metaphase delay of *pGAL-SGO1* (Clift et al., 2009), also rescued the metaphase delay in *siz1Δ siz2Δ* (Figure S1B)...”

Fig S3A does not seem useful to show as the experiment did not work. It would be better to simply explain in the lack of SUMO MS data in the text

We have removed Figure S3A and added the following explanation in the text “However, despite extensive efforts, we were unable to confidently identify any SUMOylation sites, for reasons that are unclear.”

Figure 7A labelling of panels is shifted

We have revised this panel as described above.

Figure S5A. Typo in figure legend "...and initial establishment of biorientation for the experiment (missing "B") shown in Figure 5C-F...."

Thank you, corrected.

Top of page 11: "...suggesting that SUMOylation and CPC work in the same pathway to regulate the metaphase-anaphase transition."

The argumentation here is a bit confusing. I think it makes more sense to state that the anaphase delay by *siz1/2* is in part imposed by *ipl1* i.e. error correction.

We modified the conclusion as suggested (note that this experiment has been replaced by an experiment using conditional degradation of *Bir1*). This sentence now reads: “*Conditional inactivation of CPC component Bir1 (using the auxin-induced degron) in a strain carrying Cdc14-GFP Tub1-YFP partially but significantly rescued the metaphase delay in siz1Δ siz2Δ (Figure 6C), suggesting that the delay in siz1Δ siz2Δ is at least in part imposed by CPC-dependent error correction.*”

Page 11 middle Figure S6D and E should be C and D

Addition of a new panel means that Figure S6D and E is now correct.

Reviewer #3 (Comments to the Authors (Required)):

Sister chromatids biorientation during mitosis is crucial for accurate chromosome segregation. Surveillance pathways correct erroneous kinetochore-microtubule attachments through tension sensing and arrest cell cycle until biorientation is established. The budding yeast Shugoshin (Sgo1) protein promotes biorientation by recruiting condensin and PP2A-Rts1 and maintaining the CPC at centromere as well as pericentromere regions. Once tension is achieved, Sgo1 and its associated effectors are removed from pericentromere to promote cell cycle progression. Consistent with this, SGO1 overexpression results in a pronounced metaphase delay. This work deals with Sgo1 turnover during mitotic progression. Su et al. started from an unbiased genetic screen for negative regulators of Sgo1 and identified the SUMO ligase Siz2. They showed that Siz1/Siz2 promotes efficient anaphase onset partially by antagonizing Sgo1-mediated pathways but not affecting Sgo1 localization. Specifically, Siz1/Siz2 sumoylates Sgo1 to reduce its interaction with PP2A-Rts1, and this may in turn lower centromeric CPC, although mechanism for CPC removal has not been investigated here.

Overall, this study provides a number of new findings to address how protein sumoylation

may regulate Shugoshin to modulate the kinase-phosphatase network. However, a number of concerns should be addressed, as summarized below:

Thank you to the reviewer for their support and helpful suggestions.

Major points:

1. Figure 1B and 1C, since Siz1 over-expression was not analyzed, it is improper to assume that its over-expression can mimic that of Siz2. Despite that Siz1 and Siz2 are considered paralogs, they do possess differences. In the same vein, *siz1* and *siz2* single deletion mutants should have been analyzed in Figure 1D/E/F to demonstrate their redundant role in this process.

Thank you for this suggestion. In the revised manuscript, we show that *siz1Δ* and *siz2Δ* single mutants each reduced but did not abolish Sgo1 SUMOylation (new Figure 2A). Meanwhile, they both showed a mild but significant delay in metaphase (Figure 1F).

2. Figure 2A is trivial, consider removal. Figure 2B should include the effect of deleting either Siz1 or Siz2 to verify their possible redundancy towards Sgo1. Why is the contaminant Sgs1-HA (marked by asterisk) in the Ni-elution not the same in the same experiment, given the same amount of Sgs1-HA in the input? Reproducibility issue?

We removed the schematic previously shown in Figure 2A. As described in our response to point 1, we find that Sgo1 SUMOylation is reduced in the *siz1Δ* or *siz2Δ* single mutants but abolished in *siz1Δ siz2Δ* (new Figure 2A). The presence of unmodified Sgo1 arises from insoluble material contaminating the cell-free extract, because we also observed it in the no *HIS-SMT3* vector control. There was some variation from experiment to experiment for reasons that are unclear, but all experiments always included all relevant controls and differences in SUMO bands were highly reproducible between conditions. At least two replicates of each Ni-NTA pulldown were performed, with a representative experiment shown.

3. The fluctuation of Sgo1 protein levels during the cell cycle may complicate interpretation of the timing of Sgo1 SUMOylation (Figure 2D). This issue could be addressed by the *sgo1-Δdb* mutant, which does not impair the metaphase-anaphase transition.

Thank you for this excellent suggestion. We performed SUMO pull-down in *sgo1-Δdb* cells after G1 release (new Figure S4G). This showed that although Sgo1-*dbΔ* levels remain relatively constant from 15-90 minutes after release, SUMOylation was maximally detected after 45 minutes, and declined thereafter. In addition, following a suggestion from reviewer 1, we analysed Sgo1 SUMOylation in a time course as cells progressed from G1 into a metaphase arrest with tension (Cdc20 depletion). This data (new Figure 2C) shows that SUMOylation is highest prior to the establishment of the arrest in metaphase. Taken together with fact that we observe Sgo1 SUMOylation in metaphase-arrested cells with nocodazole (Figure 2D), this finding indicates that Sgo1 SUMOylation occurs when kinetochores are unattached/not under tension and when Sgo1 is associated with the chromatin (note Sgo1 is released from the chromatin under tension). These results indicate that Sgo1-SUMO likely plays its most critical role in pro-metaphase, as kinetochore-microtubule attachments are being made.

4. Figure 3: a total of six lysines exist in the region (41-108), why not mutate all of them? It is

clear from Figure 3B/3C, *sgo1-4R* mutant still retains significant amount of sumoylation and a rather modest phenotype.

We generated this mutant but, unfortunately, found that it is not a useful tool to understand the function of Sgo1 SUMOylation. Like *sgo1Δ*, *sgo1-6R* is benomyl sensitive (Figure R2A) and displays characteristics of aneuploidy (including poor mating and sporulation; not shown), suggesting that these mutations perturb the overall structure and function of the protein. Perhaps as a consequence of this, we find that, surprisingly, Sgo1-6R is highly SUMOylated (Figure R2B). SUMO is known to be promiscuous and target disordered regions, which would be increased by protein unfolding (Gärtner and Muller, 2014; Tatham et al., 2011). We obtained two further pieces of evidence that Sgo1-6R is unfolded (i) deletion of its destruction box does not stabilise it, indicating that Sgo1-6R is not recognised by the APC/C (Figure R2B, input) and (ii) Sgo1-6R fails to recruit PP2A-Rts1 to centromeres (Figure R2C), suggesting that it is also incapable of binding PP2A-Rts1. Overall, these observations indicate that Sgo1-6R is likely unfolded and therefore not a useful tool to understand the function of Sgo1 SUMOylation. Consequently, we did not pursue it further.

As a point of note: while Sgo1-6R must be SUMOylated outside the coiled-coil region, it seems likely that this is due to protein unfolding rather than biologically relevant SUMOylation because both *sgo1-4R* and Sgo1-Δ2-108 both greatly reduce SUMOylation (Figure 3B). Nevertheless, we cannot rule out the existence of biologically important SUMO sites outside the coiled coil or the potential that the lysines mutated in *sgo1-4R* promote Sgo1 SUMOylation in an indirect way. As a result, we have added the following sentence to results section page 7: “While these findings are consistent with the possibility that Sgo1 lysines 56, 64, 70 and 85 are direct conjugation sites for SUMO, we cannot currently rule out an indirect role of these residues in promoting Sgo1 SUMOylation.” We opted not to include our analysis of *sgo1-6R* in the manuscript as it is inconclusive, but provide it here for information.

Figure R2. Mutation of all 6 lysines in the coiled-coil of Sgo1 affects protein folding. (A) *sgo1-6R* is sensitive to benomyl. Strains of the indicated genotypes were spotted onto YPDA plates containing 10 μg/ml benomyl or DMSO (solvent). (B) Sgo1-6R is SUMOylated but insensitive to APC/C. Extracts from strains with the indicated Sgo1 mutations and

carrying a *7xHIS-SMT3* plasmid (AMp773) were purified over Ni-NTA resin and anti-Myc immunoblot was performed on both input and eluate. Arrows and asterisks indicate SUMO-Sgo1-6HA and unmodified Sgo1-6HA, respectively. Kar2 was a loading control. (C) *sgo1-6R* perturbs the centromeric localization of Rts1. Quantification of Rts1-GFP foci intensity in pro-metaphase cells.

5. Sgo1 is stabilized in *siz1Δ siz2Δ*, *ubc9-1* and *slx5Δ* (Fig. S1B, S4A and S4B), suggesting the SUMOylation and ubiquitination catalyzed by Slx5-Slx8 of Sgo1 are required for its efficient degradation. The possibility that Slx5 may target Sgo1 cannot be excluded, since the ubiquitination assay analyzed overexpressed Sgo1 (Figure S4C). To address this concern, chromosomal tagged Sgo1 should be analyzed instead. The result shown in Figure S3E (*sgo1-2R/Δdb*) is inconclusive since sumoylation of Sgo1 is not completely eliminated. To test whether SUMOylation of Sgo1 is a degradation-independent mechanism of Sgo1 inactivation, consider combining *sgo1-Δdb* mutant and *sgo1* sumoylation-deficient or *siz1/siz2* mutant.

We examined ubiquitination of endogenously expressed Sgo1 as suggested and found that ubiquitination levels were unchanged in the *slx5Δ* or *siz1Δ siz2Δ* backgrounds, as for ectopically expressed Sgo1 (Figure S4E). Therefore, it is likely that the effects of Slx5 and Siz1/Siz2 are indirect.

We agree with the reviewer that analysis of *sgo1-2RΔdb* is inconclusive and have removed it from the manuscript.

As suggested, we combined the Sgo1 destruction box mutations with *siz1Δ siz2Δ* (Figure S4F) and the SUMO-deficient *sgo1-4R* (Figure S8B) and examined metaphase duration by live cell imaging, but found no additive effect compared to the SUMO mutants alone. Taken together with the fact that *sgo1-Δdb* alone does not cause a metaphase delay, despite Sgo1 stabilization (Figure S4F and G), these findings argue that SUMOylation acts at least partially independently of Sgo1 degradation to promote its inactivation. Indeed, we made a new observation that Bir1 is another SUMOylation target by Siz1/2 (Figure 8A) and when combined with *sgo1-4R-Δdb*, the triple mutant causes a metaphase delay similar to *siz1Δ siz2Δ* (Figure 8D).

6. Figure 5: if *sgo1* mutant has an unstable biorientation, is this mutant sensitive to nocodazole? Does *sgo1* sumoylation-defective mutant show elevated chromosome/plasmid loss?

It is well-established that *sgo1Δ* cells are sensitive to microtubule-depolymerizing drugs such as benomyl (e.g. (Indjeian et al., 2005); Figure S5D). In contrast, we found that *sgo1-4R* and *siz1Δ siz2Δ* are not sensitive to the microtubule-depolymerising drug, and may even be marginally resistant (Figure S5D). Indeed, several pieces of evidence indicate that biorientation pathways are *hyperactive* in *sgo1-4R* and *siz1Δ siz2Δ* mutants: (i) Initial sister kinetochore biorientation is timely (Figure 5C); (ii) they exhibit switching i.e. kinetochore-microtubule attachments are unstable (Figure 5F and S5C); and (iii) they exhibit a metaphase delay that is dependent on spindle checkpoint and error correction pathways (Figure 6A-C, Figure S6A).

We performed the suggested plasmid loss assay and did not observe a significant change in the ability of *sgo1-4R* and *siz1Δ siz2Δ* mutants to retain a *CEN*-containing plasmid compared to wild type (new Figure 5H). This is consistent with our finding that a single-

labelled chromosome also segregates faithfully in these cells, albeit with a delay (Figure 5G) and our interpretation that error correction pathways are hyperactive in the absence of SUMOylation.

7. In Figure 7F, artificial tethering of Sgo1 to Rts1 increased association of CEN4-GFP, regardless of wild type Sgo1 or sgo1-4R. However, mis-segregation is increased when Rts1 is tethered to Sgo1-4R, to a greater extent than tethered to wild type Sgo1 (Figure 7G), how to explain this observation?

We believe that this is due to mild destabilization of Sgo1-4R by the addition of the GBP tag in combination with Rts1-GFP. This conclusion is based on our western blotting (Figure S7F) where we noticed that levels of Sgo1-4R-GBP were reduced in cells which also carry Rts1-GFP. Although the reason behind the destabilization remains unclear, the reduced level of Sgo1 expression is likely sufficient account for the increased mis-segregation. We note however that this, the level of mis-segregation remained relatively low compared to that expected for *sgo1Δ*.

Minor points:

1. The abstract should be re-written to clarify the authors' model of how sumoylation of Sgo1 may stabilize biorientation supported by results. The finding of CPC removal defect is modest with no mechanistic insight provided.

We have extensively revised the abstract, especially in light of the new data we generated in response to the reviewers comments.

2. The legend of Figure 1D: '*siz1Δ siz2Δ* cells are delayed in metaphase and *sgo1Δ* had no additive effect' does not match the main text '*sgo1Δ* reduced the duration of metaphase in *siz1Δ siz2Δ* cells' on Page 5. It seems that the effect of *sgo1Δ* is subtle. Another note: page and line number should be included in the text.

We changed the legend based on the reviewer's recommendations. To support the conclusion of *sgo1Δ*, we also depleted *SGO1* in one cell cycle using auxin-mediated degron and observed significant rescue of metaphase delay in *siz1Δ siz2Δ* (Figure 1G).

3. 'We independently verified this.....under tension (Figure S6D and S6E).....(Figure 6D and S6D)'. The corresponding figures should be Figure S6C and S6D, Figure 6D and S6C. Overall, supplementary figures are arranged in a haphazard way with no particular coordination with the main text, making it difficult for readers to follow.

We changed the text to match the correct figure. Figures appear in the order they are called out in the text.

References

Davis-Roca, A.C., N.S. Divekar, R.K. Ng, and S.M. Wignall. 2018. Dynamic SUMO remodeling drives a series of critical events during the meiotic divisions in *Caenorhabditis elegans*. *PLoS Genet.* 14:e1007626.

Eshleman, H.D., and D.O. Morgan. 2014. Sgo1 recruits PP2A to chromosomes to ensure sister chromatid bi-orientation during mitosis. *J Cell Sci.* 127:4974–4983.

- Esteras, M., I.C. Liu, A.P. Snijders, A. Jarmuz, and L. Aragon. 2017. Identification of sumo conjugation sites in the budding yeast proteome. *Microb. Cell.* 4:331–341. doi:10.15698/mic2017.10.593.
- Fernández-Miranda, G., I. de Castro, M. Carmena, C. Aguirre-Portolés, S. Ruchaud, X. Fant, G. Montoya, W.C. Earnshaw, M. Malumbres, I. Pérez de Castro, M. Carmena, C. Aguirre-Portolés, S. Ruchaud, X. Fant, G. Montoya, W.C. Earnshaw, and M. Malumbres. 2010. SUMOylation modulates the function of Aurora-B kinase. *J Cell Sci.* 123:2823–2833.
- Gärtner, A., and S. Muller. 2014. PML, SUMO, and RNF4: Guardians of nuclear protein quality. *Mol. Cell.* 55:1–3. doi:10.1016/j.molcel.2014.06.022.
- Indjeian, V.B., B.M. Stern, and A.W. Murray. 2005. The centromeric protein Sgo1 is required to sense lack of tension on mitotic chromosomes. *Science.* 307:130–133.
- Nerusheva, O.O., S. Galander, J. Fernius, D. Kelly, and A.L. Marston. 2014. Tension-dependent removal of pericentromeric shugoshin is an indicator of sister chromosome biorientation. *Genes Dev.* 28:1291–1309. doi:10.1101/gad.240291.114.
- Nie, M., A.A. Vashisht, J.A. Wohlschlegel, and M.N. Boddy. 2015. High Confidence Fission Yeast SUMO Conjugates Identified by Tandem Denaturing Affinity Purification. *Sci. Rep.* 5:14389.
- Pelisch, F., L. Bel Borja, E.G. Jaffray, and R.T. Hay. 2019. Sumoylation regulates protein dynamics during meiotic chromosome segregation in *C. elegans* oocytes. *J Cell Sci.* jcs.232330.
- Pelisch, F., T. Tammsalu, B. Wang, E.G. Jaffray, A. Gartner, and R.T. Hay. 2017. A SUMO-Dependent Protein Network Regulates Chromosome Congression during Oocyte Meiosis. *Mol Cell.* 65:66–77.
- Peplowska, K., A.U. Wallek, and Z. Storchová. 2014. Sgo1 regulates both condensin and ipl1/aurora B to promote chromosome biorientation. *PLoS Genet.* 10:e1004411.
- Psakhye, I., F. Castellucci, and D. Brnzei. 2019. SUMO-Chain-Regulated Proteasomal Degradation Timing Exemplified in DNA Replication Initiation. *Mol. Cell.* 76:632-645.e6. doi:10.1016/j.molcel.2019.08.003.
- Schimmel, J., K. Eifler, J.O. Sigurðsson, S.A.G. Cuijpers, I.A. Hendriks, M. Verlaan-de Vries, C.D. Kelstrup, C. Francavilla, R.H. Medema, J. V Olsen, A.C.O. Vertegaal, J.O. Sigurðsson, S.A.G. Cuijpers, I.A. Hendriks, M. Verlaan-de Vries, C.D. Kelstrup, C. Francavilla, R.H. Medema, J. V Olsen, and A.C.O. Vertegaal. 2014. Uncovering SUMOylation dynamics during cell-cycle progression reveals FoxM1 as a key mitotic SUMO target protein. *Mol Cell.* 53:1053–1066.
- Tatham, M.H., I. Matic, M. Mann, and R.T. Hay. 2011. Comparative proteomic analysis identifies a role for SUMO in protein quality control. *Sci. Signal.* 4:rs4–rs4. doi:10.1126/scisignal.2001484.
- Verzijlbergen, K.F., O.O. Nerusheva, D. Kelly, A. Kerr, D. Clift, F. de L. Alves, J. Rappsilber, and A.L. Marston. 2014. Shugoshin biases chromosomes for biorientation through condensin recruitment to the pericentromere. *Elife.* 2014. doi:10.7554/eLife.01374.

March 12, 2021

RE: JCB Manuscript #202005130R

Prof. Adele L Marston
University of Edinburgh
Wellcome Centre for Cell Biology
Max Born Crescent
School of Biological Sciences
Edinburgh EH9 3BF
United Kingdom

Dear Prof. Marston,

Thank you for submitting your revised manuscript entitled "SUMOylation stabilizes sister kinetochore biorientation to allow timely anaphase". You will see that the reviewers appreciated the revisions and are now supportive of publication. We would be happy to publish your paper in JCB pending final revisions necessary to meet our formatting guidelines (see details below) and pending responses to Reviewer #1's points in the text and manuscript. Please provide a response to the reviews at resubmission explaining any changes made to the manuscript. No further experimentation is needed.

1) eTOC summary: A 40-word summary that describes the context and significance of the findings for a general readership should be included on the title page. The statement should be written in the present tense and refer to the work in the third person.

- Please include a summary statement on the title page of the resubmission. It should start with "First author name(s) et al..." to match our preferred style.

2) JCB Articles can have up to 10 main and 5 supplementary figures. Each figure can span up to one entire page as long as all panels fit on the page. Could you please rearrange the supplemental data to try and meet this limit (e.g., by merging some of the figures and/or moving supp data to the main figures)?

Thank you in advance for your efforts.

3) Figure formatting: Scale bars must be present on all microscopy images, including inset magnifications. Please add scale bars to 1BE, 4H, 5AE, 6D, 7C

Molecular weight or nucleic acid size markers must be included on all gel electrophoresis. Please add molecular weight with unit labels on the following panels: 1D, 3D, 4EF, 6B, S1B, S4ACD, S6AEF, S7F

4) Statistical analysis: Error bars on graphic representations of numerical data must be clearly described in the figure legend. The number of independent data points (n) represented in a graph must be indicated in the legend. Statistical methods should be explained in full in the materials and methods. For figures presenting pooled data the statistical measure should be defined in the figure

legends.

5) Materials and methods: Should be comprehensive and not simply reference a previous publication for details on how an experiment was performed. Please provide full descriptions in the text for readers who may not have access to referenced manuscripts.

- For all cell lines, vectors, constructs/cDNAs, etc. - all genetic material: please include database / vendor ID (e.g., Addgene, ATCC, etc.) or if unavailable, please briefly describe their basic genetic features *even if described in other published work or gifted to you by other investigators*

- Please include species and source for all antibodies, including secondary, as well as catalog numbers/vendor identifiers if available.

- Sequences should be provided for all oligos: primers, si/shRNA, gRNAs, etc.

- Microscope image acquisition: The following information must be provided about the acquisition and processing of images:

a. Make and model of microscope

b. Type, magnification, and numerical aperture of the objective lenses

c. Temperature

d. imaging medium

e. Fluorochromes

f. Camera make and model

g. Acquisition software

h. Any software used for image processing subsequent to data acquisition. Please include details and types of operations involved (e.g., type of deconvolution, 3D reconstitutions, surface or volume rendering, gamma adjustments, etc.).

6) A summary paragraph of all supplemental material should appear at the end of the Materials and methods section.

- Please include one brief descriptive sentence per item, including supp figures.

A. MANUSCRIPT ORGANIZATION AND FORMATTING:

Full guidelines are available on our Instructions for Authors page, <https://jcb.rupress.org/submission-guidelines#revised>. **Submission of a paper that does not conform to JCB guidelines will delay the acceptance of your manuscript.**

B. FINAL FILES:

-- High-resolution figure and video files: See our detailed guidelines for preparing your production-ready images, <https://jcb.rupress.org/fig-vid-guidelines>.

****It is JCB policy that if requested, original data images must be made available to the editors. Failure to provide original images upon request will result in unavoidable delays in publication. Please ensure that you have access to all original data images prior to final submission.****

****The license to publish form must be signed before your manuscript can be sent to production. A link to the electronic license to publish form will be sent to the corresponding author only. Please take a moment to check your funder requirements before choosing the appropriate license.****

Thank you for this interesting contribution, we look forward to publishing your paper in the Journal of Cell Biology.

Sincerely,

Arshad Desai, PhD
Editor, Journal of Cell Biology

Melina Casadio, PhD
Senior Scientific Editor, Journal of Cell Biology

Reviewer #1 (Comments to the Authors (Required)):

The revised manuscript of Su, Marston and colleagues provides substantial improvements over the previous version, including additional data and a model that better fits the data provided. The addition of Bir1 as another sumoylation substrate that affects mitotic timing and the time course of Sgo1 sumoylation are both substantial additions. I now approve of the manuscript for publication in the Journal of Cell Biology. I do however have a few questions about the new experiments and some suggestions that may improve the manuscript.

Questions:

1. Does the Sgo1-4R mutant actually lead to benomyl resistance (Figure S5D)? If so, this should probably be noted instead of simply referring to the mutant as "not more sensitive" to the drug.
2. The reemergence of Sgo1 foci in new Figure 4H and 4I appears to coincide with the kinetochore foci going back to a single focus (at 70') from two foci (at 50' and 60') in the example provided. If this is consistently seen, it would suggest that the reversion from a bioriented state ("switching") is not necessarily chromosome-autonomous as implied by the authors. As a related point, is it known if the detachment of a single chromosome is sufficient to create a detectable amount of Sgo1 signal?

Suggestions:

3. I think the manuscript would benefit with some tightening of the message. For example, the degradation of Sgo1 does not seem to play a significant role in Sgo1 regulation by sumoylation and is some of the weakest data in the manuscript. I think the main messages would come across better without the emphasis on degradation.

4. Similarly, the targeting of Sgo1 directly to the kinetochore via Mtw1 does not add to the main message of the manuscript. However, if it is included, I would suggest adding the Mtw1-GFP control back to the graph (as was included in the first submission).

5. I think the y-axis of the kinetochore localization graphs (6E and S6C) should go down to zero.

Reviewer #2 (Comments to the Authors (Required)):

The revised manuscript by Su et al has been greatly extended and significantly improved. While I had several concerns initially, the authors went to great lengths to address these (as well as those of the other reviewers). Importantly, they clarified the observation that Siz1 and 2 have roles in mitosis beyond the Sgo1 SUMOylation they initially uncovered. They now extended the work and find that Siz1/2 also controls the CPC via Bir1. The current manuscript carries a wealth of new findings well supported by the data. I fully support publication.

Reviewer #3 (Comments to the Authors (Required)):

The manuscript has been much improved and it presents a large body of new findings that are useful for the research community, which is appreciated here.

2nd Revision - Authors' Response to Reviewers: March 22, 2021

Edinburgh, 22 March 2021

Dear Arshad and Melina,

Thank you very much for your positive response to our revised manuscript, we are delighted that *JCB* will publish our paper. We have made the final corrections, as summarised below.

Thank you again for your support and efficient handling of our paper.

Best wishes,
Adele

1) eTOC summary: A 40-word summary that describes the context and significance of the findings for a general readership should be included on the title page. The statement should be written in the present tense and refer to the work in the third person.

- Please include a summary statement on the title page of the resubmission. It should start with "First author name(s) et al..." to match our preferred style.

We have included this.

2) *JCB* Articles can have up to 10 main and 5 supplementary figures. Each figure can span up to one entire page as long as all panels fit on the page. Could you please rearrange the supplemental data to try and meet this limit (e.g., by merging some of the figures and/or moving supp data to the main figures)?

Thank you in advance for your efforts.

We have rearranged the figures so that our manuscript now has 10 main and 4 supplementary figure.

3) Figure formatting: Scale bars must be present on all microscopy images, including inset magnifications.

Please add scale bars to 1BE, 4H, 5AE, 6D, 7C

Molecular weight or nucleic acid size markers must be included on all gel electrophoresis. Please add molecular weight with unit labels on the following panels: 1D, 3D, 4EF, 6B, S1B, S4ACD, S6AEF, S7F

We added scale bars as requested. Scale bars were added to the original datasets for accuracy, therefore we have shown a different representative cell for some panels (but always from the same dataset where measurements were made).

Molecular weights were added as requested.

4) Statistical analysis: Error bars on graphic representations of numerical data must be clearly described in the figure legend. The number of independent data points (n) represented in a graph must be indicated in the legend. Statistical methods should be explained in full in the materials and methods. For figures presenting pooled data the statistical measure should be defined in the figure legends.

We confirmed that statistical analysis was properly documented in the figures and legends, as requested.

5) Materials and methods: Should be comprehensive and not simply reference a previous publication for details on how an experiment was performed. Please provide full descriptions in the text for readers who may not have access to referenced manuscripts.

- For all cell lines, vectors, constructs/cDNAs, etc. - all genetic material: please include database / vendor ID (e.g., Addgene, ATCC, etc.) or if unavailable, please briefly describe their basic genetic features *even if described in other published work or gifted to you by other investigators*

- Please include species and source for all antibodies, including secondary, as well as catalog numbers/vendor identifiers if available.

- Sequences should be provided for all oligos: primers, si/shRNA, gRNAs, etc.

- Microscope image acquisition: The following information must be provided about the acquisition and processing of images:

a. Make and model of microscope

b. Type, magnification, and numerical aperture of the objective lenses

c. Temperature

d. imaging medium

e. Fluorochromes

f. Camera make and model

g. Acquisition software

h. Any software used for image processing subsequent to data acquisition. Please include details and types of operations involved (e.g., type of deconvolution, 3D reconstitutions, surface or volume rendering, gamma

adjustments, etc.).

This information is included in the methods.

6) A summary paragraph of all supplemental material should appear at the end of the Materials and methods section.

- Please include one brief descriptive sentence per item, including supp figures.

We have included this.

A. MANUSCRIPT ORGANIZATION AND FORMATTING:

Full guidelines are available on our Instructions for Authors page, <https://jcb.rupress.org/submission-guidelines#revised>. **Submission of a paper that does not conform to JCB guidelines will delay the acceptance of your manuscript.**

B. FINAL FILES:

-- High-resolution figure and video files: See our detailed guidelines for preparing your production-ready images, <https://jcb.rupress.org/fig-vid-guidelines>.

Thank you for this interesting contribution, we look forward to publishing your paper in the Journal of Cell Biology.

Sincerely,

Arshad Desai, PhD
Editor, Journal of Cell Biology

Melina Casadio, PhD
Senior Scientific Editor, Journal of Cell Biology

Reviewer #1 (Comments to the Authors (Required)):

The revised manuscript of Su, Marston and colleagues provides substantial improvements over the previous version, including additional data and a model that better fits the data provided. The addition of Bir1 as another sumoylation substrate that affects mitotic timing and the time course of Sgo1 sumoylation are both substantial additions. I now approve of the manuscript for publication in the Journal of Cell Biology. I do however have a few questions about the new experiments and some suggestions that may improve the manuscript.

Questions:

1. Does the Sgo1-4R mutant actually lead to benomyl resistance (Figure S5D)? If so, this should probably be noted instead of simply referring to the mutant as "not more sensitive" to the drug.

We do observe slight benomyl resistance as pointed out by the reviewer and have changed the text, accordingly.

2. The reemergence of Sgo1 foci in new Figure 4H and 4I appears to coincide with the kinetochore foci going back to a single focus (at 70') from two foci (at 50' and 60') in the example provided. If this is consistently seen, it would suggest that the reversion from a bioriented state ("switching") is not necessarily chromosome-autonomous as implied by the authors. As a related point, is it known if the detachment of a single chromosome is sufficient to create a detectable amount of Sgo1 signal?

We do not currently have sufficient data to address this question, but we observe instances where the bulk kinetochore foci stay apart and also where they come together. We have been careful not to refer to this chromosome behaviour as autonomous.

Suggestions:

3. I think the manuscript would benefit with some tightening of the message. For example, the degradation of Sgo1 does not seem to play a significant role in Sgo1 regulation by sumoylation and is some of the weakest data in the manuscript. I think the main messages would come across better without the emphasis on degradation.

We have toned down the message about degradation, in particular removing it from the abstract.

4. Similarly, the targeting of Sgo1 directly to the kinetochore via Mtw1 does not add to the main message of the manuscript. However, if it is included, I would suggest adding the Mtw1-GFP control back to the graph (as was included in the first submission).

We have added the Mtw1-GFP control back.

5. I think the y-axis of the kinetochore localization graphs (6E and S6C) should go down to zero.

These are arbitrary units and so the specific values have little meaning. Starting the graph at zero would mean the data is no longer visible. To circumvent this we have included a break in the y axis to make it clear that the scale what the scale represents.

Reviewer #2 (Comments to the Authors (Required)):

The revised manuscript by Su et al has been greatly extended and significantly improved. While I had several concerns initially, the authors went to great lengths to address these (as well as those of the other reviewers). Importantly, they clarified the observation that Siz1 and 2 have roles in mitosis beyond the Sgo1 SUMOylation they initially uncovered. They now extended the work and find that Siz1/2 also controls the CPC via Bir1. The current manuscript carries a wealth of new findings well supported by the data. I fully support publication.

Reviewer #3 (Comments to the Authors (Required)):

The manuscript has been much improved and it presents a large body of new findings that are useful for the research community, which is appreciated here.